# Histone methyltransferase PRDM9 promotes survival of drug-tolerant persister cells in glioblastoma

Chemotherapy often kills a large fraction of cancer cells but leaves behind a small population of drug-tolerant persister cells. These persister cells survive drug treatments through reversible, non-genetic mechanisms and cause tumour recurrence upon cessation of therapy. Here, we report a drug tolerance mechanism regulated by the germ-cell-specific H3K4 methyltransferase PRDM9. Through histone proteomic, transcriptomic, lipidomic, and ChIP-sequencing studies combined with CRISPR knockout and phenotypic drug screen, we identify that chemotherapy-induced PRDM9 upregulation promotes metabolic rewiring in glioblastoma stem cells, leading to chemotherapy tolerance. Mechanistically, PRDM9-dependent H3K4me3 at cholesterol biosynthesis genes enhances cholesterol biosynthesis, which persister cells rely on to maintain homeostasis under chemotherapy-induced oxidative stress and lipid peroxidation. PRDM9 inhibition, combined with chemotherapy, results in strong anti-cancer efficacy in preclinical glioblastoma models, significantly enhancing the magnitude and duration of the antitumor response by eliminating persisters. These findings demonstrate a role of PRDM9 in promoting metabolic reprogramming that enables the survival of drug-tolerant persister cells.

Cancer therapy failure and tumour recurrence is, in part, attributed to a cancer cell subpopulation known as drug-tolerant persister cells[1]. Unlike resistant cells, persister cells do not possess permanent resistance-conferring genetic mutations but instead enter a reversible drug-tolerant state[2]. Upon cessation of therapy or in periods between therapy cycles (drug holiday), persister cells give rise to a population of cells that are often as sensitive to the drug as the original drug naïve population. Alternatively, overtime persister cells may acquire genetic mutations that confer irreversible drug resistance[3]. The non-genetic mechanisms reported to underlie the drug-tolerant persister phenotype include epigenetic, transcriptomic, translational and metabolic reprogramming[4].

Glioblastoma is the most common and lethal primary brain tumour. The DNA-alkylating agent temozolomide, albeit largely ineffective, remains the only approved drug for this disease[5]. Several lines of pre-clinical evidence indicate that glioblastoma cells can be effectively targeted with microtubule-targeting agents (MTAs) which disrupt fundamental cell functions such as mitosis, migration, and vesicle transport[6–9]. Moreover, glioblastoma cells are connected via ultralong thin membrane protrusions known as tumour microtubes. Microtubes provide the anatomical basis for transfer of self-renewal factors between cells and their functioning depends on high content of microtubules[10,11]. However, clinical MTAs like taxanes and vinca alkaloids have limited therapeutic value in neuro-oncology due to their large molecular size and polarity which are incompatible with diffusion across the blood-brain barrier[12].

Optical blood-brain-tumour barrier modulation to enhance paclitaxel delivery and the small-molecule MTA lisavanbulin have shown promise in glioblastoma models and trials[13–15]. These findings underscore the potential of brain-permeable MTA for glioblastoma therapy. Nonetheless, MTAs give rise to drug-tolerant persister cells[8,16]. Thus, the identification of strategies to combat drug tolerance to MTAs

✉ e-mail: lenka.munoz@sydney.edu.au

holds the potential to enhance their efficacy, which is also critical for the clinical translation of brain-permeable MTAs.

PRDM9 (PR/SET domain 9) is a zinc finger protein that binds DNA and trimethylates histone H3 on lysine 4 and 36 to form H3K4me3 and H3K36me3. PRDM9 is normally expressed only in germ cells, where it regulates DNA recombination in meiosis[17]. Following fertilisation, meiotic genes are silenced. Yet, the failure of meiotic gene-silencing is common in cancer, with over 200 cancer/testis genes, including PRDM9, found in tumours[18,19]. Pan-cancer analysis of 32 different cancer types (TCGA) revealed PRDM9 upregulation in 20% of tumours compared to healthy matching tissue, with the highest PRDM9 activity detected in glioblastoma[19,20]. PRDM9 in cancer has been linked to genomic instability, likely stemming from its known recombination function in meiosis[19,20]. However, despite PRDM9's potential as an ideal and safe therapeutic target due to its germ-cell restricted expression, the functional role of PRDM9 in cancer is unknown.

In this work, using a small-molecule microtubule-targeting agent CMPD1[21] as the lead, we develop a brain-permeable microtubule-targeting agent WJA88. In pursuit of optimising the outcomes of MTA chemotherapy, we show that PRDM9's methyltransferase activity is critical for the survival of glioblastoma persister cells. Mechanistically, PRDM9-mediated H3K4me3 maintains cholesterol homeostasis under chemotherapy stress. Moreover, inhibiting PRDM9 in combination with brain-permeable MTAs results in strong anti-glioblastoma efficacy and significantly reduces the population of persister cells, the main reason for tumour recurrence.

## Results

### Glioblastoma persisters display transcriptomic and lipidomic reprogramming

MTAs induce apoptosis in glioblastoma cells[8,21], however do not kill all cells. Using the growth-rate (GR) metrics[22], we have previously shown that small-molecule MTAs do not reach $GR_{max}$ value of −1, which denotes complete killing efficacy. Across 15 glioblastoma stem cell lines, RKI1 and FPW1 cells were least responsive to MTAs[8]. Indeed, viable RKI1 and FPW1 cells can still be detected after 14 days of treatment with high concentrations (25x $GR_{50}$) of CMPD1 or tivantinib (Fig. 1a), both molecules developed as kinase inhibitors but subsequently shown to target microtubules[21,23]. We have also shown that the size of RKI1 subpopulation surviving MTAs did not decrease when cells were co-treated with inhibitors of drug efflux pumps[8]. In this study, we observed that surviving cells were morphologically different from the parent cells. As microtubule-targeting agents induce prominent changes in cell morphology, the observed shrinkage caused by disrupted microtubule cytoskeleton can be used as a marker of continuous target engagement in cells[24,25].

We used RT-qPCR to confirm that the CMPD1-surviving cells remained in a non-proliferative state throughout the treatment period, evidenced by a decrease in replication genes and upregulation of senescence and quiescence genes, particularly *CDKN1A*- encoded p21 (Supplementary Fig. 1a, b). However, in drug holiday (*i.e.*, MTA withdrawal) surviving cells recovered their morphology, began to proliferate and re-colonized the wells within 14 days (Fig. 1b). Once a substantial population of recovered cells had been regenerated, we rechallenged these cells with CMPD1 or tivantinib. Consistent with a persister cell classification[2], RKI1 and FPW1 persisters yielded MTA-sensitive progeny (Fig. 1c; Supplementary Fig. 1c, d). These observations suggest that we did not select out a subclone with a pre-existing resistance mutation and the fractional killing efficacy of CMPD1 and tivantinib leads to drug-tolerant persisters.

To investigate persisters in an unbiased manner, we compared the transcriptomes of parent and CMPD1-derived RKI1 persister cells by bulk RNA sequencing (RNA-seq). We identified 1,573 differentially expressed genes (DEGs, $P_{adj} < 0.01$; Supplementary Fig. 1e). Gene Ontology of the top 200 DEGs identified that the down-regulated

processes were related to cell division and morphogenesis, whereas lipid localisation and transport were the most up-regulated processes (Fig. 1d). Following up on these findings, we found a distinctive lipidomic signature in CMPD1-derived persisters compared to parent cells (Fig. 1e). Significant treatment-related decreases occurred with ceramide (Cer), cholesterol (Chol) and phosphatidylserine (PS), whereas sphingomyelin (SM), lysophosphatidylethanolamine (LPE), and diacylglycerol (DG) increased in persisters (Fig. 1f). Furthermore, carnitine-linked fatty acids, which are substrates for mitochondrial β-oxidation, were more abundant in persisters (Supplementary Fig. 1f), in line with data showing the dependence of melanoma persisters on fatty acid oxidation[26]. While the transcriptomic analysis of FPW1 persister cells (Supplementary Fig. 1g) revealed several distinct GO pathways being affected by long-term CMPD1 treatment (Supplementary Fig. 1h), we found a high correlation in DEGs between RKI1 and FPW1 persisters (Fig. 1g), with 4794 overlapping DEGs (Supplementary Fig. 1i), suggesting similar transcriptomic profile in two genetically-distinct glioblastoma persister models.

### Glioblastoma persisters exhibit increased histone lysine methylation

Epigenetic reprogramming is considered as a critical mechanism driving the reversible drug-tolerant state[4] that we observed in glioblastoma persisters (Fig. 1b, c). We therefore profiled histone extracts of parent and CMPD1-derived persister cells by mass-spectrometry (Supplementary Fig. 2a) and constructed H3-centric peptide heatmaps (Fig. 2a, Supplementary Fig. 2b). This comprehensive analysis revealed deposition of replication-independent histone variant H3.3 compared to canonical H3.1/H3.2 in persisters (Fig. 2b), confirmed with immunoblotting (Fig. 2c, Supplementary Fig. 2c). H3.3 preferentially occurs at active gene bodies and has been associated with aggressive metastasis of carcinomas[27]. Further, we observed significant deacetylation occurring on K27 and K36 (Fig. 2d). In line, H3K27me3 and H3K36me3 were the most prominent modifications in persisters (Fig. 2e, f, Supplementary Fig. 2d). There was no significant modification of the H3K4 and H3K9 methylation marks, except for H3K4me3 levels being low in RKI1 persisters (Fig. 2f, g, Supplementary Fig. 2d, e).

Interestingly, immunoblotting of more distal H3K27 and H3K36 methylation marks revealed an additional faster migrating species of H3 in persisters that was absent in parent cells (Fig. 2f, arrows). This was not detected with antibodies specific for H3K4 and H3K9. Given that H3.3 variant is cleaved during senescence[28], generating a shorter H3.3 tail beginning with the amino acid T22; and that persisters had elevated H3.3 deposition (Fig. 2b, c) as well as expression of senescence associated markers (Supplementary Fig. 1a, b), we used H3 cleavage-specific antibody to demonstrate that CMPD1 treatment induced H3 tail cleavage (Fig. 2h). Analysis of H3K4me3 marks revealed that while these marks increased at earlier timepoints, they steeply decline after day 10 (Fig. 2h, Supplementary Fig. 2f), in line with data observed with persister populations (Fig. 2f). Over the course of CMPD1 treatment, we observed negligible changes in H3K9me3 abundance (Fig. 2h), in line with persisters' H3K9me3 profile determined by MS (Fig. 2a, Supplementary Fig. 2b 2d) and immunoblotting (Fig. 2f). These findings suggest that CMPD1 treatment induces H3K4, H3K27 and H3K36 methylation, accompanied by histone H3 cleavage at the T22 position. As a result, increased H3K27 and H3K36 methylation is maintained in persister cells (Fig. 2f), but H3K4 marks are reduced via histone cleavage later during the treatment (Fig. 2h). This highlights that the epigenetic landscape of persister cells at day 14 is distinct from the earlier changes, underscoring time-dependent chromatin remodelling during the development of drug tolerance. Nevertheless, the deposition of H3.3, coupled with methylation of gene-activating (H3K4, H3K36) as well as gene-repressing (H3K27) residues correlates with the extensive transcriptional changes observed in persisters (Fig. 1d, g; Supplementary Fig. 1e, g).

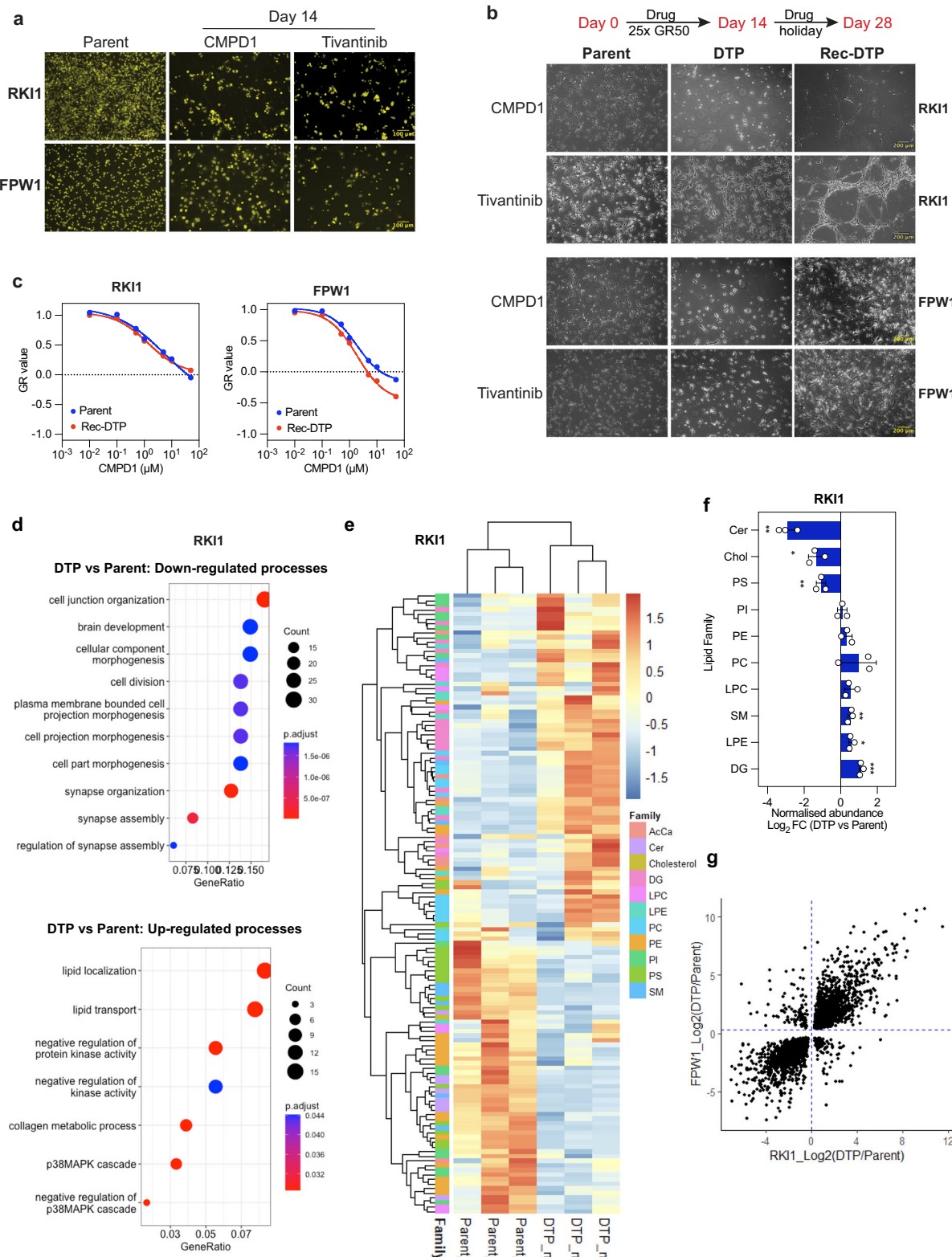

**PRDM9 inhibitor MRK-740 eliminates glioblastoma persisters**

Given the significant chromatin remodelling in persister cells, we assessed the sensitivity of MTA-tolerant cells to 37 epigenetic chemical probes (Supplementary Fig. 3a). Structurally related KMT6 (aka EZH2) inhibitors UNC-1999 and GSK-343 attenuated viability of persisters without changing the viability of the parent cells. However, GR curves for orthogonal KMT6 inhibitors UNC-1999, GSK-126, CPI-169, CPI-1205 and tazemetostat revealed minimally increased sensitivity of persisters

to KMT6 inhibitors compared to parent cells (Supplementary Fig. 3b). GSK-J4 and KDOBA-67a (targeting KDM6) and JQ1 (targeting BRD4) reduced viability of both parent and persister cells (Supplementary Fig. 3a), consistent with previous findings[29] and suggesting that they do not target a mechanism specific to MTA tolerance.

In the viability screen (Supplementary Fig. 3a), we introduced epigenetic probes to persisters after a 14-days treatment with CMPD1 or tivantinib. Considering that methylation of H3K4, H3K27, H3K36

**Fig. 1 | Characterisation of drug-tolerant persister cells in glioblastoma.**
**a** Nuclear-ID red stain images (pseudo-coloured yellow) of RKI1 and FPW1 glioblastoma stem cells treated with CMPD1 or tivantinib (25 μM). Scale bar 100 μm. Representative images on $n = 2$ biological replicates are shown. **b** Schematic and brightfield images of parent (Day 0), drug-tolerant persister (DTP) and recovered DTP (Rec-DTP) cells generated with CMPD1 or tivantinib (25 μM), followed by recovery in drug-free media (drug holiday). Scale bar 200 μm. **c** CMPD1 dose-response curves in parent and recovered DTP cells. Data are mean of $n = 3$ biological replicates, corresponding GR metrics are in Supplementary Fig. 1d. Rec-DTP cells were generated with CMPD1 (25 μM, 14 days), followed by drug holiday. **d** Gene Ontology of top 200 down- and upregulated genes in CMPD1 (25 μM, 14 days) derived drug-tolerant persister cells compared to parent RKI1 cells (RNA

sequencing of $n = 3$ biological replicates). $P$-adjusted value (Benjamin-Hochberg correction) from Fisher's exact test. **e** Heatmap of normalised lipids abundance in parent and CMPD1 (25 μM, 14 days) derived drug-tolerant persister cells ($n = 3$ biological replicates). **f** Fold-change of lipid families in CMPD1 (25 μM, 14 days) derived DTP cells compared to parent cells. Data are mean ± SD ($n = 3$ biological replicates). Multiple unpaired t-test between DTP vs Parent: * indicates $p = 0.012$, $p = 0.036$ for Chol and LPE, respectively; ** indicates $p = 0.0012$, $p = 0.005$, $p = 0.001$ for Cer, PS, SM, respectively; *** indicates $p = 0.001$ for DG. **g** Pearson correlation of DEGs in CMPD1 (25 μM, 14 days) derived drug-tolerant persister cells compared to RKI1 and FPW1 parent cells (RNA sequencing of $n = 3$ biological replicates). Source data are provided as a Source Data file.

occurs immediately after CMPD1 treatment, with proteolytic cleavage removing H3K4me3 marks (Fig. 2h), we co-treated cells with CMPD1 and inhibitors of writers/erasers of H3K4 (MRK-740, BAY-6035, KDOBA-67a); H3K27 (UNC1999, GSK-J4); H3K36 (EPZ-719) for 14 days and quantified persisters. MRK-740, an inhibitor of H3K4 methyltransferase PRDM9[30], and EPZ-719, an inhibitor of H3K36 methyltransferase SETD2, significantly reduced the number of persisters, particularly in the RKI1 cell line (Fig. 3a–d). Of note, RKI1 cells carry SETD2 R472H mutation, while FPW1 cells are SETD2 wild-type[31]. As this mutation is unique to RKI1 and not present in any of the other 11 cell lines in our QCell panel, the efficacy of EPZ-719 may represent a cell line-specific effect rather than a broadly applicable mechanism. This prompted us to focus further on PRDM9 and the anti-persister efficacy of MRK-740.

The finding that inhibition of the H3K4 methyltransferase PRDM9 with MRK-740 reduced the number of persister cells, while inhibition of H3K4 methyltransferase SMYD3 with BAY-6035 did not; and that PRDM9-specific H3K4me3 methylation is critical for persister cell survival, was unexpected, particularly given that previous studies identified H3K4me3 demethylation by KDM5 as a vulnerability in drug-tolerant persisters[2,32]. In line with these studies, we also found significantly increased KDM5A expression (Supplementary Fig. 3c) in glioblastoma persisters after 14 days of chemotherapy. However, treatment with the KDM5 inhibitor KDOBA-67a or the dual KDM5/6 inhibitor GSK-J4 failed to eliminate persister cells (Fig. 3a, b). Similarly, although the transcripts of histone-cleaving enzyme cathepsin L1 (CTSL1), which also removes H3K4me3, increased over time (Supplementary Fig. 3d), CTSL1 inhibition did not reduce the number of persister cells (Fig. 3e, f). Together, these findings suggest that early PRDM9-dependent H3K4me3 is critical for the survival of persister cells, whereas the subsequent removal of this mark - either through demethylation by KDM5 or histone cleavage by CTSL1 - is not a vulnerability of glioblastoma persister cells.

To further establish the efficacy of MRK-740, five glioblastoma stem cell lines, each with distinct combinations of genetic mutations[31] were co-treated with CMPD1 and either MRK-740 or matched inactive compound MRK-740-NC which does not inhibit PRDM9[30], followed by drug holiday (Fig. 3g). In addition to its efficacy in RKI1 cells, MRK-740 reduced the colony area of recovered persisters in FPW1, HW1, SB2b, and MMK1 cells (Fig. 3h). Of note, single MRK-740 induced G0/G1 arrest while present in the cell culture medium (Supplementary Fig. 3e, f), but cells fully recovered during the drug holiday (Supplementary Fig. 3g). MRK-740-NC had no effect on cell viability (Supplementary Fig. 3g) and did not reduce the number of persister-derived colonies (Fig. 3g, h).

To confirm PRDM9 inhibition in cells, we demonstrate that MRK-740 (3 μM), but not MRK-740-NC, decreased bulk H3K4me3 in RKI1 cells treated with CMPD1 without reducing H3K4me3 when used as a single agent (Fig. 3i, Supplementary Fig. 3h). To further validate PRDM9 inhibition by MRK-740 in cells and exclude inhibition of other H3K4 methyltransferases, we performed H3K4me3 ChIP-Seq

(chromatin immunoprecipitation followed by next-generation sequencing) analyses of RKI1 cells treated with CMPD1 ± MRK-740. The intensity of 27,749 H3K4me3 peaks (q-value < 0.05, Supplementary Data 1) increased in cells treated with CMPD1, while MRK-740, alone or combined with CMPD1, reduced the signal (Supplementary Fig. 3i). Genomic regions with the most significant PRDM9 motif matches ($P < 2.67 \times 10^{-7}$, MA1723.2, JASPAR database, Supplementary Data 2) identified that 3,872 out of 27,749 H3K4me3 peaks overlap with at least one PRDM9 motif (Supplementary Data 3). Plotting these 3,872 peaks confirmed alignment with the PRDM9 motif center and reduced H3K4me3 signal in cells treated with MRK-740 (Fig. 3j). While 80% of these peaks are located at promoters (Supplementary Fig. 3j) where H3K4me3 are also formed by the COMPASS complex[33], H3K4me3 peaks at distal intergenic sites can be attributed solely to PRDM9 activity[30]. We found near complete eradication of H3K4me3 at intergenic regions in cells treated with MRK-740 (Fig. 3k). These data unequivocally confirm inhibition of PRDM9's methyltransferase activity by MRK-740 in cells.

## PRDM9 is non-essential in glioblastoma but becomes a chemotherapy-induced vulnerability

To further establish PRDM9 as a critical methyltransferase in persister cells, we examined its expression alongside other H3K4 methyltransferases from the KMT2 family, including SETD1A (KMT2F) and SETD1B (KMT2G), both of which are components of the COMPASS complex[33]. We found that PRDM9 expression was the most upregulated among H3K4 methyltransferases in persister cells (Fig. 4a), increased in a time-dependent manner following CMPD1 treatment (Fig. 4b) and across six genetically diverse glioblastoma cell lines (Fig. 4c). Given that PRDM9 expression in germ cells is restricted to the prophase, the first phase of meiosis[34], and that microtubule-targeting agents, including CMPD1 induce mitotic arrest[21], we reasoned that PRDM9 upregulation could results from the mitotic arrest induced by CMPD1. To test this hypothesis, we used CDK1 inhibitor Ro-3306 to induce mitotic arrest (Fig. 4d)[35] and found Ro-3306 induced PRDM9 mRNA upregulation (Fig. 4e). Next, we evaluated whether the timing of MRK-740 administration affects the elimination of persister cells. We established that both simultaneous treatment with CMPD1 and MRK-740, and sequential addition of MRK-740 after CMPD1, reduced the number of persister cells. In contrast, pre-treating cells with MRK-740 before CMPD1 had no effect (Fig. 4f, g). These results suggest that PRDM9 inhibition is only effective once PRDM9 is upregulated during mitotic arrest induced by CMPD1.

Next, we knocked out PRDM9 in RKI1 cells to test whether genetic loss of PRDM9 phenocopies the anti-persister efficacy of MRK-740. As controls, we included SETD1A and SETD1B, two widely studied H3K4 methyltransferases[33]. DepMap database revealed that SETD1A is a common essential gene (1070/1095 cell lines with Chronos score ≤ -0.5), and 154 cell lines showed probability of SETD1B dependency. However, no cell lines had a probability of PRDM9 dependency (Fig. 4h), aligning with transient cytostatic efficacy of MRK-740 in

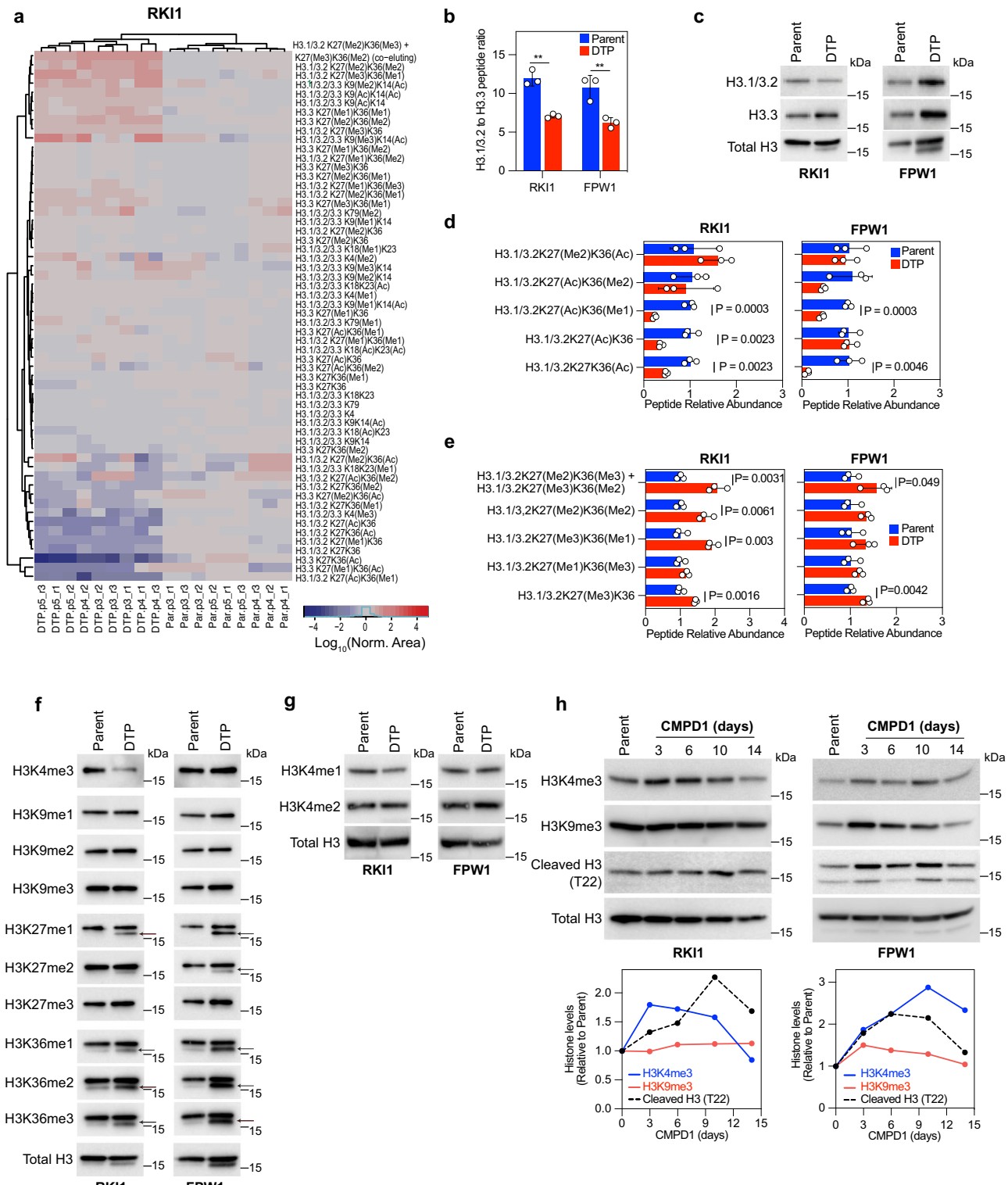

RKI1 cells (Supplementary Fig. 3f, g). Knocking down SETD1B (Fig. 4i) and PRDM9 (Fig. 4j) had no impact on RKI1 cell fitness (Fig. 4k), suggesting that these genes are not essential in unchallenged cells. However, when these cells were treated with CMPD1 for 14 days, PRDM9 knockout decreased the number of persister cells compared to parent and NTC cells, whereas SETD1B knockout had no significant effect (Fig. 4k, l). Together, these findings suggest that while PRDM9 is not an essential gene in glioblastoma, its expression increases in mitotically arrested cells and the development of a drug-tolerant state during chemotherapy is contingent on this methyltransferase.

## PRDM9-dependent H3K4me3 regulates cholesterol biosynthesis genes

To unravel the mechanism by which PRDM9 promotes survival of drug-tolerant persisters, we performed RNA sequencing (RNAseq) of our pharmacological and genetic PRDM9 inhibition models (MRK-740 and PRDM9 knockout) head-to-head. RKI1 cells were treated with CMPD1 ± MRK-740 for 3 days and harvested for analysis. In parallel, RKI1 cells transduced with NTC sgRNA (referred to as NTC cells) and PRDM9(1) sgRNA (referred to as PRDM9 KO cells) were analysed after 3 days treatment with CMPD1. Given that H3K4me3 is a transcription-

**Fig. 2 | Increased H3 methylation in drug-tolerant persister cells. a** Heatmap of peak areas for H3 peptides in parent (Par) and CMPD1 (25 µM, 14 days) derived drug-tolerant persister (DTP) cells. Data are parent-DTP pairs of 3 biological replicates performed in triplicate. **b** The sum of H3.1/3.2K27K36 peptides peak area was divided by the sum of H3.3K27K36 peptides peak area within each sample. Data are mean ± SD ($n = 3$ biological replicates). Multiple unpaired t-test between DTP vs Parent: ** indicates $p = 0.0012$, $p = 0.0096$ for RKI1 and FPW1, respectively. **c** Representative immunoblots of total H3, H3.1/H3.2 and H3.3 variants in parent and CMPD1 (25 µM, 14 days) derived drug-tolerant persister cells. Quantification of $n = 3$ biological replicates is provided in the Supplementary Fig. 2c. **d, e** Relative abundance of H3.1/3.2K27K36 peptides in parent and CMPD1 (25 µM, 14 days) derived drug-tolerant persister cells. Data are mean ± SD ($n = 3$ biological replicates;

multiple unpaired t-test). **f** Representative immunoblots of total and methylated H3 in parent and CMPD1 (25 µM, 14 days) derived drug-tolerant persister cells. Quantification of $n = 3$ biological replicates is provided in the Supplementary Fig. 2d. **g** Representative immunoblots of H3K4me1, H3K4me2 and total H3 in parent and CMPD1 (25 µM, 14 days) derived drug-tolerant persister cells. Quantification of $n = 2$ biological replicates is provided in the Supplementary Fig. 2e. **h** Representative immunoblots of H3K4me3, H3K9me3, cleaved histone H3 and total H3 in RKI1 and FPW1 cells treated with CMPD1 (25 µM). Average histone levels ($n = 3$ biological replicates for H3K9me3 in RKI1 cells; $n = 4$ for all other treatments) are shown below the immunoblots, with individual quantification provided in the Supplementary Fig. 2f. Source data are provided as a Source Data file.

activating chromatin mark[36] and aiming to identify a process exclusive to CMPD1 + MRK-740 treatment, which is absent in CMPD1 mono-therapy and thus likely to underscore MRK-740-induced persisters' death, we focused on top 50 down-regulated genes (down-DEGs, $P_{adj} < 0.05$) in cells treated with CMPD1 + MRK-740 versus CMPD1-only cells (Fig. 5a). Gene Ontology (GO) mapped these down-DEGs to cholesterol biosynthesis (Fig. 5b). In line, transcripts for numerous enzymes involved in cholesterol biosynthesis, including HMGCR, HMGCS1, MVD, EBP, SQLE, DHCR7 and DHCR24, were down-regulated upon single MRK-740 treatment (Fig. 5a). Mapping the 50 down-DEGs from Fig. 5a on the waterfall plot ranking differentially expressed genes in PRDM9 KO cells identified DHCR24, DHCR7, EBP and SEMA7A transcripts as most significantly impacted by PRDM9 loss (Fig. 5c).

Next, we reanalysed our H3K4me3 ChIP-seq datasets obtained from RKI1 cells treated with CMPD1 ± MRK-740 to determine whether PRDM9-dependent H3K4me3 peaks localise to cholesterol biosynthesis genes and regulate their transcription. First, we confirmed that the positive correlation between H3K4me3 peak width, which has been found instructive for gene transcription[36], and transcripts levels remained constant during all treatments (Supplementary Fig. 4a). Using the K-means clustering of H3K4me3 peak intensities, we divided genes in untreated RKI1 cells into Cluster 1 (genes with H3K4me3) and Cluster 2 (genes without H3K4me3; Supplementary Fig. 4b). Cluster 1 genes had significantly higher mRNA expression compared to Cluster 2 genes (Supplementary Fig. 4c). Additionally, when treated with CMPD1, Cluster 1 genes were kept to a tighter expression range compared to Cluster 2 genes (Supplementary Fig. 4d). This confirms that H3K4me3 maintains transcriptional levels and consistency in RKI1-cells, as reported previously[36].

Genome-wide analysis revealed that H3K4me3 peak intensity was highest at the promoters in CMPD1-treated cells and lowest in cells co-treated with CMPD1 and MRK-740 (Fig. 5d, Supplementary Fig. 4e, f). By plotting changes in H3K4me3 peak width at promoters of individual genes versus changes in mRNA expression, we found no correlation (Fig. 5e). This suggests that not all genes' transcription is sensitive to MRK-740 induced changes in H3K4me3 peak width. However, decreased H3K4me3 intensity (Fig. 5f) and peak width for the top 50 down-DEGs more strongly correlated with decreased mRNA levels (Fig. 5g), indicating that the transcription of cholesterol biosynthesis genes is particularly sensitive to MRK-740 treatment. To identify PRDM9-dependent H3K4me3 marks, a Venn diagram of 14,652 genes annotations with a PRDM9 motif and the 50 down-DEGs revealed that 20 genes, including cholesterol biosynthesis genes *DHCR7*, *DHCR24*, *EBP*, and *MVD*, contain the PRDM9 binding motif (Fig. 5h). The remaining 30 genes, including cholesterol biosynthesis genes *HMGCS1, HMGCR, SQLE, FDPS*, despite lacking the PRDM9 motif, could be methylated by PRDM9 via its KRAB domain interactions with CXXC1, a COMPASS complex subunit[37,38]. Individual genome tracks confirmed that *DHCR7* and *DHCR24* have PRDM9 motifs at their promoters, while *EBP* and *MVD* contain an intronic PRDM9 motif (Fig. 5i).

Importantly, reduced H3K4me3 intensity at these genes' promoters was found in cells co-treated with CMPD1 and MRK-740.

To functionally support that the observed effects of MRK-740 are truly PRDM9-dependent, we performed H3K4me3 ChIP-Seq of NTC and PRDM9 KO cells treated with CMPD1 for 3 days. We found that CMPD1 reduced total H3K4me3 in NTC cells (Supplementary Fig. 4g), an effect not seen in un-transduced RKI1 cells treated with CMPD1 (Supplementary Fig. 3i). Although the reason for the decreased H3K4me3 in CRISPR-Cas9 modified NTC cells remains unclear, the intensity of 23,073 H3K4me3 peaks ($q$-value < 0.05, Supplementary Data 4, Supplementary Fig. 4g); genome-wide H3K4me3 peaks (Fig. 6a); 1,905 H3K4me3 peaks overlapping with at least one PRDM9 motif (Fig. 6b, Supplementary Data 5); and individual H3K4me3 peaks at intergenic regions overlapping with a PRDM9 motif (Fig. 6c, Supplementary Fig. 4h) was reduced in PRDM9 KO cells, especially when treated with CMPD1. Genome tracks of cholesterol genes *DHCR7*, *DHCR24*, *EBP*, and *MVD* confirmed most significantly reduced H3K4me3 in PRDM9 KO cells treated with CMPD1 (Fig. 6d). Overall, CMPD1 treatment combined with PRDM9 loss phenocopied the results obtained with CMPD1 + MRK-740.

Our results indicate that PRDM9 regulates H3K4me3 and transcription at cholesterol biosynthesis genes. To investigate the clinical relevance of these findings, we accessed H3K4me3 ChIP-Seq data of 19 glioblastoma patients (GSE121723)[39]. Analysing two patients' datasets, we found 3,577 H3K4me3 peaks in patient AK124 and 3,565 peaks in patient AK231 to align with the PRDM9 motif centre (Fig. 6e, Supplementary Data 6-7), similar to our results with RKI1 glioblastoma cells. We leveraged parallel H3K36me3 ChIP-seq data, and overlapped intergenic H3K4me3 and H3K36me3 peaks with the PRDM9 motif (Fig. 6f) to confirm PRDM9 activity in glioblastoma, as the co-occurrence of H3K4me3, H3K36me3 and PRDM9 motif in intergenic regions is uniquely associated with PRDM9[40]. Genome wide H3K4me3 signals for each patient (Supplementary Fig. 5a) were consistent with our RKI1 cell data (Fig. 5d), and we found significantly higher H3K4me3 intensities for the top 50 down-DEGs (Supplementary Fig. 5b) compared to the top 50 upregulated genes (Supplementary Fig. 5c, d). *DHCR7* and *DHCR24* genome tracks revealed strong H3K4me3 peaks at their promoters, aligning with the PRDM9 motif (Fig. 6g), as observed in RKI1 cells (Figs. 5i, 6d). Finally, the widths of H3K4me3 peaks at the *DHCR7* and *DHCR24* promoters in 19 patients showed a positive correlation with their corresponding mRNA levels (Fig. 6h), supporting our findings obtained with RKI1 glioblastoma cells.

### PRDM9 inhibitor MRK-740 depletes cholesterol in persisters
Our data thus far suggest that PRDM9 is a vulnerability in drug-tolerant persister cells in glioblastoma. Mechanistically, we established that PRDM9 regulates transcripts of cholesterol biosynthesis enzymes (Fig. 7a) containing PRDM9 motif (*e.g.*, MVD, EBP, DHCR7, DHCR24) as well as transcripts of cholesterol enzymes lacking PRDM9 motif (*e.g.*, HMGCS1, HMGCR; Fig. 5h). Supporting this, protein levels of HMGCS1,

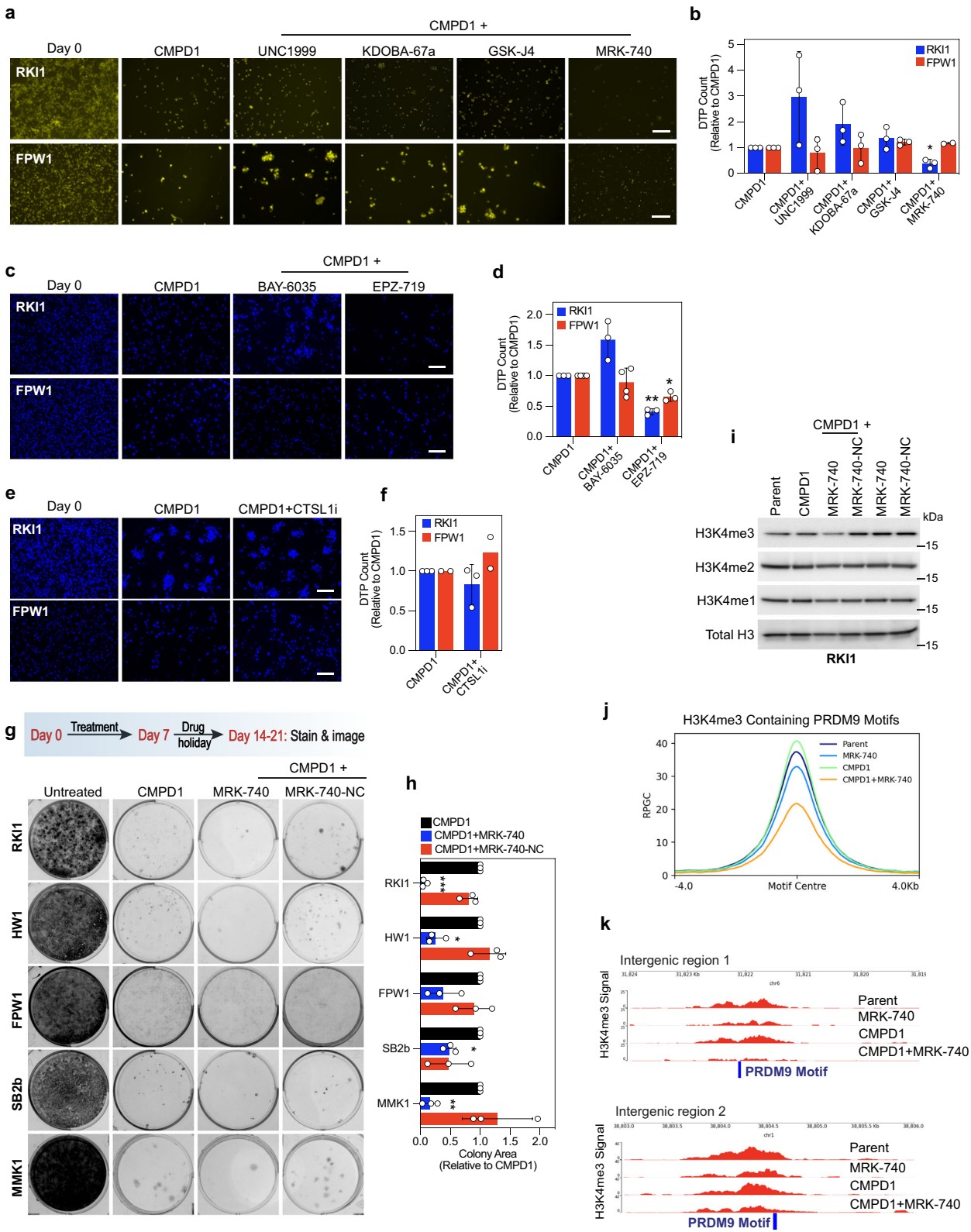

MVD and DHCR24 were reduced following PRDM9 inhibition with MRK-740 or PRDM9 knockout in RKI1 cells (Fig. 7b, Supplementary Fig. 6a), particularly in cells co-treated with CMPD1. HMGCS1, MVD and DHCR24 expression did not significantly change in FPW1 cells treated with CMPD1±MRK-740 (Fig. 7c; Supplementary Fig. 6b), consistent with the weaker anti-persister efficacy of MRK-740 in this cell line (Fig. 3h). We extended this investigation to MMK1 cells that showed high sensitivity to CMPD1 + MRK-740 (Fig. 3h) and found strongest

HMGCS1, MVD and DHCR24 reduction in co-treated cells (Fig. 7d, Supplementary Fig. 6c).

To assess how PRDM9 inhibition impacts cholesterol levels in cells, we quantified cholesterol and its intermediates in RKI1 cells treated with CMPD1 combined with MRK-740 or the inactive MRK-740-NC. Single MRK-740 reduced lathosterol, zymosterol and desmosterol (Supplementary Fig. 6d), leading to ~60% reduction in cholesterol (Fig. 7e). CMPD1 nearly depleted desmosterol (Supplementary Fig. 6d),

**Fig. 3 | PRDM9 activity is required for survival of drug-tolerant persisters.**
**a**, **b** Nuclear-ID red stain images (pseudo-coloured yellow) and quantification of drug-tolerant persister (DTP) cells surviving CMPD1 (25 μM, 14 days) treatment combined with UNC1999 (3 μM), KDOBA-67a (5 μM), GSK-J4 (5 μM) and MRK-740 (3 μM). Data are mean ± SD (*n* = 3 biological replicates). One sample t-test between co-treatment vs CMPD1: * indicates p (two tailed) = 0.0234. Scale bar = 100 μm. **c**, **d** DAPI stain images and quantification of DTP cells surviving CMPD1 (25 μM, 14 days) treatment combined with BAY-6035 (3 μM) or EPZ-719 (1 μM). Data are mean ± SD (*n* = 4 biological replicates for CMPD1 + BAY-6035 in FPW1; *n* = 3 for all other treatments). One sample t-test between co-treatment vs CMPD1: * indicates *p* (two tailed) = 0.0163, ** indicates p (two tailed) = 0.0024. Scale bar = 100 μm. **e**, **f** DAPI stain images and quantification of DTP cells surviving CMPD1 (25 μM, 14 days) treatment combined with CTSL1i (3 μM). Data are mean (*n* = 2 biological replicates) for FPW1 cells; and mean ± SD (n = 3 biological replicates) for RKI1 cells.

Scale bar = 100 μm. **g**, **h** Representative images and quantification of glioblastoma colonies treated with CMPD1 (25 μM, 7 days) ± MRK-740 (3 μM) or MRK-740-NC (3 μM), followed by drug holiday. Data are mean ± SD (*n* = 3 biological replicates). One sample t-test between co-treatment vs CMPD1: * indicates *p* (two tailed) = 0.0136, p (two tailed) = 0.0139 for HW1 and SB2b cells, respectively; ** indicates *p* (two tailed) = 0.0078 for MMK1 cells; *** indicates *p* (two tailed) = 0.0007 for RKI1 cells. **i** Representative immunoblots of H3K4 methylation in RKI1 cells treated with CMPD1 (10 μM, 3 days) ± MRK-740 (3 μM) ± MRK-740-NC (3 μM). Quantification of *n* = 3 biological replicates is provided in the Supplementary Fig. 3h. **j** Combined intensity (RPGC) of H3K4me3 peaks aligning with a PRDM9 motif in RKI1 cells treated with CMPD1 (10 μM, 3 days) ± MRK-740 (3 μM). **k** Individual genome track of H3K4me3 intensity (RPGC) at intergenic regions aligning with PRDM9 motif (blue bar) in RKI1 cells treated with CMPD1 (10 μM, 3 days) ± MRK-740 (3 μM). Source data are provided as a Source Data file.

the final cholesterol precursor in the Bloch pathway, indicating that persisters use the Kandutsch-Russell pathway. Indeed, lathosterol and cholesterol were reduced by 75%, but not depleted in CMPD1-treated cells. Importantly, cholesterol levels were further reduced in cells co-treated with CMPD1 + MRK-740 (Fig. 7e). The compound MRK-740-NC, which does not inhibit PRDM9[30], H3K4me3 (Fig. 3i) and failed to eliminate persister cells (Fig. 3g, h), did not affect cholesterol levels (Fig. 7e). PRDM9 knockout phenocopied cholesterol-lowering efficacy of MRK-740 in RKI1 cells (Fig. 7f). Surprisingly, MRK-740 despite not changing MVD and DHCR24 protein expression in FPW1 cells (Fig. 7c), reduced cholesterol levels in these cells (Fig. 7g), likely via regulating other enzymes in the cholesterol biosynthesis pathways. These findings suggest that PRDM9 regulates cholesterol biosynthesis enzymes either directly, through binding to promoter regions containing the PRDM9 motif (e.g., *MVD*, *DHCR24*), or indirectly (e.g., *HMGCS1* gene lacking PRDM9 motif). The anti-cholesterol effects of PRDM9 inhibition were more pronounced in RKI1 and MMK1 cells and less evident in FPW1 cells, correlating with the different anti-persister efficacy of MRK-740 across these cell lines. Taken together, these results support a model in which PRDM9 epigenetically regulates cholesterol biosynthesis necessary for the survival of glioblastoma persister cells.

If true, persisters should be sensitive to any agent targeting cholesterol. Statins inhibiting HMGCR in the mevalonate pathway would be the most obvious agents to test this hypothesis. However, the essentiality of mevalonate pathway[41], coupled with the fact that statins have pleiotropic cholesterol-independent activity[42], would not allow us to specifically validate cholesterol as critical to persister survival when using statins. We therefore targeted DHCR7 and DHCR24, as both are non-essential cancer genes (Supplementary Fig. 7a) and contain PRDM9 motif at their promoters (Fig. 5i). The accumulation of the DHCR7 substrate zymosterol by the DHCR7 inhibitor AY-9944, along with the accumulation of the DHCR24 substrate desmosterol by the DHCR24 inhibitor SH-42, confirmed target engagement. (Supplementary Fig. 7b). As single agents, both AY-9944 and SH-42 reduced cholesterol levels to 20% compared to untreated cells (Supplementary Fig. 7c). However, this reduction had no effect on cell viability (Supplementary Fig. 7d), indicating that in the absence of chemotherapy, cells tolerate disruptions to cholesterol biosynthesis, as observed with single MRK-740 (Supplementary Fig. 3g). Nevertheless, when combined with CMPD1, both DHCR7 and DHCR24 inhibitors significantly reduced persister-derived colonies, phenocopying CMPD1 + MRK-740 treatment (Supplementary Fig. 7d).

Of note, glioblastoma stem cells are cultured in serum-free media without an external cholesterol supply, and their survival depends on their ability to synthesise cholesterol. These culturing conditions accurately replicate the human brain environment, where the only source of cholesterol is de novo biosynthesis as dietary/peripheral cholesterol cannot penetrate the blood-brain barrier[43]. To further confirm cholesterol biosynthesis as critical to the survival of drug-tolerant persister cells, we performed rescue experiments. Adding

exogenous cholesterol to the cell culture media increased the number of CMPD1-tolerant RKI1 persister cells by 10-fold (Fig. 7h), fully rescued MRK-740-induced cytostatic effect (Supplementary Fig. 7e) and persister cell death upon PRDM9 inhibition with MRK-740 or PRDM9 knockout in RKI1 (Fig. 7h) as well as FPW1 and MMK1 (Fig. 7i) cells.

Thus, our findings confirm PRDM9-H3K4me3-dependent cholesterol biosynthesis as a vulnerability in persister cells, yet they also raise an important question: why do persister cells rely on cholesterol biosynthesis, despite their non-proliferative state generally being linked to reduced cholesterol demand[44]. In line, we observed a reduction in cholesterol precursors and final cholesterol levels in cells treated with CMPD1 for 3 days (Fig. 7e–g) and 14 days (Fig. 1f), even though the transcripts of cholesterol biosynthesis genes either remained unchanged or were upregulated by CMPD1 after 3 days (Fig. 5a) as well as after 14 days (Supplementary Fig. 6e). Given that chemotherapy, including microtubule-targeting agents induces oxidative stress which in turn oxidises lipids[9], we questioned whether cholesterol could be sequestered through oxidation during CMPD1 treatment. Indeed, reactive-oxygen species (ROS) dramatically increased within 3 days and remained high until day 14 (Supplementary Fig. 7f), leading to lipid peroxidation in CMPD1-treated cells (Fig. 7j). Upon revisiting the pathway analysis of transcriptomic data from persister cells, we observed that cholesterol homeostasis was indeed upregulated, with a stronger effect in RKI1 and a weaker disruption in FPW1 cells (Supplementary Fig. 7g). While MRK-740 did not generate ROS or exacerbate the oxidative stress caused by CMPD1, it increased the proportion of peroxidised lipids in CMPD1-treated cells, without changing the total lipid levels (Fig. 7j). To further examine the role of oxidative stress in PRDM9-dependent persisters survival, we performed rescue experiments with the antioxidant N-acetylcysteine (NAC) and observed that the addition of NAC into the cell culture media rescued the anti-persister efficacy of MRK-740 or PRDM9 knockout (Supplementary Fig. 7h-i). All in all, these data suggest that CMPD1-triggered oxidative stress leads to cholesterol oxidation. To withstand this disruption of cholesterol homeostasis, persister cells depend on PRDM9-H3K4me3 regulated de novo cholesterol biosynthesis. However, when PRDM9 is inhibited, cholesterol supply is turned off. Combined effects of CMPD1 and MRK-740 have a more pronounced impact on cholesterol homeostasis compared to each agent alone, leading to persister cell death (Fig. 7k).

## Brain permeable microtubule-targeting agent in combination with cholesterol depletion increases survival in glioblastoma xenografts

The mechanistic studies were completed with CMPD1, which is an MTA with physicochemical properties compatible with blood-brain barrier permeability[21]. However, CMPD1 exhibited short ($t_{1/2}$ = 4.6 min) half-life in the stability assay using human liver microsomes, preventing its use in vivo (Fig. 8a). To improve its metabolic stability, we performed medicinal chemistry optimisation of CMPD1 (Supplementary Table 1)

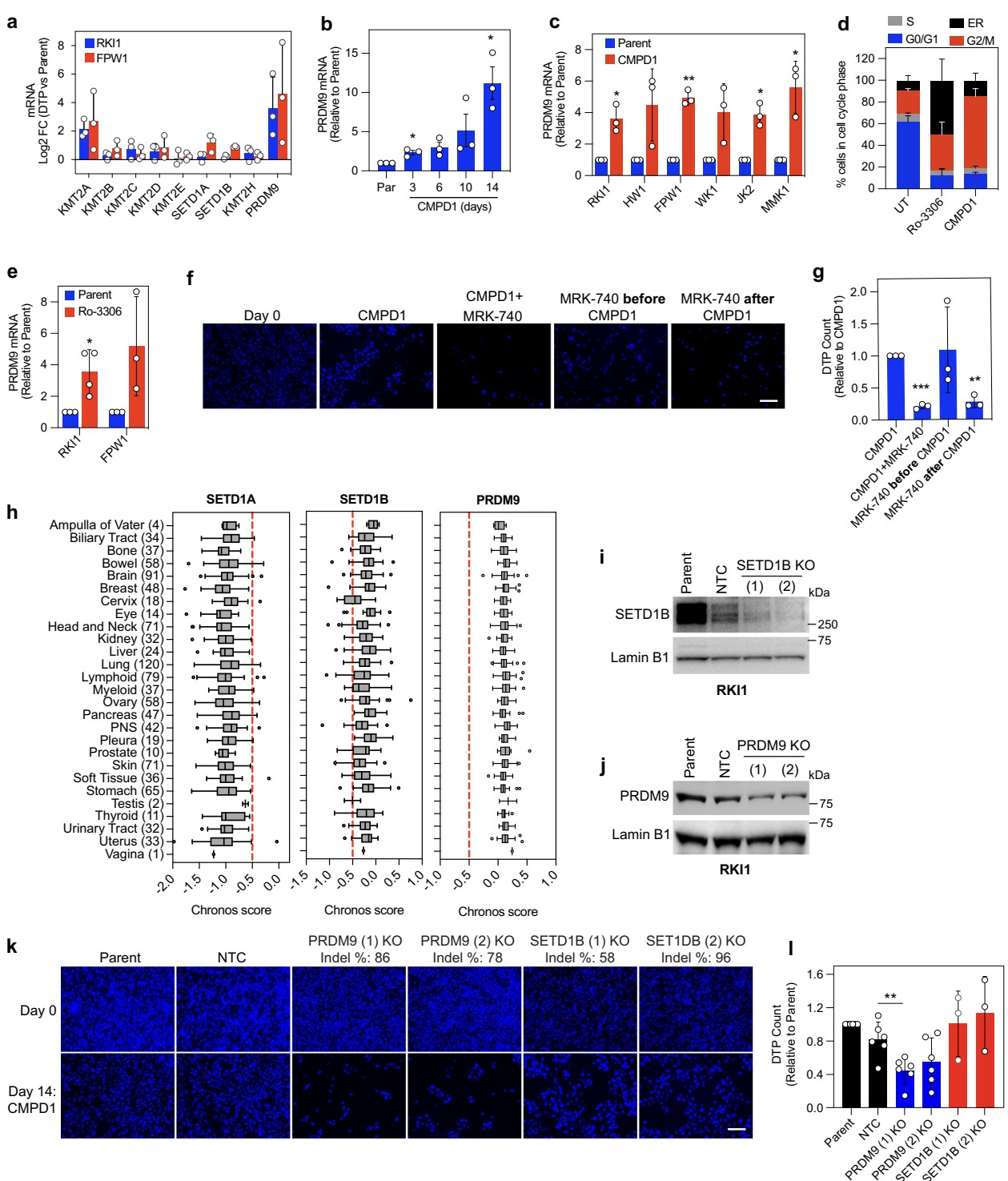

and developed analogue WJA88 with improved cellular efficacy (EC$_{50}$ = 150 nM, A172 cell viability) and metabolic stability (t$_{1/2}$ = 34.2 min) compared to CMPD1 (Fig. 8a, b). We confirmed that both CMPD1 and WJA88 inhibit tubulin polymerisation and bind into the colchicine binding site (Supplementary Fig. 8a, b). In functional assays, WJA88 inhibited proliferation of glioblastoma stem cell lines with nanomolar potency (Supplementary Table 2) and attenuated the growth of MMK1 spheroids (Supplementary Fig. 8c). Pharmacokinetic analysis of mice administered a single 50 mg/kg WJA88 dose intra-peritoneally revealed that WJA88 had an in vivo half-life (t$_{1/2}$) of 71.4 min and peak serum concentration (c$_{max}$) of 7907 ng/mL (Supplementary Table 3). Distribution across the blood-brain barrier was

evaluated in mice dosed with 50 mg/kg WJA88 once intravenously, which showed a brain/plasma ratio of 0.8 and 0.76 at 30 min and 60 min, respectively (Fig. 8c). Therefore, we reasoned that WJA88 with improved stability and sufficient brain uptake could be tested in intracranial models. We first determined WJA88 efficacy and safety in the immunocompetent somatic-cell electroporation model of glio-blastoma. In this model, tumour-inducing plasmids, knocking out three most commonly mutated genes of mesenchymal glioblastoma (*Nf1*, *Trp53*, *Pten*), were injected into the lateral ventricle of two-day old, anaesthetised mice pups, followed by electroporation of the brain; leading to the development of tumours that closely recapitulate the histopathological, molecular, and cellular features of human

**Fig. 4 | PRDM9 is a chemotherapy-induced vulnerability. a** KMT2 family mRNA in CMPD1 (25 μM, 14 days) derived drug-tolerant persister (DTP) cells compared to parent cells. Data are mean ± SD ($n = 3$ biological replicates). **b** PRDM9 mRNA in RKI1 cells treated with CMPD1 (25 μM). Data are mean ± SD ($n = 3$ biological replicates). One sample t-test between CMPD1 vs Parent: * indicates $p$ (two tailed) = 0.0382, $p$ (two tailed) = 0.0386 for 3 and 14 days, respectively. **c** PRDM9 mRNA in glioblastoma stem cell lines treated with CMPD1 (25 μM, 3 days). Data are mean ± SD ($n = 3$ biological replicates). One sample $t$-test between CMPD1 vs Parent: * indicates p (two tailed) = 0.0351, $p$ (two tailed) = 0.0205, $p$ (two tailed) = 0.0406 for RKI1, JK2 and MMK1 cells, respectively; ** indicates $p$ (two tailed) = 0.0045 for FPW1 cells. **d** Cell cycle distribution of RKI1 cells treated with Ro-3306 (10 μM) or CMPD1 (25 μM, 3 days). Data are mean ± SD ($n = 4$ biological replicates). ER: endoreduplication (DNA content > 4 N). **e** PRDM9 mRNA in cells treated with Ro-3306 (10 μM, 3 days). Data are mean ± SD ($n = 3$ biological replicates). One sample t-test between Ro-3306 vs parent: * indicates $p$ (two tailed) = 0.0359. **f, g** DAPI stained images and quantification of DTP cells surviving CMPD1 (25 μM, 14 days) treatment combined simultaneously with MRK-740 (3 μM) or added 3 days before or after CMPD1. Data are mean ± SD ($n = 3$ biological replicates). One sample t-test between co-treatment vs CMPD1: ** indicates $p$ (two tailed) = 0.0058; *** indicates p (two tailed) = 0.0007. Scale bar = 100 μm. **h** Chronos scores for SETD1A, SETD1B, and PRDM9 knockout in 1095 cancer cell lines (23Q2 + Score dataset, DepMap). Box plots display the mean as the centre line, the box representing the 25th–75th percentiles, and the whiskers extending to 1.5x the interquartile range (IQR) beyond the box limits. Outliers are shown as individual data points. **i** Representative immunoblots of RKI1 cells transduced with non-targeted control (NTC) sgRNA and orthogonal SETD1B sgRNAs ($n = 2$ biological replicates). **j** Representative immunoblots of RKI1 cells transduced with non-targeted control (NTC) sgRNA and orthogonal PRDM9 sgRNAs ($n = 3$ biological replicates). **k, l** DAPI stained images and quantification of RKI1 cells following SETD1B and PRDM9 knock-out (KO) and treated with CMPD1 (25 μM, 14 days). Data are mean ± SD ($n = 3$ biological replicates for SETD1B KO lines; $n = 6$ biological replicates for parent, non-targeted control (NTC) and PRDM9 KO lines). Unpaired $t$-test between KO lines vs NTC: ** indicates p (two tailed) = 0.0065. Scale bar = 100 μm. Source data are provided as a Source Data file.

---

glioblastoma[45,46]. Three weeks post-electroporation, mice were treated with 50 mg/kg WJA88 for 10 days and euthanised 24 h after the final dose. While WJA88 reduced weight compared to vehicle-treated mice (Supplementary Fig. 8d), mice showed overall good tolerability to WJA88. Importantly, treatment with WJA88 resulted in a significant decrease in proliferative tumour cells (Ki67⁺/tdTomato⁺, WJA88 = 10.6% vs Vehicle = 15.4%; Fig. 8d–f), confirming anti-proliferative efficacy of WJA88 in vivo.

Encouraged by in vivo efficacy of WJA88, we investigated whether it might leave behind drug-tolerant persisters cells. Similarly to CMPD1, WJA88, even at a high concentration of 25 μM, led to the emergence of persister cells, and MRK-740 significantly reduced the number of WJA88-tolerant persisters (Fig. 8g, Supplementary Fig. 8e). However, MRK-740 lacks pharmacokinetics suitable for in vivo studies[30]. As an alternative, we used the Liver X Receptor (LXR) agonist LXR-623 due to its high brain uptake in rodents[47]. Single LXR-623 (400 mg/kg) suppressed glioblastoma growth by enhancing ABCA1-mediated cholesterol efflux[47]. We confirmed that LXR-623 increased ABCA1 expression (Supplementary Fig. 8f) and reduced the number of WJA88-tolerant persisters in both RKI1 and FPW1 cells (Fig. 8g). Further, the anti-persister efficacy of LXR-623 was fully rescued by exogenous cholesterol (Supplementary Fig. 8g). Thus, LXR-623 showed a persister-killing efficacy comparable to MRK-740, albeit through different mechanisms: MRK-740 inhibits cholesterol biosynthesis, whereas LXR-623 promotes cholesterol efflux.

To assess the safety of WJA88 in combination with MRK-740 and LXR-623, we examined their impact on the viability of mouse-derived neural progenitor cells and astrocytes (Supplementary Fig. 8h). WJA88 is a microtubule-targeting agent and, like clinical MTAs paclitaxel and vinblastine, is expected to inhibit the proliferation of all dividing cells. Consistent with this, WJA88 suppressed the proliferation of neural stem cells, which are highly proliferative in culture. However, MRK-740 and LXR-623, either alone or in combination, had no effect on neural stem cell viability and did not enhance WJA88 efficacy. Importantly, in astrocytes, none of the treatments, whether alone or combined, affected cell viability. These results suggest that WJA88 may have a safety profile similar to that of clinical MTAs and that the co-treatment with cholesterol depleting agents should be well-tolerated in vivo.

Finally, to test WJA88 and LXR-623 combination in vivo, we used GBM6 cells carrying EGFRvIII mutation and forming aggressive fast-growing tumours in animals[48–50]. We first confirmed the efficacy of the co-treatment in GBM6 spheroids (Fig. 8h, i). Next, GBM6 cells were surgically engrafted into the cortex of mice, and mice were randomized to treatment cohorts on Day 5. As monotherapy, WJA88 (50 mg/kg) demonstrated short but significant extension in survival relative to vehicle control (median 31 days vs. 28 days: log-rank $P < 0.039$). LXR-623 (100 mg/kg) monotherapy had negligible effect on survival

(median 29 days). Importantly, the combination of WJA88 and LXR-623 effectively inhibited tumour growth (Fig. 8j). Reduced tumour growth translated into prolonged survival of orthotopic tumour-bearing mice treated with the combination therapy compared with monotherapies or vehicle control (median 40.5 days vs. 28 days: log-rank $P < 0.0001$; Fig. 8k). The WJA88 and LXR-623 combination was well tolerated with no significant change in body weight (Fig. 8l). In contrast, the minor reduction (up to 5%) in body weight observed with both WJA88 and LXR-623 when administered as standalone treatments was reversed when these two agents were combined. Furthermore, only mild effects on blood parameters were observed in mice receiving WJA88 and LXR-623 (Supplementary Fig. 8i). Taken together, these findings imply that the reliance on cholesterol following MTA chemotherapy was more prominent in glioblastoma cells compared with cells of other vital organs.

## Discussion

Efforts to target drug-tolerant persister cells in cancer have primarily focused on enhancing the efficacy of clinical drugs. However, effective treatments for glioblastoma are lacking, which limits drug tolerance research in neuro-oncology to experimental therapies. Here, using brain-permeable microtubule-targeting agents we show that glioblastoma persister cells rely on the histone methyltransferase PRDM9 for survival. Conversely, PRDM9 inhibition increased MTA chemosensitivity, enhancing both the magnitude and duration of the anti-tumor response by eliminating persister cells.

PRDM9 had been exclusively studied in DNA recombination during meiosis[17]. We found that PRDM9 is non-essential enzyme in glioblastoma cells, and its inhibition leads to reversible cytostatic effects. However, chemotherapy-induced mitotic arrest causes PRDM9 upregulation and under the oxidative stress of chemotherapy, PRDM9 promotes cholesterol homeostasis and persister cell survival. The variability among cell lines in cholesterol dependency was not unexpected, given the inherent heterogeneity of glioblastoma. While RKI1 persisters were highly dependent on cholesterol, this dependency appears less pronounced in FPW1 and MMK1 cell lines (Fig. 7), suggesting additional survival mechanisms beyond cholesterol. Nevertheless, in all models, cholesterol supplementation rescued the anti-persister effects of the PRDM9 inhibitor MRK-740, supporting a broader role for PRDM9 in regulating cholesterol biosynthesis and persisters survival across glioblastoma models. This mechanism may apply to other cancers as well. The observed increase in PRDM9 levels in mitotically arrested cells suggests that tumours treated with MTAs or drugs inducing mitotic arrest, such as Polo-like or Aurora kinase inhibitors, could upregulate PRDM9, enabling the emergence of persister cells and tumour recurrence. Further, our findings on the molecular and cellular functions of PRDM9 in glioblastoma expand the

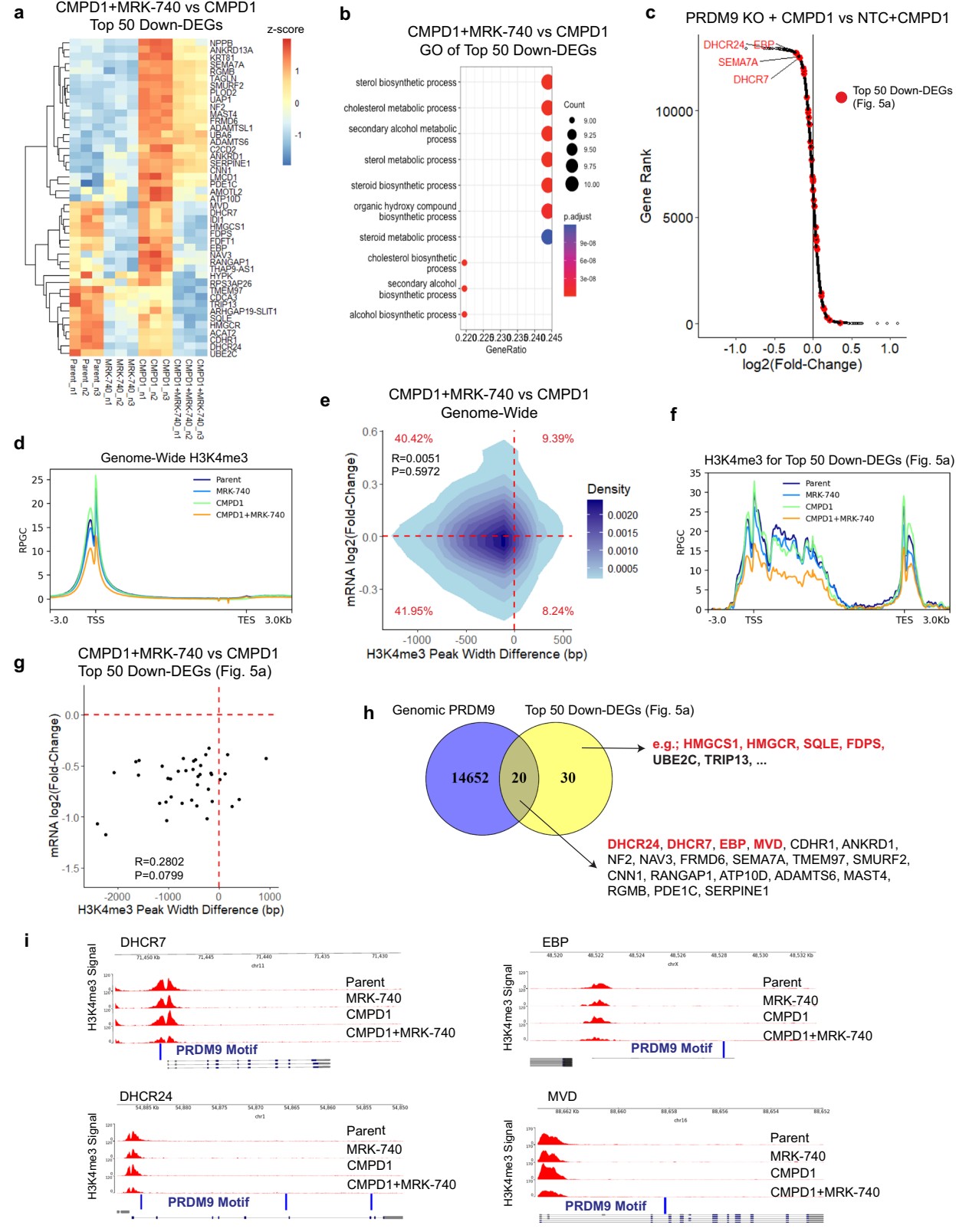

understanding of cancer-testis genes role in cancer. Notably, other cancer-testis genes, such as BORIS and MAGE-A3/6, have been also linked to drug resistance and metabolic rewiring in tumours[51,52].

A trait often observed in persister cells is the demethylation of H3K4me3 due to increased KDM5 activity[53,54]. KDM5 inhibitors, by increasing H3K4me3 abundance, reduced the number of drug-tolerant

persisters in multiple cancer types[2,32]. Like previous studies, we observed increased KDM5A transcripts and decreased H3K4me3 after 14 days of chemotherapy. Our data also suggest that the loss of H3K4me3 could stem from the proteolytic cleavage of histone tails. Histone cleavage, a well-established mechanism for epigenetic regulation, has been associated with induction of senescence[28] which is a

**Fig. 5 | PRDM9-dependent H3K4me3 marks cholesterol biosynthesis genes.**
**a**, **b** Heatmap and Gene Ontology of top 50 down-regulated DEGs (RNA sequencing of $n = 3$ biological replicates) in RKI1 cells treated with CMPD1 (10 μM, 3 days) ± MRK-740 (3 μM). *P*-adjusted value (Benjamini-Hochberg correction) from Fisher's exact test. **c** Waterfall plot of differentially expressed genes (RNA sequencing of $n = 3$ biological replicates) in CMPD1 (10 μM, 3 days) treated RKI1 cells transduced with NTC sgRNA (NTC) or PRDM9 sgRNA (PRDM9 KO). **d** Genome wide H3K4me3 signal intensity (RPGC) (± 3 kb from transcription start/end site) in RKI1 cells treated with CMPD1 (10 μM, 3 days) ± MRK-740 (3 μM). **e** 2D kernel density plot showing the relationship between changes in H3K4me3 peak width and mRNA expression in RKI1 cells treated with CMPD1 (10 μM, 3 days) ± MRK-740 (3 μM). Each data point corresponds to a gene that was detected by RNA sequencing and contained a H3K4me3 peak with > 1000 bp in CMPD1-treated cells. The colour bar

reflects the density. Pearson's product-moment correlation was used to calculate the correlation coefficient (R) and statistical significance (P). **f** H3K4me3 signal intensity (RPGC) for down-regulated genes (listed in Fig. 5a; ± 3 kb from transcription start/end site) in RKI1 cells treated with CMPD1 (10 μM, 3 days) ± MRK-740 (3 μM). **g** Scatter plot of top 39 down-DEGs listed in Fig. 5a and containing a H3K4me3 peak > 1000 bp in CMPD1-treated cells. The plot shows correlation between changes in H3K4me3 peak width and mRNA expression in RKI1 cells treated with CMPD1 (10 μM, 3 days) ± MRK-740 (3 μM). Pearson's product-moment correlation was used to calculate the correlation coefficient (R) and statistical significance (P). **h** Venn diagram of functional genomic PRDM9 annotated regions (Homo Sapiens PRDM9 matrix; Supplementary Data 2) and down-regulated genes listed in Fig. 5a. **i** H3K4me3 genome tracks for cholesterol biosynthesis genes in RKI1 cells treated with CMPD1 (10 μM, 3 days) ± MRK-740 (3 μM).

hallmark of drug-tolerant persisters. Yet, inhibiting KDM5 or the histone-cleaving CTSL1 failed to eliminate persisters, suggesting that preventing H3K4me3 loss cannot eradicate persister cells in our glioblastoma models. Instead, we show that PRDM9-dependent H3K4 methylation, occurring before histone cleavage and potential KDM5-dependent demethylation, is a transient yet critical event for establishing the drug-tolerant state. Our findings align with a study revealing that the persisters survival in breast cancer is primed through the bivalent chromatin, where genes prepared for activation by H3K4me3 are concurrently repressed by H3K27me3. Maintaining H3K27me3 with a KDM6 inhibitor, thereby blocking H3K4me3 activity, eliminated persisters[55]. Thus, in a cancer-dependent context, both reduced and elevated H3K4me3 abundance can provide a survival advantage to drug-tolerant persisters. Furthermore, our study highlights that interrogating the transient changes occurring within the first 3 days of treatment is more effective in identifying actionable vulnerabilities in persister cells than examining changes observed after 14 days. These findings also suggest that the window for effectively disrupting the persister state is narrow and may require early therapeutic intervention.

H3K4me3 marks transcription start sites and is widely believed to regulate transcriptional initiation, with recent study revealing a role for H3K4me3 in RNA polymerase pause-release and elongation[56]. Either way, H3K4me3 creates a permissive environment for gene activation during cell development, differentiation, and response to environmental cues. Additionally, H3K4me3 is recognized for its role in connecting metabolic reprogramming to epigenetic alterations. Changes in methionine availability, the precursor of S-adenosylmethionine which acts as a co-factor for lysine methyltransferases, impact H3K4me3 peak width and, consequently, gene expression[57]. In turn, H3K4me3 marks metabolic genes, contributes to a transgenerational epigenetic inheritance of obesity and regulates purine and pyrimidine nucleotide synthesis pathways[58–60]. We extend these studies by linking PRDM9-dependent H3K4me3 to the regulation of cholesterol biosynthesis genes, further supporting the H3K4me3-dependent connection between epigenetic and metabolic plasticity.

Cholesterol has been long recognised as a key factor in cancer treatment failure[47,61]. Our study uncovers a role for cholesterol biosynthesis in non-proliferating cancer persister cells, extending the view that links cholesterol metabolism largely to rapidly dividing cells. Specifically, we show that in a quiescent state induced by chemotherapy, cancer cells rely on endogenous cholesterol biosynthesis to withstand treatment-induced oxidative stress. This sustained biosynthetic capacity is most likely needed to preserve essential membrane integrity and cellular function. This finding expands the understanding of cholesterol in cancer treatment failure and suggests that targeting cholesterol may be effective against dormant, drug-tolerant cell populations. Yet, cancer clinical trials with cholesterol-lowering drugs (mostly statins) have not shown consistent or significant benefits for cancer patients. While the reasons for this are diverse, one of the issues is that statins directly (by targeting

ubiquitously expressed HMGCR) and indirectly (by lowering circulating cholesterol) reduce cholesterol levels in multiple cell types, adversely affecting membrane microdomains, steroidogenesis and cell viability[62]. The cholesterol-lowering effect of PRDM9 inhibition, combined with its tumour-restricted expression, encourages the development of PRDM9-targeting drugs that could selectively target cholesterol in tumours. To generate a proof-of-concept in glioblastoma models, we developed a brain-permeable microtubule-targeting agent WJA88 and show that a combination of WJA88 with cholesterol depletion (using the brain-permeable LXR agonist LXR-623) significantly reduced tumour growth and prolongs survival in glioblastoma-bearing mice. In summary, our findings reveal a role of the testis-specific histone methyltransferase PRDM9 in cancer. We uncover a molecular mechanism through which PRDM9 promotes survival of drug-tolerant persister cells, with important implications for enhancing the clinical effectiveness of chemotherapy with microtubule-targeting agents.

## Methods
### Ethical compliance
This research complies with all relevant ethical regulations. GBM6 xenograft studies at the Telethon Kids Institute (Perth) were approved by the Telethon Kids Institute Animal Ethics Committee (AEC 362) and conducted in accordance with the Australian Code for the Care and Use of Animals for Scientific Purposes. Under this protocol, mice with brain tumours were required to be euthanised at the first emergence of symptoms, which is considered the maximum allowable tumour burden; this limit was not exceeded. The electroporation glioblastoma model studies at QIMR Berghofer (Brisbane) were approved by the QIMR Berghofer Animal Ethics Committee (A2308-608) and performed in compliance with the same Code. In these studies, animals were euthanised immediately after the two-week treatment period, well before reaching the humane endpoint of approximately 10 weeks after tumour initiation. Experiments to derive mouse neural stem cell lines were also conducted at QIMR Berghofer (Brisbane) with approval from the QIMR Berghofer Animal Ethics Committee (A2304-602).

### Cell lines
Glioblastoma stem cell lines RKI1, FPW1, HW1, WK1, MMK1, JK2, and SB2b were derived from glioblastoma patient specimens. Patient tumour tissue was collected from patients undergoing surgery at the Royal Brisbane and Women's Hospital (RBWH), with human ethics approval from the QIMR Berghofer Medical Research Institute and RBWH Human Research Ethics Committees. All human studies have been performed in accordance with the ethical standards laid down in the 2013 version of the 1964 Declaration of Helsinki. Written informed consent from patients for the generation and use of these cell lines, including consent to publish patient information, age and gender has been obtained. These details, as well as characterisation via RNA sequencing, mutational profiling, subtype assignment and proteomic data are available online at https://www.qimrberghofer.edu.au/our-

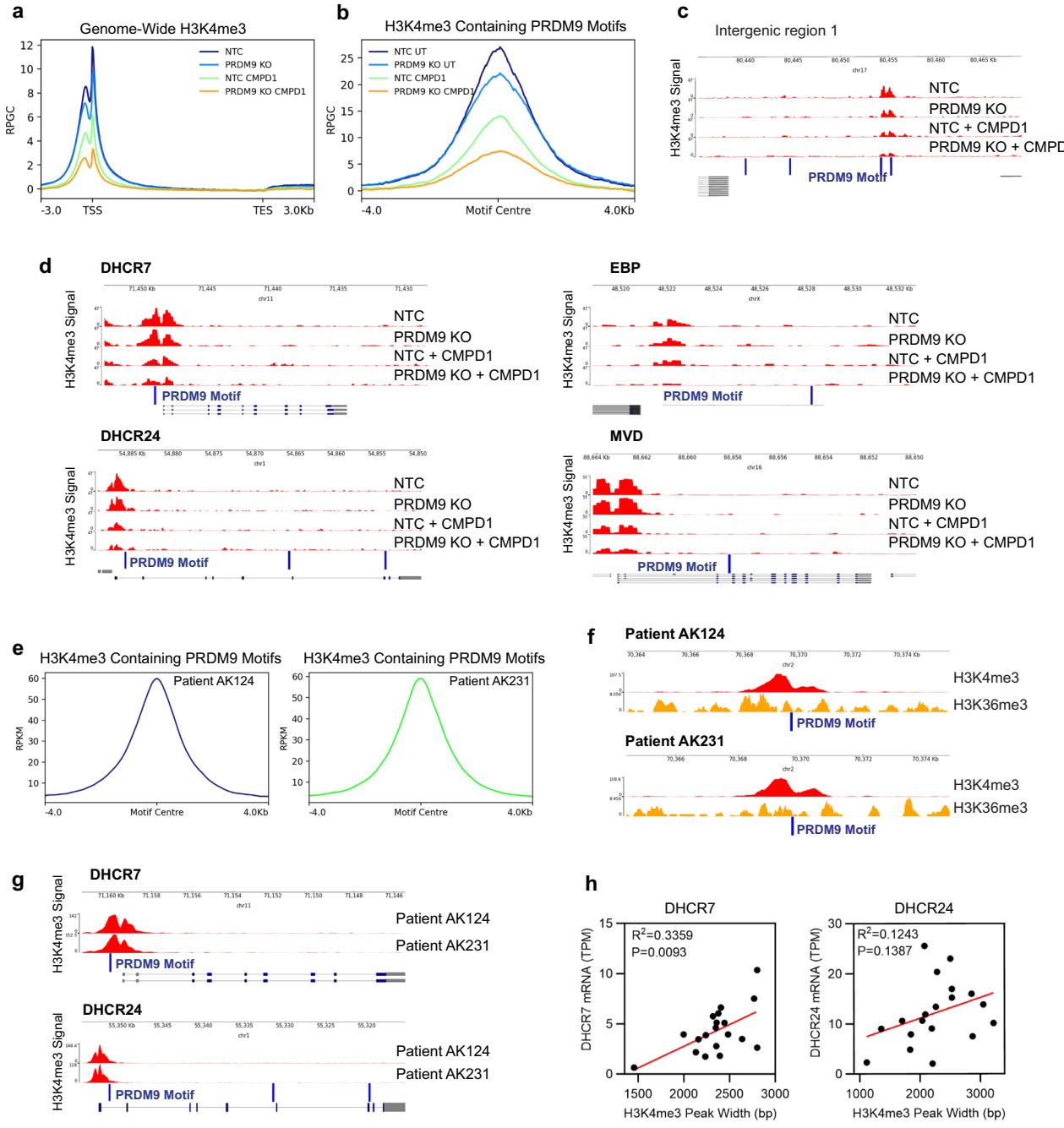

**Fig. 6 | H3K4me3 regulates transcription of cholesterol biosynthesis genes in glioblastoma. a** Genome wide H3K4me3 ChIPseq signal intensity (RPGC) (± 3 kb from transcription start/end site) in CMPD1 (10 μM, 3 days) treated RKI1 cells transduced with NTC sgRNA (NTC) or PRDM9 sgRNA (PRDM9 KO). **b** Combined intensity (RPGC) of H3K4me3 peaks aligning with a PRDM9 motif in CMPD1 (10 μM, 3 days) treated RKI1 cells transduced with NTC sgRNA (NTC) or PRDM9 sgRNA (PRDM9 KO). **c** Individual genome track of H3K4me3 intensity (RPGC) at an intergenic region aligning with PRDM9 motif (blue lines) in CMPD1 (10 μM, 3 days) treated RKI1 cells transduced with NTC sgRNA (NTC) or PRDM9 sgRNA (PRDM9 KO). **d** H3K4me3 genome tracks for cholesterol biosynthesis genes in CMPD1

(10 μM, 3 days) treated RKI1 cells transduced with NTC sgRNA (NTC) or PRDM9 sgRNA (PRDM9 KO). **e** Combined intensity (RPKM) of H3K4me3 peaks aligning with a PRDM9 motif in glioblastoma patient AK124 and AK231 (GSE121723)[39]. **f** H3K4me3 and H3K36me3 genome tracks at intergenic regions aligning with a PRDM9 motif in glioblastoma patients AK124 and AK231 (GSE121723)[39]. **g** H3K4me3 genome tracks for cholesterol biosynthesis genes in glioblastoma patients AK124 and AK231 (GSE121723)[39]. **h** Pearson's correlation showing relationship between H3K4me3 peak widths and mRNA expression for DHCR7 and DHCR24 in *n* = 19 glioblastoma patients (GSE121723)[39]. Source data are provided as a Source Data file.

research/commercialisation/q-cell/ and in ref. 31. GBM6 glioblastoma cells (EGFR/EGFRvIII over-expression) were obtained from Paul Mischel, Ludwig Institute of Cancer Research, USA. Cells were cultured in KnockOut DMEM/F-12 basal medium kit with neural supplement, EGF (20 ng/mL) and FGF-β (10 ng/mL) (ThermoFisher Scientific, Cat# A1050901). GlutaMAX-ICTS (2 mM) (ThermoFisher Scientific, Cat#

A1286001) and Antibiotic-Antimycotic (ThermoFisher Scientific, Cat# 15240112) were also added. Adherent cells were plated on flasks coated with 0.15% in PBS MatriGel Matrix (Corning Life Sciences, Cat# BDAA356237), incubated at 37 °C, 5% $CO_2$. A172 glioblastoma cell line (Cat# 88062428) was purchased from the European Collection of Authenticated Cell Cultures (ECACC, Salisbury, UK) through Cell Bank

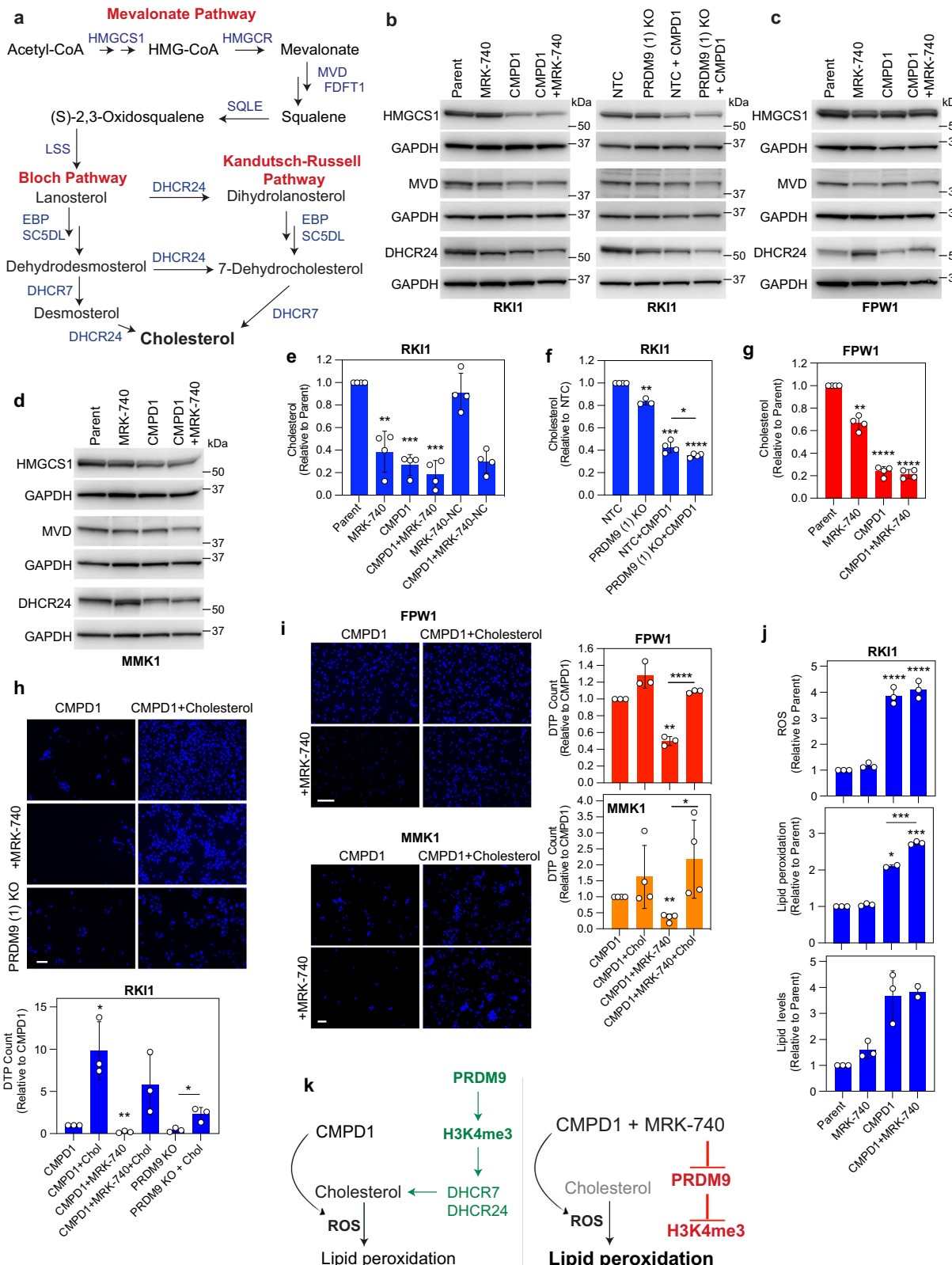

Australia and cultured as previously described in ref. 8. Mouse-derived neural stem cells (NSC) from the subventricular zone were derived at QIMR Berghofer as described in ref. 63 using 4–7 week old male and female C57Bl/6 mice from Australian BioResources (Sydney, NSW, Australia). To differentiate NSCs into astrocytes, NSCs were plated and grown for 2 days in standard culture medium prior to replacing the medium to differentiation medium – DMEM (Thermo Fisher Scientific,

Cat# 11995) supplemented with 1X N-2 supplement (Thermo Fisher Scientific, Cat# 17502048), 1X GlutaMAX (ThermoFisher Scientific, Cat# 35050061), 1% fetal bovine serum (ThermoFisher Scientific, Cat# 10099141), and 100 U/mL penicillin-streptomycin (ThermoFisher Scientific, Cat# 15140122). Cells were cultured in this medium for nine days with a 60% medium change every 2–4 days. Astrocytic phenotype was confirmed through the loss of proliferation marker Ki-67, and

**Fig. 7 | PRDM9 maintains cholesterol homeostasis under chemotherapy stress.** **a** Simplified scheme of cholesterol biosynthesis. **b** Representative immunoblots of HMGCS1, MVD and DHCR24 in RKI1 cells treated with CMPD1 (10 μM, 3 days) ± MRK-740 (3 μM; left) or transduced with PRDM9 sgRNA (PRDM9 KO, right). ($n = 2$ biological replicates for MVD; $n = 3$ biological replicates for HMGCS1 and DHCR24; quantification provided in the Supplementary Fig. 6a). **c** Representative immunoblots of HMGCS1, MVD and DHCR24 in FPW1 cells treated with CMPD1 (10 μM, 6 days) ± MRK-740 (5 μM). Quantification of $n = 4$ biological replicates is provided in the Supplementary Fig. 6b. **d** Representative immunoblots of HMGCS1, MVD and DHCR24 in MMK1 cells treated with CMPD1 (10 μM, 3 days) ± MRK-740 (5 μM). Quantification of $n = 4$ biological replicates is provided in the Supplementary Fig. 6c. **e** Cholesterol in RKI1 cells treated with CMPD1 (10 μM, 3 days) ± MRK-740 (3 μM)/MRK-740-NC (3 μM). Data are mean ± SD ($n = 4$ biological replicates). One sample t-test between treatment vs Parent: ** indicates $p$ (two tailed) = 0.0067; *** indicates $p$ (two tailed) = 0.0005, $p$ (two tailed) = 0.0009 for CMPD1 and CMPD1 + MRK-740, respectively. **f** Cholesterol in CMPD1 (10 μM, 3 days) treated RKI1 cells transduced with NTC sgRNA (NTC) or PRDM9 sgRNA (PRDM9 KO). Data are mean ± SD ($n = 3$ biological replicates for PRDM9 (1) KO; $n = 4$ biological replicates for all other conditions). One sample t-test between treatment vs NTC: ** indicates $p$ (two tailed) = 0.0086, *** indicates $p$ (two tailed) = 0.0002, **** indicates $p$ (two tailed) <0.0001. Unpaired t-test between NTC + CMPD1 vs PRDM9 (1) KO + CMPD1: * indicates p (two tailed) = 0.033. **g** Cholesterol in FPW1 cells treated with CMPD1 (10 μM, 6 days) ± MRK-740 (5 μM). Data are mean ± SD ($n = 4$ biological

replicates). One sample t-test between treatment vs Parent: ** indicates $p$ (two tailed) = 0.0018; **** indicates p (two tailed) <0.0001. **h** DAPI-stained images and quantification of DTPs in RKI1 cells treated with CMPD1 (25 μM, 14 days) ± MRK-740 (3 μM) ± cholesterol (5 μg/mL). Bottom two images are of RKI1 cells transduced with PRDM9 sgRNA (PRDM9 KO). Data are mean ± SD ($n = 3$ biological replicates). One sample t-test between treatment vs CMPD1: * indicates p (two tailed) = 0.0473, ** indicates $p$ (two tailed) = 0.0071. Unpaired t-test between PRDM9 KO vs PRDM9 KO +Chol: * indicates p (two tailed) = 0.0215. Scale bar = 100 μm. **i** DAPI stained images and quantification of DTPs in FPW1 and MMK1 cells treated with CMPD1 (25 μM, 14 days) ± MRK-740 (5 μM) ± cholesterol (5 μg/mL). Data are mean ± SD (n = 3 biological replicates for FPW1 cells, n = 4 biological replicates for MMK1 cells). One sample t-test between treatment vs CMPD1: ** indicate $p$ (two tailed) = 0.0057, $p$ (two tailed) = 0.0014 for FPW1 and MMK1 cells, respectively. Unpaired t-test between CMPD1 + MRK-740 vs CMPD1 + MRK-740+Chol: * indicates $p$ (two tailed) = 0.0244; **** indicates $p$ (two tailed) <0.0001. Scale bar = 100 μm. **j** Quantification of reactive oxygen species (ROS), lipid peroxidation and total lipids in RKI1 cells treated with CMPD1 (10 μM, 3 days) ± MRK-740 (3 μM). Data are mean ± SD (n = 3 biological replicates). One sample t-test between treatment vs Parent: * indicates p (two tailed) = 0.0159, *** indicates $p$ (two tailed) = 0.0002, **** indicates $p < 0.0001$. Unpaired t-test between CMPD1 vs CMPD1 + MRK-740 for lipid peroxidation: *** indicates $p$ (two tailed) = 0.0005. **k** Working model of PRDM9-dependent cholesterol biosynthesis and persister survival. Source data are provided as a Source Data file.

morphological changes visualised through glial fibrillary acidic protein expression. No cell line authentication was performed for any models used in this study. All cell cultures were routinely tested for mycoplasma infection, and the cumulative length of culturing did not exceed 10 passages.

## Chemical probes and inhibitors

CMPD1 (CAS# 41179-33-3, Cat# sc-203138) was purchased from Santa Cruz. Colchicine (CAS# 64-86-8, Cat# 1364), paclitaxel (CAS# 33069-62-4, Cat# 1097) and vinblastine (CAS# 143-67-9, Cat# 1256) were purchased from Tocris. SGC Probe Set (Supplementary Fig. 3a; Cat#17748), CPI-169 (CAS# 1450655-76-1, Cat# 18299), EPZ6438 (tazemetostat, CAS# 1403254-99-8, Cat# 16174), GSK126 (CAS# 1346574-57-9, Cat# 15415), AY-9944 (CAS# 366-93-8, Cat# 14611), SH-42 (CAS# 2143952-36-5, Cat# 34677), BAY-6035 (CAS# 2247890-13-5, Cat# 25925-10MG) and Ro-3306 (CAS# 872573-93-8, Cat# 15149) were purchased from Cayman Chemicals. MRK-740 (CAS# 2387510-80-5, Cat# HY-114209) and EPZ-719 (CAS# 2697176-16-0, Cat# HY-139626) were purchased from MedChem Express. Tivantinib (CAS# 905854-02-6, Cat# S2753) was purchased from Selleckchem. CPI-1205 (CAS# 1621862-70-1, Cat# A16357) was purchased from AdooQ Bioscience. MRK-740-NC (CAS# 2421146-31-6, Cat# SML2536) and Cathepsin L Inhibitor I (CTSLi) (CAS#108005-94-3, Cat#219421) was purchased from Sigma Aldrich. LXR-623 (CAS# 875787-07-8, Cat# 21117) was purchased from Sapphire Biosciences. Cholesterol complexed to methyl-beta-cyclodextrin (Cat# C4951) was purchased from Sigma Aldrich.

## Animal models

**GBM6 xenografts.** BALB/c Nude mice (Animal Resource Centre, Perth, Western Australia) were housed thorough the study in standard cages with a 12 h light/dark cycle and ad libitum access to food and water. Following a 7-day acclimatisation period with routine health checks, mice were orthotopically xenografted with tumour cells and closely monitored - twice daily initially, then daily - before imaging and treatment commenced. Throughout the treatment phase, animals were scored three times weekly across multiple health parameters (*e.g.*, body weight, body posture, social behaviour, skin condition, grimace scoring, respiration, hydration, eyes, ears and nose, diarrhea, hydrocephalus, surgical wound gaping, infection), with humane euthanasia performed for severe or tumour-related symptoms. Analgesia (ibuprofen) was provided pre- and post-surgery, and weekly

imaging under isoflurane was conducted, with non-recovering mice euthanised when tumour burden was high. To establish xenografts, mice were injected with $3 \times 10^5$ GBM6 cells in the cortex, and tumour imaging was done on Day 5. On Day 6, animals were randomly grouped into the following treatment groups: *i)* Vehicle, *ii)* single WJA88, *iii)* single LXR-623 and *iv)* WJA88 + LXR-623. WJA88 was formulated using 40% Captisol and administered intraperitoneally. LXR-623 was formulated using 10% DMSO, 15% PEG 300, 5% Tween 80, and 70% Milli Q water and administrated orally. Starting on Day 6 post-implantation, WJA88 and LXR-623 were administered for two weeks (5 days on, 2 days off), and no significant weight loss was observed. WJA88 was dosed at 50 mg/kg, and LXR-623 was dosed at 100 mg/kg BID (i.e., 2 x 50 mg/kg). Animals received ad-libitium food and water, with nutritional enrichment when required. Tumour growth was monitored using the bioluminescence imaging once a week during and after the treatment to evaluate the efficacy of the treatments. Toxicity was monitored twice weekly from all treatment groups, blood counts ($n = 3$ per group) from whole blood collected in tubes coated with heparin, and sample analysis was performed using BC 5000 hematology analyzer from Mindray. Animals were euthanised with Avertin (tribromoethanol, 50 mg/kg intraperitoneally). Post-treatment, 3 animals in the combination treatment group were euthanised for non-tumour related issues, and during necropsy, we did observe cardiomegaly or bilateral renal enlargement compared to control mice.

**Electroporation model of glioblastoma.** Time-mated CD-1 females (embryonic day 14.5) were purchased from Australian BioResources (Sydney, NSW, Australia) and group housed in standard cages with a 12 h light/dark cycle and ad libitum access to food and water. Both male and female mice pups were utilised for brain electroporation. The hGFAPmin-SpCas9-T2A-PBase and PB_U6-Nf1, Pten, Trp53-EF1a-tdTomato (NPP-tdTomato) plasmid (kindly provided by Prof. Simona Parrinello, University College London, UK)[45,46] as well as pPB-mProm1-Venus-p27k- plasmids (kindly provided by Associate Prof. Luca Tiberi, Center for Integrative Biology Trento, Italy)[64] were used for the generation of a high-grade glioma tumours. All plasmids for in vivo injections were transformed into NEB® Stable Competent E. coli (New England Biolabs, Cat# C3040I) according to the recommended protocol with 100 ng of plasmid DNA. Transformed bacteria were grown at 30 °C for 24 h. Extraction and purification were performed using and following the protocol of the EndoFree Plasmid Maxi Kit (Qiagen, Cat# 12362). Final plasmid DNA was eluted in MilliQ water and DNA

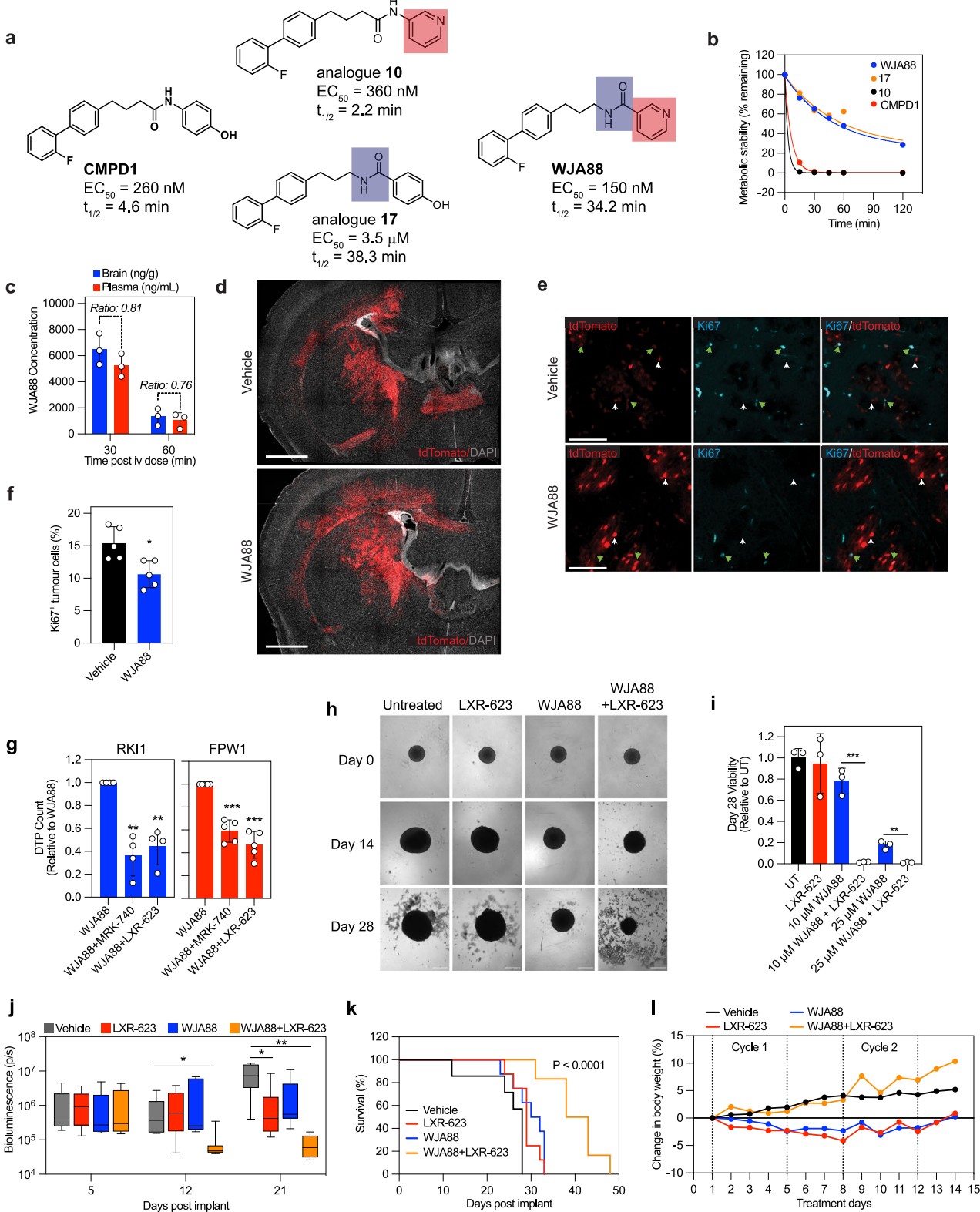

concentration was measured via Nanodrop (ND-ONE, Thermo Fisher Scientific). Plasmids were run on a gel to confirm correct plasmid size. Further plasmid validation was performed via Sanger sequencing, confirming correct DNA sequences. In vitro transfection of HEK293T cells (obtained from the American Type Culture Collection) was performed to confirm correct fluorescent protein expression (NPP-tdTomato) or successful integration capacity (PBase) through

multiple cell passages, following the protocol for Lipofectamine 2000 (ThermoFisher Scientific, Cat# 11668019). To generate the somatic-cell electroporation model (NPP-tdTomato model)[45,46], DNA-mixes were prepared containing; PiggyBase and piggyBacs at a 1:1:1 ratio (total DNA 2.44 mg/ml), saline and Fast Green FCF (0.5 µg/µl, Sigma-Aldrich Cat# F7258). CD-1 mice pups (postnatal day 2) were anaesthetised on ice for 5-10 min. Using a microinjector (µPUMP, PV850 WPI),

**Fig. 8 | Efficacy of brain-permeable microtubule-targeting agent and LXR agonist in glioblastoma xenografts. a, b** Structures, efficacy (EC$_{50}$; A172 cell viability) and metabolic stability (t$_{1/2}$; human liver microsomes) of CMPD1, analogues 10 and 17, and WJA88. **c** Plasma and brain concentrations of WJA88 (50 mg/kg, i.v., $n = 3$ mice). **d** Images of NPP-tdTomato tumour sections following vehicle or WJA88 treatment ($n = 5$ per treatment). Tumour cells are indicated by expression of tdTomato (red), with sections stained for DAPI (grey). Representative images of 3 sections per brain are shown. Scale bar = 1000 μm. **e** Ki67 immunostaining of WJA88 and vehicle-treated NPP-tdTomato tumours. Tumour cells (red) are marked for proliferation with Ki67 (cyan). Green arrows indicate proliferative (Ki67$^+$/tdTomato$^+$) and white arrows indicate non-proliferative (Ki67$^-$/tdTomato$^+$) tumour cells. Representative images of 3 sections per brain are shown. Scale bar = 100 μm. **f** Quantification of proliferative (Ki67$^+$/tdTomato$^+$) tumour cells. Data are mean ± SD ($n = 5$ per treatment). Unpaired t-test with Welch's correction: * indicates p (two tailed) = 0.0117. **g** Quantification of RKI1 and FPW1 drug-tolerant persister (DTP) cells surviving WJA88 (25 μM, 14 days) ± MRK-740 (3 μM) or LXR-623 (1 μM). Data are mean ± SD ($n = 3$ biological replicates). One sample t-test between co-treatment vs WJA88: ** indicates p (two tailed) = 0.0054, p (two tailed) = 0.0061 for WJA88 +

MRK-740 and WJA88 + LXR-623, respectively in RKI1 cells, *** indicates p (two tailed) = 0.0007, p (two tailed) = 0.0005 for WJA88 + MRK-740 and WJA88 + LXR-623, respectively in FPW1 cells. Representative images are provided in the Supplementary Fig. 8e. **h, i** Images and quantification of GBM6 spheroids viability when treated with WJA88 (10 or 25 μM) and LXR-623 (1 μM). Data are mean ± SD of 3 spheroids per treatment. Unpaired t-test between treatments: ** indicates p (two tailed) = 0.0019, *** indicates p (two tailed) = 0.0004. Scale bar = 100 nm. **j** Bioluminescence of GBM6 orthotopic tumours in mice treated with WJA88 (50 mg/kg) ± LXR-623 (100 mg/kg). Boxplots show mean (middle line) ± SD, whiskers represent minimum to maximum data points ($n = 8$ per treatment). Unpaired t-test between treatments: * indicates p (two tailed) = 0.0466, p (two tailed) = 0.029 for 12 and 21 days, respectively; ** indicates p (two tailed) = 0.0054. **k** Kaplan-Meier regression showing survival (%) in mice with GBM6 orthotopic tumours and treated with WJA88 (50 mg/kg) ± LXR-623 (100 mg/kg). Significance was determined with log rank (Mantel-Cox) test between the survival curves for vehicle vs treatment ($n = 8$ per treatment). **l** Changes in body weight of mice with GBM6 orthotopic tumours treated as in (**j**). Source data are provided as a Source Data file.

approximately 1 μl DNA-mix was injected into the right lateral ventricle (lambda −1,5 D/V, + 0.8 M/L) with the following settings; injection pressure: 1.0 PSI, compensation pressure: 0.2 PSI, duration: manual. PBS-wetted tweezertrodes (7 mm, Fisher Biotec, Cat# 45-0488) were placed in front of the ears on each side of the head, with the positive electrode on the injected side of the pup's head. Electroporation settings were the following: 100 V square wave, 50 ms pulses, 800 ms intervals, 5 times[46]. According to individual cohorts, the tail of each pup was tattooed (green/black dot) using 30 G needles. For full recovery from anaesthesia, pups were warmed by hand, checked for breathing reflex and placed on a heating mat. Pups were rubbed with cage litter before being returned to the mother to reduce cannibalisation incidences. After weaning, mice received individual ear notches and were observed daily for 7 days, then monitored three times per week, with increased monitoring in response to clinical signs. Animals are assessed using defined scoring criteria, and those reaching independent humane endpoints are to be promptly euthanised. No mice reached these criteria in this study.

**WJA88 treatment.** Captisol (Cydex Pharmaceuticals) was dissolved in MilliQ at 40% (0.4 g/mL), rotated for dissolution (35 rpm, 30 min at RT) and then sterile filtered (0.22 μm PES filter), after which it was stored at 4 °C. WJA88 was dissolved in 40% Captisol to a stock concentration of 5 mg/ml and sonicated for 1 h at 37 °C protected from light, after which it was stored as single-use aliquots at −80 °C. At three weeks post-electroporation, mice were divided in treatment groups for WJA88 ($n = 5$) or its vehicle ($n = 5$) and treated daily in two five-day treatment cycles with a two-day break in between. WJA88 or 40% Captisol were administered to mice via intraperitoneal injection (1% body weight) with WJA88 administered at a final dose of 50 mg/kg. Mice were euthanised 24 h after the last dose via intraperitoneal injection of pentobarbital (162.5 mg/mL) and lidocaine (10 mg/mL), followed by transcardial perfusion with PBS and then 4% paraformaldehyde.

**Immunostaining.** Fixed mouse brains were cut coronally in 40 μm sections using a VT1000S Vibratome (Leica). For antigen retrieval of mouse brains, sections were placed in sodium citrate solution (10 mM, pH 6.0) on a heating block at 95 °C for 8 min. Sections were cooled, rinsed with PBS and blocked for 2 h at room temperature (RT) in 2% donkey serum with 0.2% Triton-X-100 in PBS (blocking buffer). All antibodies were prepared in blocking buffer. Sections were washed with PBS and stained with primary antibody Ki67 (mouse, 1:100, #550609, BD Pharmingen), overnight at 4 °C. After three times 20 min washes with PBS, sections were incubated with anti-mouse 647 secondary antibody (donkey, 1:500, #715-606-151, Jackson ImmunoResearch) for 3 h at RT. Following another 20 min PBS wash, sections

were stained with DAPI (1:5000, 62248, Thermo Fisher Scientific) in PBS for 20 min at RT, before a final PBS wash. Sections were mounted onto Superfrost slides (SF41296SP, Thermo Fisher Scientific) using fluorescence mounting medium (S3023, Dako).

**Microscopy.** For whole-tumour imaging of tdTomato signal and DAPI staining, brain sections were imaged with a Stellaris confocal microscope (Leica) using a 10x objective and a 0.75x zoom factor, taking z planes every 2.41 μm across a range of 33.72 μm, which were then used to generate a maximum projection image. For Ki67 immunostaining, brain sections were imaged with a spinning disk confocal microscope (ANDOR WD Revolution) using a 20x objective, taking 10 μm z planes with a 2 μm step size. One position of three different brain sections per brain were imaged, including mid-SVZ (bregma -0.745 mm), late SVZ (either bregma -0.245 mm) and either late SVZ (bregma -0.38 mm) or the early-mid hippocampus region (bregma -1.455 mm). Positions were based on highest tumour cell density, which included the corpus callosum and caudoputamen. One quarter of each image was analysed in one plane. Overall tdTomato$^+$ and tdTomato$^+$Ki67$^+$ cells were counted manually using ImageJ. The proportion of tdTomato$^+$Ki67$^+$ cells across three positions was calculated for each animal and the average proportion for each treatment group was calculated.

**Statistical analysis.** To assess change in weight of mice treated with WJA88 compared to vehicle, fold-change in weight was calculated relative to day 1 and a two-way mixed effects ANOVA with Holm-Sidak multiple comparisons was performed. To assess change in the proportion of Ki67$^+$ tumour cells between WJA88 and vehicle treated tumours an unpaired t-test with Welch's correction was performed.

**Persister generation and expansion**

FPW1 and RKI1 cells ($1.5 \times 10^4$ cells/cm$^2$) were treated with CMPD1 (25 μM) or tivantinib (25 μM) for 14 days. Every 3 days, the media containing CMPD1 or tivantinib was replaced. At Day 14, drug-tolerant persister (DTP) cells were collected for further analyses. Alternatively, DTP cells were allowed to recover in drug-free media (drug holiday) until they regained their morphology and expansion started when cells resembled their parental counterparts. Images were taken using Zeiss Axio Vert.A1 microscope using the Zeiss ZEN software.

**Persisters quantification**

Cells ($1.5 \times 10^5$ cells/well) were seeded on black imaging 24-well plates (Eppendorf) and treated with tested compounds for 14 days. To remove dead cells, media was replaced with fresh media containing tested compounds every 3 days. Untreated (Day 0) and treated (Day 14) cells were stained with Nuclear-ID red stain (Enzo Lifesciences) at

1:1,000 dilution in StemPro media. Cells were incubated with the stain for 30 min prior to washing with PBS. Alternatively, when DAPI stain was used, on Day 14 cells were fixed in 1% PFA/PBS for 10 min, washed three times with PBS, stained and permeabilised in 1 μg/mL DAPI and 0.1% Triton X-100 in PBS for 15 min. Cells were imaged using Zeiss Axio Scope.A1 and Zeiss ZEN software. ImageJ Fiji was used for quantification (nine images per well).

## Cell viability and GR metrics

Neural stem cells, astrocytes, treatment-naïve glioblastoma cells (all at $2 \times 10^3$ cells/well) or drug-tolerant persister cells ($8 \times 10^3$ cells/well) were treated with DMSO or tested compounds at 8-point dilution row for 5 days. CellTitre-Blue (Promega, Cat# G808B) was added (1:10) and incubated at 37 °C for 2–4 h. Fluorescence was measured with a Tecan M200 PRO+ microplate reader (Tecan) at Ex/Em 530/590 with readings collected on Tecan Magellan software. Data were normalised to DMSO-treated controls (set as 1). Per-division $GR_{50}$, $GR_{max}$, $h_{GR}$, and $GR_{AOC}$ metrics were calculated from cell viability data on Day 0 and Day 5 using the *GRcalculator* online tool[22]. Graphs were recreated using GR values from the *GRcalculator* and Prism v10.1.2 (GraphPad).

## Colony formation assay

Cells ($1 \times 10^4$ cells/well) were treated with the tested compounds for 7 days, then washed with PBS and allowed to recover in drug-free media for an additional 7–21 days (depending on the cell line). Once visible colonies were formed in CMPD1-only treatments, colonies were stained with 1% toluidine blue (> 4 h, 4 °C). Plates were imaged using Bio-Rad Gel Doc and colony area quantified using the ImageJ software.

## Spheroid growth assay

MMK1 and GBM6 cells ($10 \times 10^3$ cells/well) were seeded into 96-well round bottom low attachment plates and centrifuged at 190 g for 3 min. Cells were left undisturbed for 48 h to form spheroids, then treated with tested compounds or DMSO (vehicle) for 28–35 days, with drug-containing media replenished every 4 days. Spheroids were imaged on an AxioVert A1 microscope immediately prior to treatment and every 7 days following the initiation of treatment to determine spheroid diameter using ImageJ. After 28 days of treatment, a PrestoBlue (Thermofisher Scientific, Cat # A13261) assay was performed according to the manufacturer's instructions with an incubation period of 3 h. Fluorescence was measured with a FLUOstar Omega microplate reader (BMG Labtech), and readings were normalised to the average of DMSO-treated vehicle controls. Graphing and statistical analyses were performed with GraphPad Prism v10.1.2.

## Incucyte Live Cell Imaging

Cells ($4 \times 10^4$ cells/well) were treated as described and placed into the Sartorius Incucyte SX5 at 37 °C and 5% $CO_2$. 16 images were taken per well every 2 h. Confluency was calculated using the Sartorius Incucyte 2022 A software.

## RNA sequencing

RNA extraction was performed with RNeasy Minikit (Qiagen, Cat# 74104), according to manufacturer's instructions, with additional on column DNA digestion using RNase free DNase set (Qiagen, Cat# 79254). For the sequencing of DTPs (Fig. 1), sequencing libraries were generated using NEBNext® UltraTM RNA Library Prep Kit for Illumina (New England Biolabs, Cat# E770) following manufacturer's instructions. For sequencing of RKI1 cells treated with CMPD1 ± MRK-740 or RKI1 cells transduced with NTC/PRDM9 sgRNA (Fig. 5), RNA extracted from samples underwent library preparation using mRNA stranded library preparation kit (Illumina, Cat# 20040532) with associated indexes and anchors, following manufacturer's instructions. Libraries were quantified using NEBnext Ultra II DNA library preparation kit for Illumina (New England Biolabs, Cat# E7630L). For sequencing in Fig. 1,

clustering of the index-coded samples was performed on a cBot Cluster Generation System using PE Cluster Kit cBot-HS (Illumina, discontinued) according to the manufacturer's instructions. After cluster generation, the library preparations were sequenced on an Illumina Novaseq. For Fig. 5, the libraries were sequenced on Illumina Nextseq2000 instrument and Illumina Nextseq1000/2000 Control Software Suite v1.7.1. FASTQ generation, alignments and counts were processed on instrument using Illumina Dragen RNA (v4.2.7) pipeline. Heatmaps were generated using Pheatmap package, gene sets were mapped to gene ontology terms using clusterProfiler package and plots were generated using ggplot2 package, all in RStudio. Gene annotation packages used in RStudio included biomaRt, org.Hs.eg.db, EnsDbHsapiens.v86 and AnnotationDbi. Venn diagrams were generated with Venny software. For data in (Fig. 5c). batch correction was performed using RUV and a set of stably expressed genes using Combat package in RStudio[65]. Samples were processed in three biological replicates with significance determined through adjusted *P*-value.

## LC-MS/MS analysis of histone peptides

All solvents, acids and bases used were HPLC-grade (Sigma-Aldrich) and all reactions were carried out in Protein-LoBind Eppendorf tubes. Chromatin-bound fractions from snap-frozen cell pellets were extracted using the Sub-Cellular Fractionation Kit (ThermoFisher Scientific, Cat # 78840) as per manufacturer's instructions. The samples were supplemented with 50 mM sodium bicarbonate, Protease Inhibitor Cocktail, 1 mM PMSF, 1 mM sodium orthovanadate and 10 mM sodium butyrate (All Sigma-Aldrich). Protein concentration was measured using Pierce BCA Protein Assay (ThermoFisher Scientific, Cat # 23225). Proteins were precipitated with methanol:chloroform, air-dried, reconstituted to ~ 2.5 mg/mL in 50 mM sodium bicarbonate (pH 8.0) and sonicated (SonoPlus Mini20 (Bandelin)). Histones were propionylated and trypsinised for bottom-up analysis as described by Sidoli et al. Samples were then acidified with 1% trifluoroacetic acid and desalted using Oasis HLB Sep-Pak columns (Waters) and vacuum dried.

For LC/MS-MS, 1 μg of propionylated histones in loading buffer (3% acetonitrile, 0.1% TFA) was injected onto a 30 cm × 75 μm inner diameter column packed in-house with 1.9 μm C18AQ particles (Dr Maisch GmbH, HPLC) using a Dionex Ultimate 3000 nanoflow UHPLC. Peptides were separated using a linear gradient of 5–35% buffer B over 120 min at 300 nL/min at 55 °C (buffer A: 0.1% (v/v) formic acid; buffer B: 80% (v/v) acetonitrile and 0.1% (v/v) formic acid). All MS analyses were performed using a Q-Exactive HFX mass spectrometer. For Data-Dependent Acquisition (DDA): after each full-scan MS1 (R = 120,000 at 200 *m/z*, 300–1600 *m/z*; $3 \times 10^6$ AGC; 110 ms max injection time), up to 10 most abundant precursor ions were selected for MS/MS (R = 45,000 at 200 *m/z*; $2 \times 10^5$ AGC; 86 ms max injection time; 30 normalised collision energy; peptide match preferred; exclude isotopes; 1.3 *m/z* isolation window; minimum charge state of +2; dynamic exclusion of 15 s). For DIA: after each full-scan MS1 (R = 60,000 at 200 *m/z* (300–1600 *m/z*; $3 \times 10^6$ AGC; 100 ms max injection time), 54 × 10 *m/z* isolations windows (loop count = 27) in the 390–930 *m/z* range were sequentially isolated and subjected to MS/MS (R = 15,000 at 200 *m/z*; $5 \times 10^5$ AGC; 22 ms max injection time; 30 normalised collision energy). 10 m/z isolation window placements were optimised in Skyline[66] to result in an inclusion list starting at 395.4296 *m/z* with increments of 10.00455 *m/z*. This resulted in a duty cycle of ~2.2 s.

Database searches were performed using Mascot v2.4 against the human SwissProt database (May 2019; 559,634 entries) using a precursor-ion and product-ion mass tolerance of ± 10 ppm and ± 0.02 Da, respectively. The enzyme was specified as ArgC with a maximum of 1 missed cleavage. Variable modifications were set as: acetyl(K), propionyl(K), monomethyl + propionyl(K), dimethyl(K) and trimethyl(K), propionyl(N-term), oxidation(M), carbamidomethyl(C). All DIA data were processed using Skyline (v20.1)[66]. Reference spectral

libraries were built in Skyline using Mascot.dat files using the BiblioSpec algorithm[67]. A False Discovery Rate (FDR) of 5% was set and a reverse decoy database was generated using Skyline[68].

## Data normalisation and statistical analysis

Processed MS1 quantifications were first Log transformed (base 2) and then quantile-normalised across samples. Data from FPW1 and RKI1 were subsequently analysed independently. For data from each cell type, Combat R package[69] was used to remove experimental batch effects[69]. Batch corrected data were used for downstream analyses including hierarchical clustering, heatmap visualisation and bar graphs.

## ChIP sequencing

RKI1 cells treated with CMPD1 ± MRK-740 or RKI1 cells transduced with NTC/PRDM9 sgRNA ± CMPD1 were harvested, cell pellets were cross-linked with 0.75% PFA/PBS for 10 min, quenched in 1 M glycine, washed in PBS, and snap frozen in liquid nitrogen. Cells were lysed in ChIP lysis buffer (50 mM HEPES-KOH pH 7.5, 140 mM NaCl, 1 mM EDTA pH 8, 1% Triton X-100, 0.1% sodium deoxycholate, 0.1% SDS, Protease Inhibitor Cocktail) and sonicated to shear chromatin to -300 bp DNA fragment size. DNA fragment size was checked via 1.5% agarose gel electrophoresis against a 100–1000 bp DNA. Input samples were aliquoted for later DNA extraction processing. Chromatin immunoprecipitation was carried out with H3K4me3 antibody (Cell Signalling Technology, Cat # 9751) added 1:50 to chromatin lysates. The samples were agitated overnight at 4 °C with bead capture of antibody bound chromatin fragments using Dynabeads Protein A (Thermofisher Scientific, Cat# 10002D). The samples were then washed in the following buffers: 1) low salt wash buffer (0.1% SDS, 1% Triton X-100, 2 mM EDTA, 20 mM Tris-HCl pH 8.0, 150 mM NaCl); 2) high salt wash buffer (0.1% SDS, 1% Triton X-100, 2 mM EDTA, 20 mM Tris-HCl pH 8.0, 500 mM NaCl); 3) LiCl wash buffer (0.25 M LiCl, 1% NP-40, 1% sodium deoxycholate, 1 mM EDTA, 10 mM Tris-HCl pH 8.0). DNA protein complexes were eluted from antibody captured beads in 1% SDS, 100 mM NaHCO₃. Chromatin fragments were subject to overnight de-cross linking in 0.32 M NaCl and RNA digestion with 10 μg RNAse A at 65 °C. Protein was digested using 20 μg Proteinase K for 1 h at 60 °C. DNA was extracted using ChIP Clean and Concentrate Kit (Zymo Research, Cat# D5205). Input and ChIP fragments were quantified using QUBIT dsDNA kit (Thermofisher Scientific, Cat# Q33230) and 5 ng DNA fragments was subjected to library preparation using NEBNext Ultra II DNA library preparation kit for Illumina (New England Biolabs, Cat# E7645S). Libraries were quantified, sequenced on Illumina Nextseq2000 instrument and Illumina Nextseq1000/2000 Control Software Suite v1.7.1. Illumina Dragen BCL Convert (v4.2.7) was used to generate FASTQ sequences for further downstream analysis using Deeptools 2.0 pipeline. Briefly, FASTQ files were aligned to hg38 homo sapiens reference genome, then bigWig density files were generated from the BAM alignments for subsequent downstream analyses. pyGenomeTracks was used for generating individual gene tracks of H3K4me3 enrichments. To identify PRDM9 motif occurrences across the genome, we obtained the reference genome sequence (hg38 for RKI1, hg19 for patient data GSE121723) in FASTA format, and homo sapiens PRDM9 motif matrix MA1723.2 was scored against it using FIMO from MEME Suite v5.5.7[70]. MACS2 callpeak was used to calculate H3K4me3 ChIPseq peak widths. Downstream analysis was conducted in RStudio using dpylr, ChIPseeker, GenomicRanges, EnsDb.Hsapiens.v86, AnnotationDbi and biomaRt packages. ChIP-seq experiments were analysed in biological duplicate ($n = 2$) according to ENCODE guidelines.

## LC-MS/MS analysis of lipids and cholesterol

RKI1 parent and CMPD1-derived drug-tolerant persister cells (Fig. 1e) were collected, homogenized in sample extraction buffer (50 mM Hepes pH 7.4, 25 mM KCl, Protease Inhibitor Cocktail) by sonicating for 5 min (30 s on/30 s off) at 4 °C with a Qsonica Q800R2 sonicating bath. Protein concentration was determined with the BCA assay. Lipids were extracted from 200 μL lysate (-200 μg protein) using the methyl-tert-butyl ether (MTBE)/methanol/water protocol as previously described[71]. Cell homogenate was combined with 250 μL methanol containing 0.01% 3,5-di-tert-4-butylhydroxyltoluene (BHT), internal standards and 850 μL MTBE, sonicated in a 4 °C water bath for 30 min, and phase separation was induced by centrifuging at 2000 g for 5 min. The upper organic phase was transferred to a 5 mL glass tube, and the aqueous phase was re-extracted with 500 μL MTBE and 150 μL methanol. The organic phase from the second extraction was combined with the first, after which the extracts were dried in a Savant SC210 SpeedVac dessicator (ThermoFisher Scientific). Lipids were reconstituted in 400 μL of 80% (v/v) methanol:20% water containing 0.01% BHT, 1 mM ammonium formate, and 0.1% formic acid, and stored at -80 °C. Lipids were detected in multiple reaction monitoring mode on a TSQ Altis triple quadrupole mass spectrometer with a Vanquish HPLC (ThermoFisher Scientific) and Waters Acquity UPLC CSH 2.1×100 mm C18 column (1.7 μm particle size), as previously described[72]. Run time was 25 min with a flow rate of 0.28 ml/min, using mobile phases A (10 mM ammonium formate, 0.1% formic acid, 60% acetonitrile and 40% water) and B (10 mM ammonium formate, 0.1% formic acid, 10% acetonitrile and 90% isopropanol) with the following binary gradient: 0–3 min, 20% B; 3-5.5 min, ramp to 45% B; 5.5–8 min, ramp to 65% B; 8–13 min, ramp to 85% B; 13-14 min, ramp to 100% B; 14-20 min, hold at 100% B; 20–25 min, decrease to 20% B and hold to 25 min. Acylcarnitine (AcCa), ceramide, sphingomyelin (SM), phosphatidylcholine (PC), lysophosphatidylcholine (LPC), and lysophosphatidylethanolamine (LPE) species were identified as the $[M + H]^+$ precursor ion, with product ion m/z values of 85.0 for AcCa, 184.1 for PC, LPC and SM, 264.3 for ceramide and SM, and neutral loss of 141.0 for LPE. Diacylglycerol (DG) was detected as the $[M + NH_4]^+$ precursor ion, with product ions corresponding to neutral loss of 35.0 and $RCOOH + NH_3$. Cholesterol was detected as precursor m/z 369.4 and product ion m/z 161.1. Phosphatidylethanolamine (PE), phosphatidylinositol (PI), and phosphatidylserine (PS) species were detected as the $[M-H]^-$ precursor ion, with product ions corresponding to the fatty acyl anion. TraceFinder 4.1 (ThermoFisher) was used to integrate the peaks. The molar amount of each lipid was calculated with reference to its class-specific internal standard, then normalized to protein content.

For targeted cholesterol analysis (Fig. 7f, g), lipids were extracted as described above with the addition of 5 nmoles of d7-cholesterol internal standard, followed by reconstitution in 360 μL of 25% 1-butanol:55% Methanol:20% H₂O containing 0.01% BHT. 300 μL was transferred to fused-insert glass HPLC vials. Lipids were detected in multiple reaction monitoring mode on a TSQ Altis triple quadrupole mass spectrometer coupled to a Vanquish HPLC. Lipids were resolved on a Phenomenex Kinetex 3×100 mm (2.6 μm particle size) C8 column. Run time was 12.5 minutes, with a flow rate of 0.5 mL/min using the following binary gradient: 0-4.1 min 90% B; 4.1-4.2 100% B; 4.2-10.5 min 100% B; 10.5–10.6 min 90% B; 10.6-12.5 90% B, where mobile phase A was 1 mM ammonium formate and 0.1% formic acid in water, and mobile phase B was 1 mM ammonium formate and 0.1% formic acid in methanol. Cholesterol was detected as precursor m/z 369.4 and product ion m/z 161.1. Lipids were quantified in TraceFinder and expressed relative to the internal standard and protein amount per sample.

## GC/MS analysis of cholesterol

RKI1 cells treated with CMPD ± MRK-740 ± MRK-740-NC (Fig. 7e, Supplementary Fig. 6d, Supplementary Fig. 7b, c) were collected, cell pellets washed in PBS, lysed with 10 M NaOH containing protease inhibitors and proteins quantified by BCA assay following manufacturers' instruction. Heavy isotope standards (100 ng per sample) and methanol supplemented with 0.02% BHT (8:1 v/v) were added to each sample and samples were heated at 60 °C for 1 h in 1 M KOH. The

reaction was neutralised with 300 mM HCl, and sterols extracted in hexane. The samples were dried, derivatised (60 °C, 1 h) in acetonitrile:BSTFA (1:3 v/v) and toluene (4:1 v/v) was added. Derivatives were analysed by gas chromatography (Agilent 8890) and mass spectrometry (Agilent 5977 System). Briefly, derivatized samples (1 µl) were injected into a fused silica capillary column (60 m x 0.25 mm internal diameter) coated with cross-linked 50% phenylmethylsiloxane (film thickness 0.25 um; Restek Rxi-5ms) using a splitless injection for sterol intermediate analysis or a 30:1 split ratio for cholesterol analysis. The helium carrier gas flow rate was 1 mL/min. Selected ion monitoring was performed by using the electron-ionization (EI) mode at 70 eV. Quantification was performed by monitoring specific ions at standard retention times and compared using the peak area ratio of the sterol vs the internal standard, on the Agilent Masshunter software. Data was then normalised to protein quantifications. The following standards were used: Cholesterol-d7 (Cat# 791645), Zymosterol-d5 (Cat# 700072), Zymostenol-d7 (Cat# 700117), Desmosterol-d6 (Cat# 700040 P) (all Avanti Polar Lipids), Lathosterol-d4 (Cat# D-5546), 7-Alpha-Hydroxycholesterol-d7 (Cat# D-4064), 7-Beta-hydroxycholesterol-d7 (Cat# D-4123) (all CDN isotopes), 24-hydroxycholesterol-d7 (Cat#, D730), 27-hydroxycholesterol-d5 (Cat# D733) (all Medical Isotopes), Squalene-d6 (Cat# TRC-S683802), Lanosterol-d6 (Cat# TR-L174584), 7-dehydrocholesterol-d7 (Cat# TRC-D229457), 25-hydroxycholesterol-d6 (Cat# TRC-H918032) (all TRC).

## Western blotting

For histone immunoblots in Fig. 2, histones were extracted using the Histone Extraction Kit (Abcam, Cat # ab113476) as per manufacturer's instructions, and protein concentrations were determined. 1 µg of histone extracts were resolved (30 min, 200 V) on 12% Bolt Bis-Tris gels (Cat# NW00127BOX) and transferred onto nitrocellulose membranes (Cat# IB301002) using iBlot 2 protein transfer system (Cat# IB21001) at 15 V, 7 min (all Thermofisher Scientific). For whole cell lysates, cell pellets were lysed in RIPA buffer containing protease inhibitor PMSF, protease inhibitor cocktail and phosphatase inhibitors (sodium orthovanadate), sonicated to remove viscosity, then protein concentration was measured. Samples were prepared with Bolt LDS loading buffer (Cat# B0007) and Bolt sample reducing agent (Cat# B0009) (both Thermofisher Scientific). Samples were heated at 95 °C for 5 min. 20–30 µg of total protein were resolved (2 h, 95 V) on 4-12% Bolt Bis-Tris gels (Cat# NW04120BOX) and dry transferred onto PVDF membranes (Cat# IB24001) at 20 V, 7 min (all Thermofisher Scientific). Membranes were blocked with 5% skim milk powder in TBST, washed in TBST, and incubated with primary antibody in 5% BSA in TBST overnight at 4 °C. Membranes were washed in TBST and incubated with secondary antibody for 1 h at room temperature. Detection was performed with Immobilon Western HRP Substrate Luminol-Peroxidase kit (MerckMillipore, Cat # WBKLS0500) and the Chemi-Doc MP Imaging System (Bio-Rad). Densitometry quantification was done in ImageLab software (Bio-Rad). The following antibodies were purchased from Cell Signalling Technologies: GAPDH (1:2000, Cat# 97166), H3K4me1 (1:1000, Cat# 5326), H3K4me2 (1:1000, Cat# 9725), H3K4me3 (1:1000, Cat# 9751S), H3K27me1 (1:1000, Cat# 84932), H3K27me3 (1:1000, Cat# 9733), Histone H3 (1:2000, Cat# 4499), Cleaved Histone H3 (Thr22) (1:1000, Cat# 12576), DHCR24 (1:1000, Cat# 2033), HMGCS1 (1:1000, Cat# 36877S), SETD1B (1:1000, Cat# 44922S), Rabbit IgG HRP-linked (1:2000, Cat# 7074) and Mouse IgG HRP-linked (1:2000, Cat# 7076). The following antibodies were purchased from ABCAM: H3K4me3 (1:1000, Cat# ab12209), H3K9me1 (1:1000, Cat# ab9045), H3K9me2 (1:1000, Cat# ab1220), H3K9me3 (1:1000, Cat# ab8898), H3K27me2 (1:1000, Cat# ab24684), H3K36me1 (1:1000, Cat# ab9048), H3K36me2 (1:1000, Cat# ab9049), H3K36me3 (1:1000, Cat# ab9050), Histone H3.3 (1:1000, Cat# ab176840) and MVD (1:1000, Cat# ab129061). Histone H3.1/3.2 (1 µg/mL, Cat# ABE154)

antibody was purchased from MerckMillipore and antibody against PRDM9 (1:1000, Cat# MA5-51196) was purchased from ThermoFisher.

## RT-qPCR

After RNA extraction, cDNA was generated using Applied Biosystems High-Capacity cDNA Reverse Transcription kit (Thermofisher Scientific, Cat# 4368814) as per manufacturer's instructions. RT-qPCR was performed using Quantitect validated primers (Qiagen) or custom primers, with KAPA SYBR FAST Universal 2× qPCR Master Mix (Kapa Biosystems, (Cat# KK4602). RT-PCR was run on LightCycler 480 (Roche). The cycling condition were as follows: 10 min at 95 °C followed by 45 cycles, each consisting of 10 s at 95 °C and 30 s at 60 °C. Samples were run in triplicate. Threshold cycles ($C_T$) were calculated using the LightCycler® 480 software. Relative quantification using the comparative $C_T$ method was used to analyse to the data output. GAPDH were used as loading controls. Values were expressed as fold change over corresponding values for the control by the $2\text{-}\Delta\Delta C_T$ method. The following primers were used, all from Qiagen Quantitect Primer Assay (Cat# 249900): Hs_ACTB_1_SG (Identifier: QT00095431), Hs_GAPDH_1_SG (Identifier: QT00079247), Hs_CDKN1A_SG (Identifier: QT00062090), Hs_PML_1_SG (Identifier: QT00090447), Hs_Y-PEL3_1_SG (Identifier: QT00078589), Hs_BHLHE41_1_SG (Identifier: QT00032697), Hs_CDKN1B_2_SG (Identifier: QT00998445), Hs_NR2F1_1_SG (Identifier: QT00089355), Hs_SOX2_1_SG (Identifier: QT00237601), Hs_SOX9_1_SG (Identifier: QT00001498), Hs_ORC1_1_SG (Identifier: QT00005341), Hs_NEK2_2_SG (Identifier: QT01668394), Hs_MCM5_1_SG (Identifier: QT00084000), Hs_KMT2A_2_SG (Identifier: QT00247464), Hs_KMT2B_1_SG (Identifier: QT00015778), Hs_KMT2C_1_SG (Identifier: QT00029316), Hs_KMT2D_2_SG (Identifier: QT01762096), Hs_KMT2E_1_SG (Identifier: QT00053900), Hs_KIAA0339_1_SG (KMT2F/SETD1A, Identifier: QT00042427), Hs_KIAA1076_1_SG (KMT2G/SETD1B; Identifier: QT01016799), Hs_ASH1L_1_SG (KMT2H, Identifier: QT00083335), Hs_BUB1_1_SG (Identifier: QT00082929), Hs_ABCA1_SG (Identifier: QT00074606). In Fig. 4a, Hs_PRDM9_1_SG (KMT8B, Identifier: QT01023631) was used. In Fig. 4b, c, e the PRDM9 custom designed primers (Forward: TGAAAGAATTGTCAAGAACAGCA; Reverse: CTCCTTCTTCCTGAGTTCCAGT) were used (manufactured by Integrated DNA Technologies). KDM5A custom designed primers (Forward: ACCCCAACGTGCTAATGGAG; Reverse: TCCAAT GGGCAACCAGTCAG) and CTSL1 custom designed primers (Forward: TCTGCTGGCCTTGAGGTTTT; Reverse: GGCAATTCCCAGGCAAAAGG) were manufactured by Integrated DNA Technologies.

## Retroviral packaging and CRISPR-Cas9 infection

SET1A, SET1B, PRDM9 and non-targeting control (NTC) sgRNAs were cloned by traditional restriction digestion into pLentiCRISPRv2 (Addgene plasmid 52961). HEK293T cells ($7.2 \times 10^5$ per well, 6-well plate, one for each sgRNA) were seeded and once cells reached 70–90% confluency, cells were transfected using Lipofectamine 3000 (Thermofisher Scientific, Cat# L3000015) with packaging the plasmids pCAG-VSVG (Addgene plasmid 35616) and psPAX2 (Addgene plasmid 12260) and the respective sgRNA pLentiCRISPRv2 vector at a 1:3:3 ratio. 16 h after transfection the medium was replenished with fresh medium. At 48 h post transfection the lentivirus-containing supernatant was collected and filtered through a 0.45 µm ultra-low protein binding filter and concentrated using PEG according to manufacturer's instructions. The concentrated lentivirus was stored at -80 °C. RKI1 cells ($7 \times 10^4$ cells/well) were incubated with polybrene-containing (8 µg/mL) media and lentiviral sgRNA particles for 16 hours, followed by resting the cells in fresh media for 24 h. Cells were challenged in puromycin-containing media (1 µg/mL) for 72 h, or until all untransduced cells were eliminated. Puromycin-containing media was replaced with fresh media and knock-out/control cell lines were expanded. To validate gene knockout, genomic DNA was extracted

from cell pellets with the ISOLATE II Genomic DNA Kit (Bioline, Cat# BIO-52067), subjected to PCR amplification and Sanger sequencing of each sgRNA target site and comparison to untransduced RKI1 cells through Synthego ICE analysis tool (Supplementary Fig. 9)[73]. sgRNA sequences and primers against gDNA are listed below.

| sgRNA | Sequence | Manufacturer |
|---|---|---|
| Non-Targeted Control (NTC) | GCATTCTTCAAACTATCCCA | Integrated DNA Technologies |
| sgSETD1B (1) | TGGGCAGCATTTCAGCCTGG | Integrated DNA Technologies |
| sgSETD1B (2) | CATGGGCAACATTATCCACG | Integrated DNA Technologies |
| sgPRDM9 (1) | TCCACTCTTAAGGCCATCCA | Integrated DNA Technologies |
| sgPRDM9 (2) | CAACAATGGATACTCCTGGC | Integrated DNA Technologies |
| **Primers** | **Sequence** | **Manufacturer** |
| SETD1B (1) | Forward: TGCTGGTTTTCAGGTTGGGT Reverse: TTTTCGAACTCGGGGAGGTG | Integrated DNA Technologies |
| SETD1B (2) | Forward: CCTCCGAAGCAGGT-GACATT Reverse: TCCCCTAACACAGGCAACAC | Integrated DNA Technologies |
| PRDM9 (1) | Forward: ACTTGGAATCAT-CACTCTCTGAT Reverse: ACCTTCTGGTGTTTACGCTGT | Integrated DNA Technologies |
| PRDM9 (2) | Forward: AGGA-CAGTGCAGTGGACAAG Reverse: TTTGGGCCTGCATAGCTTCA | Integrated DNA Technologies |

**Flow Cytometry for reactive oxygen species, lipid peroxidation and cell cycle**

Flow cytometric analyses of reactive oxygen species using CellROX™ Green Reagent (ThermoFisher Scientific, Cat# C10444), lipid peroxidation using Image-iT Lipid Peroxidation kit (ThermoFisher Scientific, Cat# C10445) were carried out following manufacturer's instructions. Briefly, following treatments, cells were stained for 30 min, harvested, washed in PBS and resuspended in media in flow cytometry tubes. For ROS quantification, staining intensity was assessed in the FITC channel (495/525) using a BD LSR Fortessa X-20. Lipid peroxidation was assessed in the FITC channel (495/525) for oxidised lipid species and Texas Red channels (586/603) for total lipid staining using a BD LSR Fortessa X-20. Cell cycle distribution was determined using propidium iodide (Sigma-Aldrich, Cat# 81845) staining. Briefly, cells were harvested, fixed overnight at -20 °C in 70% ethanol, washed in PBS, incubated in RNAse A (1 µg /mL; 30 min, 37 °C) and stained in Propidium Iodide (1 µg/mL in PBS, 20 min, room temperature, protected from light), and resuspended in flow buffer (2% FBS in PBS) for fluorescence measurements of DNA content in Texas Red channel (586/603) using a

BD LSR Fortessa X-20 instrument and BD FACSDiv software. Data were analysed using FlowJo v10.8.1.

**Tubulin polymerization assay**

Tubulin polymerization assay was done using an assay kit (Cytoskeleton, Cat# BK006P) per manufacturer's instructions. Briefly, porcine brain tubulin (10 mg/ml) was thawed in water before placing on ice and used within 2 h. Tested compounds were incubated at 37 °C with reaction mixture containing tubulin, in 55 µL reaction volume, and fluorescence over time was measured using TECAN Infite® 1000 PRO plate reader (Männedorf, Switzerland) with Tecan Magellan software.

**[³H]Colchicine binding assay**

A radioligand competition assay was performed using the centrifugal gel filtration method. The reaction mixture (12 µL) contained 2 µM purified tubulin and 0.1 µCi (1 µl of 0.1 mCi/ml, specific activity 80%) [³H]colchicine in PEM buffer (100 mM PIPES, 10 mM $CaCl_2$, 1 mM Na-EGTA, and 1 mM $MgCl_2$ [pH 6.9]). Competition binding reaction was started by the addition of 2 µL of tested compounds dissolved in $dH_2O$ (final concentration of 10 µM). Reaction mixtures were incubated at room temperature for 45 min, unlabelled radioactive material was filtered by using 75 µL spin column Zeba Micro Desalt Spin Column (Pierce Biotechnology) following manufacturer's guidelines. Elute was added to the scintillation fluid to measure radioactivity of [³H]-colchicine bound tubulin using a scintillation counter.

**In vitro metabolic stability**

All reactions were performed in 200 µL reaction volume in duplicate. 10 µM of tested compounds were incubated with human liver microsomes (0.4 mg/ml) in potassium phosphate buffer (0.1 M, pH 7.4) at 37 °C with gentle shaking for 5 min. Assay was initiated by adding 12 µL of NADPH regenerating system (containing final concentrations of 1 mM NADP, 3 mM glucose-6-phosphate, 3.3 mM $MgCl_2$ and 0.4 u/ml glucose-6-phosphate dehydrogenase). Reaction was quenched by adding 130 µL of ice-cold methanol, vortexed vigorously and centrifuged at $15,000\,g$ at 4 °C for 10 min. 5 µL of the supernatant was analysed using an Agilent 1260 LC system coupled to a QTRAP 6500 mass spectrometer. For LC, zorbax Extend-C18 (2.1 × 50 mm 1.8 um) column was used in reversed-phase mode at flow rate of 200 µL/min, with gradient elution starting with 10% of phase B (0.1% formic acid in water) and 90% of phase A (0.1% formic acid in acetonitrile). The amount of phase B was linearly increased from 10% to 90% in 5 min followed by 2 min at 90 % B then back to initial conditions at 8 min. The MS detector was operated with an ESI positive ionization mode. Source temperature and capillary voltage were set at 300 °C and 4000 V, respectively. The Analyst software was used to control the instruments and data acquisition. Assaying of test compounds was carried out utilizing the mode of multiple reaction monitoring (MRM) using the following conditions for each compound. Ion transitions were 350.3 to 241 for CMPD1, 350.3 to 230 for analogue **17**, 335 to 106 for WJA88 and 335 to 185 for analogue **10**. Fragmentor voltage was set to 145 V with collision energy of 30 for all the compounds.

**Brain uptake and pharmacokinetics of WJA88**

Brain uptake of WJA88 was assessed in male CD-1 mice ($n = 3$ per timepoint) following administration of a single bolus dose of 50 mg/kg by the intravenous route. Terminal blood and brain samples were collected from sub-groups of mice at 30 min and 60 min post-dose, the concentration of WJA88 in each sample was determined using LC-MS/MS. Pharmacokinetic parameters were assessed in male CD-1 mice ($n = 3$) following administration of a single 50 mg/kg dose intraperitoneally. Blood samples were collected at seven time-points up to 24 h, transferred into tubes containing $K_2$-EDTA and WJA88 concentration in each sample was determined with LC-MS/MS.

Pharmacokinetic parameters were calculated using Phoenix WinNonlin 6.3 software. All experiments and calculations were performed by WUXI, Study No. 410466-20191217-MPK.

### Data analysis

Statistical analyses were performed using Prism v10.1.2 (GraphPad) or R Studio for sequencing data. Generally, an unpaired *t*-test was used when comparing between different treatments and one sample *t*-test was used when comparing between a sample and a control that the sample was normalised to. For large data, without normal distribution, Wilcoxon rank-sum test was used. Statistical comparison of survival rates was carried out using the Mantel-Cox log-rank test. Correlation analyses were carried out using Pearson's product moment correlation coefficient. The adjusted *P*-value was taken for determining significance of gene expression changes for RNA sequencing experiments. For ChIP sequencing experiments, *q*-value of 0.01 was taken for determination of high-confidence peaks. For histone peptide LC-MS/MS and lipidomic experiments, normalised peak intensities were tested for significance, between experimental treatments, using an unpaired *t*-test or one-sample *t*-test. Throughout all figures: * indicates $p < 0.05$; ** indicates $p < 0.01$; *** indicates $p < 0.001$ and **** indicates $p < 0.0001$, exact P values are provided in figure legends.

### Chemistry general experimental methods

Unless otherwise stated, reactions were conducted under positive pressure of a dry nitrogen or argon atmosphere. Temperatures of 0 °C and -10 °C were obtained using a water/ice bath or salt/ice bath respectively. Reaction mixture temperatures were reported according to the oil bath/cooling bath temperature unless otherwise stated. Anhydrous dichloromethane, triethylamine and diisopropyl ethylamine was obtained by distillation over calcium hydride. Anhydrous DMF, methanol, THF, and acetonitrile were obtained from a PureSolv MD 7 solvent purific ation system (Innovative Technology, Inc.). Unless noted otherwise, commercially obtained reagents were used as purchased without further purification. Analytical thin-layer chromatography (TLC) was performed using Merck aluminium backed silica gel 60 F254 (0.2 mm) plates which were visualised with shortwave (254 nm) and/or longwave (365 nm) ultraviolet (UV) light, potassium permanganate, vanillin, *p*-anisaldehyde, ninhydrin or cerium molybdate ("Goofy's Dip") stains. Flash chromatography was performed using Grace Davisil silica gel, pore size 60 Å, 230–400 mesh particle size. Solvents for flash chromatography were distilled prior to use, or used as purchased for HPLC grade, with the eluent mixture reported as the volume/volume ratio (v/v).

Melting points were measured with open capillaries using a Stanford Research Systems (SRS) MPA160 melting point apparatus with a ramp rate of 0.5–2.0 °C/min and are uncorrected. Infrared absorption spectra were recorded on a Bruker ALPHA FT-IR spectrometer, and the data are reported as vibrational frequency (cm⁻¹). Nuclear magnetic resonance spectra were recorded at 300 K unless stated otherwise, using either a Bruker AVANCE DRX200 (200 MHz), DRX300 (300 MHz), DRX400 (400.1 MHz), or AVANCE III 500 Ascend (500.1 MHz) spectrometer. The data is reported as the chemical shift (δ ppm) relative to the solvent residual peak, relative integral, multiplicity (s = singlet, d = doublet, t = triplet, q = quartet, p = pentet, m = multiplet, br = broad, dd = doublet of doublets), coupling constant (*J* Hz). Spectra for some compounds were observed as rotamers, in these cases the major peaks are reported. Low resolution mass spectra (LRMS) were recorded using electrospray ionisation (ESI) or atmospheric pressure chemical ionisation (APCI) recorded on a Finnigan LCQ ion trap spectrometer, or by electron impact gas chromatography mass spectrometry (GC/MS). High resolution mass spectra were run on a Bruker 7 T Apex Qe Fourier Transform Ion

Cyclotron resonance mass spectrometer equipped with an Apollo II ESI/APCI/MALDI Dual source by the Mass Spectrometry Facility of the School of Chemistry at The University of Sydney. Samples run by ESI were directly infused (150 μL/hr) using a Cole Palmer syringe pump. Samples run by APCI were injected (5 μL) into a flow of methanol (0.3 mL/min) by HPLC (Agilent 1100) coupled to the mass spectrometer. Elemental analysis was obtained from the Chemical Analysis Facility in the Department of Chemistry and Biomolecular Sciences, Macquarie University, Australia.

Analytical HPLC purity traces were taken on a Waters 2695 Separations module equipped with Waters 2996 Photodiode Array detector (set at 230, 254 and 271 nm). All samples were eluted through a Waters SunFire™ C18 5 μm column (2.1 × 150 mm) using a flow rate of 0.2 mL/min of Solvent A: MilliQ water (+0.1% trifluoroacetic acid or 0.1% formic acid) and Solvent B: acetonitrile (+0.1% trifluoroacetic acid or 0.1% formic acid). This method consisted of gradient elution (0-100% Solvent A:B over 30 min). Data acquisition and processing was performed with the Waters Empower 2 software. Reported data for all compounds are based on the 254 nm channel.

### General synthetic methods

**Suzuki cross coupling.** Tetrakis(triphenylphosphine)palladium(0) (0.05 mol%) was added to a stirred suspension of boronic acid (1.2 eq.), $Cs_2CO_3$ (3 eq.) and 4-bromophenylbutanoic acid (1 eq.) in a degassed THF/water (9:1 v/v) mixture. The resultant mixture stirred at reflux for 8 h. Upon completion the reaction mixture was diluted with HCl (1 M, 1 x solvent volume) and extracted with EtOAc (3 × 50 mL). The combined organic phases subsequently washed with brine, before being dried ($MgSO_4$), filtered and concentrated under reduced pressure. The resultant acid was purified by flash column chromatography (silica, 1:1 v/v EtOAc:Hex).

**PyBOP® amidation.** An ice-cold stirred solution of acid (1 eq.), aniline (1–1.2 eq.) and $^iPr_2Net$ (2 eq.) in DMF was treated with PyBOP® (1 eq.), allowed to warm to room temperature and stirring continued for 12 h. The reaction mixture was diluted with $CH_2Cl_2$ (10 x solvent volume) and water (10 x solvent volume), the separated aqueous phase was subsequently extracted with $CH_2Cl_2$ (2 x with 10 x solvent volume) and combined organics washed with water (3 x with 10 x solvent volume) sat. aq. Solution of $NaHCO_3$ (5 x solvent volume) and brine (100 mL) before being dried ($MgSO_4$), filtered and purified via flash column chromatography (silica, 1:1 v/v EtOAc:Hex) to give the required amides.

**4-(2'-fluoro-[1,1'-biphenyl]-4-yl)butanoic acid.**

According to the general procedure for Suzuki Cross couplings, the title compound was synthesised from 2-fluorophenylboronic acid (1.70 g, 12.3 mmol, 1.2 eq.), and 4-bromophenylbutanoic acid (2.50 g, 10.2 mmol, 1.0 eq.) in dry degassed THF/water (100 mL, 9:1 v/v mixture). The crude product was purified by flash column chromatography (silica, 1:1 v/v EtOAc:Hex) to give the title compound as a white crystalline solid (1.6 g, 61%).

$R_f$: 0.22 ($SiO_2$, 1:1 EtOAc:Hex), ¹H NMR (300 MHz, DMSO-$d_6$) δ 12.06 (br s, 1H), 7.57 – 7.23 (m, 8H), 2.64 (t, *J* = 7.4 Hz, 2H), 2.25 (t, *J* = 7.4 Hz, 2H), 1.88 – 1.80 (m, 2H). ¹³C NMR (75 MHz, DMSO-$d_6$) δ 174.2, 159.1 (d, $^1J_{CF}$ = 245.8 Hz), 141.3, 132.6 (2 C), 130.7 (d, $^4J_{CF}$ = 3.7 Hz), 129.3 (d, $^3J_{CF}$ = 8.5 Hz), 128.7 (d, $^4J_{CF}$ = 3.0 Hz), 128.6 (2 C), 128.2 (d, $^3J_{CF}$ = 13.9 Hz), 124.9 (d, $^4J_{CF}$ = 3.7 Hz), 116.0 (d, $^2J_{CF}$ = 22.6 Hz), 34.1, 33.1, 26.2. ¹⁹F NMR (282 MHz, DMSO-$d_6$) δ -118.4. IR (diamond cell, neat)

$v_{max}$: 3026, 2902, 1696, 1480, 1435, 1250, 1202, 939, 799, 756, 563 cm$^{-1}$. LRMS (ESI-) m/z: 257 [(M-H)$^-$, 100%].

## CMPD1 (4-(2'-fluoro-[1,1'-biphenyl]-4-yl)-N-(4-hydroxyphenyl) butanamide).

According to the general procedure for Suzuki Cross couplings, the title compound was synthesised from 2-fluorophenylboronic acid (84.0 g, 0.6 mmol, 1.0 eq.), and 4-(4-bromophenyl)-N-(4-hydroxyphenyl)butanamide (200 mg, 0.6 mmol, 1.0 eq.) in dry degassed THF/water (4 mL, 9:1 v/v mixture). The crude product was purified by flash column chromatography (silica, 1:1 v/v EtOAc:Hex) to give the title compound as a white solid (150 mg, 71%).

$R_f$: 0.38 (SiO$_2$, 1:1 EtOAc:Hex), $^1$H NMR (300 MHz, Chloroform-$d$) δ 7.49 (dd, $J$ = 8.1, 1.7 Hz, 2H), 7.43 (td, J = 7.7, 1.9 Hz, 1H), 7.34 – 7.27 (m, 5H), 7.23 – 7.11 (m, 2H), 7.01 (s, 1H), 6.76 (d, $J$ = 8.9 Hz, 2H), 5.25 (s, 1H), 2.77 (t, $J$ = 7.4 Hz, 2H), 2.36 (t, $J$ = 7.4 Hz, 2H), 2.12 (p, $J$ = 7.4 Hz, 2H).$^{13}$C NMR (75 MHz, Chloroform-$d$) δ 171.2, 159.9 (d, $^1J_{CF}$ = 247.6 Hz), 152.9, 141.0, 133.8, 130.8 (d, $^4J_{CF}$ = 3.7 Hz, 2 C), 130.7, 129.2 (d, $^3J_{CF}$ = 3.1 Hz), 128.9 (d, $^3J_{CF}$ = 8.2 Hz), 128.8 (2 C), 124.5 (d, $^4J_{CF}$ = 3.8 Hz), 122.5 (2 C), 116.2 (d, $^2J_{CF}$ = 22.9 Hz), 115.9 (2 C), 36.7, 34.9, 26.9. $^{19}$F NMR (282 MHz, Chloroform-$d$) δ -118.1. IR (diamond cell, neat) $v_{max}$: 3317, 2965, 2816, 1654, 1514, 1350, 1230, 1100, 880 cm$^{-1}$. LRMS (ESI + ) m/z: 372 [(M+Na)$^+$, 100%]. HRMS (ESI + ) calcd for C$_{22}$H$_{20}$FNNaO$_2$ (M+Na)$^+$, 372.13703; Found 372.13687.

## 4-(2'-fluoro-[1,1'-biphenyl]-4-yl)-N-(2-hydroxyphenyl) butanamide (2).

According to the general procedure for PyBOP® amidation, the title compound was synthesised from 4-(2'-fluoro-[1,1'-biphenyl]-4-yl) butanoic acid (130 mg, 0.5 mmol, 1.0 eq.), and 2-hydroxyaniline (58.6 mg, 0.5 mmol, 1.0 eq.) in dry DMF (5 mL). The crude product was purified by flash column chromatography (silica, 1:1 v/v EtOAc:Hex) to give the title compound as a yellow oil (126 mg, 69%).

$R_f$: 0.51 (SiO$_2$, 1:1 EtOAc:Hex), m.p.: 53 – 55 °C, $^1$H NMR (500 MHz, DMSO-$d_6$) δ 9.72 (s, 1H), 9.29 (s, 1H), 7.69 – 7.66 (m, 1H), 7.53 – 7.37 (m, 4H), 7.34 – 7.17 (m, 4H), 6.96 – 6.93 (m, 1H), 6.88 – 6.86 (m, 1H), 6.78 – 6.75 (m, 1H), 2.70 – 2.58 (m, 2H), 2.47 – 2.38 (m, 2H), 1.98 – 1.84 (m, 2H). $^{13}$C NMR (126 MHz, DMSO-$d_6$) δ 171.6, 159.1 (d, $^1J_{CF}$ = 245.7 Hz), 148.0, 141.5, 132.6, 131.1, 130.6 (d, $^4J_{CF}$ = 2.7 Hz, 2 C), 129.2 (d, $^3J_{CF}$ = 8.2 Hz), 128.7 (d, $^4J_{CF}$ = 2.9 Hz), 128.6 (2 C), 128.2 (d, $^3J_{CF}$ = 13.2 Hz), 126.4, 124.9 (d, $^4J_{CF}$ = 3.5 Hz), 124.7, 122.5, 119.0, 116.1, 116.0 (d, $^4J_{CF}$ = 2.7 Hz), 35.4, 34.3, 26.9. $^{19}$F NMR (471 MHz, DMSO-$d_6$) δ -118.4. IR (diamond cell, neat) $v_{max}$: 3175, 1633, 1594, 1524, 1483, 1453, 1362, 1283, 1103, 821, 744, 458 cm$^{-1}$. LRMS (ESI + ) m/z: 372 [(M+Na)$^+$, 100%].

## 4-(2'-fluoro-[1,1'-biphenyl]-4-yl)-N-(2-methoxyphenyl) butanamide (3).

According to the general procedure for PyBOP® amidation, the title compound was synthesised from 4-(2'-fluoro-[1,1'-biphenyl]-4-yl) butanoic acid (130 mg, 0.5 mmol, 1.0 eq), and 2-methoxyaniline (61.6 mg, 0.5 mmol, 1.0 eq) in dry DMF (5 mL). The crude product was purified by flash column chromatography (silica, 1:1 v/v EtOAc:Hex) to give the title compound as a yellow oil (126 mg, 69%).

$R_f$: 0.41 (SiO$_2$, 1:1 EtOAc:Hex), $^1$H NMR (500 MHz, DMSO-$d_6$) δ 9.07 (s, 1H), 7.94 (t, $J$ = 9.1 Hz, 1H), 7.57 – 7.34 (m, 4H), 7.34 – 7.13 (m, 4H), 7.11 – 6.96 (m, 2H), 6.95 – 6.84 (m, 1H), 3.82 (s, 3H), 2.75 – 2.55 (m, 2H), 2.48 – 2.33 (m, 2H), 2.05 – 1.81 (m, 2H). $^{13}$C NMR (126 MHz, DMSO-$d_6$) δ 171.1, 159.1 (d, $^1J_{CF}$ = 245.6 Hz), 149.7, 141.5, 132.6, 130.6 (2 C), 129.2 (d, $^3J_{CF}$ = 8.3 Hz), 128.7 (d, $^4J_{CF}$ = 2.8 Hz), 128.6 (2 C), 128.2 (d, $^3J_{CF}$ = 13.2 Hz), 127.4, 124.9 (d, $^4J_{CF}$ = 3.5 Hz), 124.2, 122.2, 120.1, 116.0 (d, $^2J_{CF}$ = 22.7 Hz), 111.1, 55.6, 35.6, 34.3, 26.8. $^{19}$F NMR (471 MHz, DMSO-$d_6$) δ -118.4. IR (diamond cell, neat) $v_{max}$: 3417, 3311, 2934, 1676, 1599, 1518, 1482, 1457, 1432, 1288, 1249, 1216, 1115, 1026, 745 cm$^{-1}$. LRMS (ESI + ) m/z: 386 [(M+Na)$^+$, 100%].

## 4-(2'-fluoro-[1,1'-biphenyl]-4-yl)-N-(3-methoxyphenyl) butanamide (4).

According to the general procedure for PyBOP® amidation, the title compound was synthesised from 4-(2'-fluoro-[1,1'-biphenyl]-4-yl) butanoic acid (130 mg, 0.5 mmol, 1.0 eq.), and 3-methoxyaniline (61.6 mg, 0.5 mmol, 1.0 eq.) in dry DMF (5 mL). The crude product was purified by flash column chromatography (silica, 1:1 v/v EtOAc:Hex) to give the title compound as a colourless oil (151 mg, 83%).

$R_f$: 0.41 (SiO$_2$, 1:1 EtOAc:Hex), $^1$H NMR (500 MHz, DMSO-$d_6$) δ 9.88 (s, 1H), 7.56 – 7.24 (m, 8H), 7.24 – 7.07 (m, 3H), 6.72 – 6.49 (m, 1H), 3.71 (s, 3H), 2.73 – 2.56 (m, 2H), 2.43 – 2.24 (m, 2H), 2.00 – 1.83 (m, 2H). $^{13}$C NMR (126 MHz, DMSO-$d_6$) δ 171.0, 159.1 (d, $^1J_{CF}$ = 245.6 Hz), 141.4, 140.5, 132.6, 131.1, 130.6 (d, $^4J_{CF}$ = 2.7 Hz, 2 C), 129.4, 129.2 (d, $^3J_{CF}$ = 8.2 Hz), 128.7 (d, $^4J_{CF}$ = 2.9 Hz), 128.6 (2 C), 128.2 (d, $^3J_{CF}$ = 13.1 Hz), 124.9 (d, $^4J_{CF}$ = 3.4 Hz), 116.0 (d, $^2J_{CF}$ = 22.5 Hz), 111.4, 108.4, 104.9, 54.9, 35.8, 34.3, 26.5. $^{19}$F NMR (471 MHz, DMSO-$d_6$) δ -118.4. IR (diamond cell, neat) $v_{max}$: 1659, 1597, 1543, 1483, 1451, 1416, 1284, 1209, 1155, 1042, 755, 687, 565 cm$^{-1}$. LRMS (ESI + ) m/z: 386 [(M+Na)$^+$, 100%].

## 4-(2'-fluoro-[1,1'-biphenyl]-4-yl)-N-(4-methoxyphenyl) butanamide (5).

According to the general procedure for PyBOP® amidation, the title compound was synthesised from 4-(2'-fluoro-[1,1'-biphenyl]-4-yl) butanoic acid (130 mg, 0.5 mmol, 1.0 eq.), and 4-methoxyaniline

(61.6 mg, 0.5 mmol, 1.0 eq.) in dry DMF (5 mL). The crude product was purified by flash column chromatography (silica, 1:1 v/v EtOAc:Hex) to give the title compound as a pink powder (133 mg, 73%).

$R_f$: 0.40 (SiO$_2$, 1:1 EtOAc:Hex), m.p.: 103 – 106 °C, [1]H NMR (300 MHz, DMSO-$d_6$) δ [1]H NMR (500 MHz, DMSO-$d_6$) δ 9.75 – 9.72 (m, 1H), 7.53 – 7.45 (m, 5H), 7.40 – 7.16 (m, 5H), 6.86 (d, $J$ = 9.1 Hz, 2H), 3.71 (s, 3H), 2.69 – 2.58 (m, 2H), 2.34 – 2.26 (m, 2H), 1.97 – 1.84 (m, 2H). [13]C NMR (126 MHz, DMSO-$d_6$) δ 170.4, 159.1 (d, $^1J_{CF}$ = 245.6 Hz), 155.0, 141.5, 132.6 (d, $^3J_{CF}$ = 9.6 Hz), 131.1, 130.6 (2 C), 129.2 (d, $^3J_{CF}$ = 8.4 Hz), 128.7 (d, $^4J_{CF}$ = 2.9 Hz), 128.6 (2 C), 124.9 (d, $^4J_{CF}$ = 3.5 Hz), 120.6 (2 C), 116.1 (d, $^2J_{CF}$ = 22.7 Hz), 113.8 (2 C), 55.1, 35.7, 34.4, 26.7. [19]F NMR (471 MHz, DMSO-$d_6$) δ -118.4. IR (diamond cell, neat) ν$_{max}$: 3283, 2914, 1646, 1510, 1483, 1409, 1245, 1106, 1030, 827, 802, 753, 601, 568, 521 cm$^{-1}$. LRMS (ESI + ) m/z: 386 [(M+Na)$^+$, 100%].

## 4-(2′-fluoro-[1,1′-biphenyl]-4-yl)-N-(3,4,5-trihydroxyphenyl)butanamide (6).

BBr$_3$ (1.75 mL of a 1 M soln. in CH$_2$Cl$_2$, 1.75 mmol, 5.0 eq.) was added dropwise to an ice cold, stirring solution of 4-(2′-fluoro-[1,1′-biphenyl]-4-yl)-N-(3,4,5-trimethoxyphenyl)butanamide (150 mg, 0.35 mmol, 1.0 eq.) in dry CH$_2$Cl$_2$ (5 mL) over 15 minutes. The resultant solution was warmed to room temperature and stirring continued for 8 h. The reaction was then diluted with CH$_2$Cl$_2$ (20 mL), washed with water (25 mL), NaHCO$_3$ (25 mL of a sat. aq. Soln.) before being dried (MgSO$_4$), filtered and concentrated under reduced pressure. The crude mass was purified by flash column chromatography (silica, 1:1 v/v EtOAc:Hex) to give the title compound as a white crystalline solid (116 mg, 87%).

$R_f$: 0.09 (SiO$_2$, 1:1 EtOAc:Hex), m.p.: 168 – 170 °C, [1]H NMR (500 MHz, DMSO-$d_6$) δ 9.41 (s, 1H), 8.76 (s, 2H), 7.73 (s, 1H), 7.53 – 7.37 (m, 4H), 7.32 – 7.27 (m, 4H), 6.63 (s, 2H), 2.66 – 2.55 (m, 2H), 2.28 – 2.20 (m, 2H), 1.93 – 1.82 (m, 2H). [13]C NMR (126 MHz, DMSO-$d_6$) δ 170.0, 159.1 (d, $^1J_{CF}$ = 245.6 Hz), 145.8 (2 C), 141.5, 132.6, 131.1, 130.9, 130.7 (d, $^4J_{CF}$ = 3.5 Hz, 2 C), 129.3 (d, $^3J_{CF}$ = 8.3 Hz), 128.7 (d, $^4J_{CF}$ = 2.8 Hz), 128.6 (2 C), 128.2 (d, $^3J_{CF}$ = 13.0 Hz), 124.9 (d, $^4J_{CF}$ = 3.7 Hz), 116.1 (d, $^2J_{CF}$ = 22.6 Hz), 99.0 (2 C), 35.8, 34.4, 26.8. [19]F NMR (471 MHz, DMSO-$d_6$) δ -118.4. IR (diamond cell, neat) ν$_{max}$: 3520, 3370, 3119, 1628, 1541, 1483, 1448, 1374, 1336, 1290, 1190, 1048, 852, 822, 758, 567 cm$^{-1}$. LRMS (ESI + ) m/z: 404 [(M+Na)$^+$, 100%].

## 4-(2′-fluoro-[1,1′-biphenyl]-4-yl)-N-(3,4,5-trimethoxyphenyl)butanamide (7).

According to the general procedure for PyBOP® amidation, the title compound was synthesised from 4-(2′-fluoro-[1,1′-biphenyl]-4-yl)butanoic acid (300 mg, 1.16 mmol, 1.0 eq.), and 3,4,5-methoxyaniline (216 mg, 1.16 mmol, 1.0 eq.) in dry DMF (10 mL). The crude product was purified by flash column chromatography (silica, 1:1 v/v EtOAc:Hex) to give the title compound as a white crystalline solid (275 mg, 56%).

$R_f$: 0.37 (SiO$_2$, 1:1 EtOAc:Hex), m.p.: 130 – 132 °C, [1]H NMR (500 MHz, DMSO-$d_6$) δ 9.84 (s, 1H), 7.51 – 7.36 (m, 4H), 7.33 – 7.26 (m, 4H), 7.03 (s, 2H), 3.74 (s, 6H), 3.62 (s, 3H), 2.68 (t, $J$ = 7.6 Hz, 2H), 2.35 (t, $J$ = 7.4 Hz, 2H), 1.98 – 1.92 (m, 2H). [13]C NMR (126 MHz, DMSO-$d_6$) δ 170.8,

159.1 (d, $^1J_{CF}$ = 245.7 Hz), 152.7 (2 C), 141.4, 135.5, 133.2, 132.6, 130.6 (d, $^4J_{CF}$ = 3.3 Hz, 2 C), 129.2 (d, $^3J_{CF}$ = 8.4 Hz), 128.7 (d, $^4J_{CF}$ = 2.9 Hz), 128.6 (2 C), 128.2 (d, $^3J_{CF}$ = 13.2 Hz), 124.9 (d, $^4J_{CF}$ = 3.5 Hz), 116.0 (d, $^2J_{CF}$ = 22.6 Hz), 96.8 (2 C), 60.0, 55.6 (2 C), 35.9, 34.3, 26.5. [19]F NMR (471 MHz, DMSO-$d_6$) δ -118.4. IR (diamond cell, neat) ν$_{max}$: 3331, 2937, 1686, 1610, 1544, 1509, 1482, 1446, 1222, 1130, 985, 843, 819, 766, 637, 456 cm$^{-1}$. LRMS (ESI + ) m/z: 446 [(M+Na)$^+$, 100%].

## N-(4-aminophenyl)-4-(2′-fluoro-[1,1′-biphenyl]-4-yl)butanamide (8).

A stirring suspension of 4-(2′-fluoro-[1,1′-biphenyl]-4-yl)-N-(4-nitrophenyl)butanamide (250 mg, 0.66 mmol, 1.0 eq.) and palladium on carbon (10% w/w, 50 mg) in EtOAc (10 mL) was placed under an atmosphere of hydrogen (1 atm.) and stirring continued for 18 h. The reaction volume was filtered through a Celite & basic alumina plug. The residue was washed with EtOAc (3 × 20 mL) and combined filtrates concentrated under reduced pressure. The resultant product was purified by flash column chromatography (silica, 1:1 v/v EtOAc:Hex) to give the title compound as a white crystalline solid (141 mg, 61%).

$R_f$: 0.26 (SiO$_2$, 1:1 EtOAc:Hex), m.p.: 100 – 102 °C, [1]H NMR (500 MHz, DMSO-$d_6$) δ 9.45 (s, 1H), 7.53 – 7.37 (m, 4H), 7.32 (d, $J$ = 8.1 Hz, 2H), 7.28 (t, $J$ = 7.1 Hz, 1H), 7.23 (d, $J$ = 8.6 Hz, 2H), 6.50 (d, $J$ = 8.6 Hz, 2H), 4.81 (br s, 2H), 2.66 (t, $J$ = 7.6 Hz, 2H), 2.28 (t, $J$ = 7.4 Hz, 2H), 1.92 (p, $J$ = 7.6 Hz, 2H). [13]C NMR (126 MHz, DMSO-$d_6$) δ 169.8, 159.1 (d, $^1J_{CF}$ = 245.5 Hz), 144.5, 141.5, 132.6, 130.6 (d, $^4J_{CF}$ = 3.4 Hz, 2 C), 129.3 (d, $^3J_{CF}$ = 8.2 Hz), 129.1, 128.7 (d, $^4J_{CF}$ = 2.9 Hz), 128.6 (2 C), 128.2 (d, $^3J_{CF}$ = 13.1 Hz), 124.9 (d, $^4J_{CF}$ = 3.5 Hz), 120.9 (2 C), 116.0 (d, $^2J_{CF}$ = 22.6 Hz), 113.8 (2 C), 35.6, 34.4, 26.8. [19]F NMR (471 MHz, DMSO-$d_6$) δ -118.4. IR (diamond cell, neat) ν$_{max}$: 3457, 3374, 3276, 1641, 1536, 1514, 1482, 1424, 1275, 1252, 1204, 943, 823, 801, 755, 569, 510, 471 cm$^{-1}$. LRMS (ESI + ) m/z: 371 [(M + H)$^+$, 100%].

## 4-(2′-fluoro-[1,1′-biphenyl]-4-yl)-N-(4-nitrophenyl)butanamide (9).

According to the general procedure for PyBOP® amidation, the title compound was synthesised from 4-(2′-fluoro-[1,1′-biphenyl]-4-yl)butanoic acid (420 mg, 1.6 mmol, 1.03 eq), and 4-nitroaniline (213 mg, 1.55 mmol, 1 eq) in dry DMF (10 mL). The crude product was purified by flash column chromatography (silica, 1:1 v/v EtOAc:Hex) to give the title compound as a yellow powder (473 mg, 78%).

$R_f$: 0.33 (SiO$_2$, 1:1 EtOAc:Hex), m.p.: 118 – 120 °C, [1]H NMR (400 MHz, Chloroform-$d$) δ 8.17 – 8.04 (m, 2H), 7.87 (s, 1H), 7.70 – 7.67 (m, 2H), 7.48 – 7.46 (m, 2H), 7.39 (td, $J$ = 7.7, 1.9 Hz, 1H), 7.32 – 7.27 (m, 1H), 7.26 – 7.24 (m, 2H), 7.21 – 7.11 (m, 2H), 2.76 (t, $J$ = 7.3 Hz, 2H), 2.42 (t, $J$ = 7.4 Hz, 2H), 2.15 – 2.08 (m, 2H). [13]C NMR (101 MHz, Chloroform-$d$) δ 171.7, 159.9 (d, $^1J_{CF}$ = 247.1 Hz), 144.1, 143.4, 140.7, 133.8, 130.7 (d, $^4J_{CF}$ = 3.5 Hz, 2 C), 129.2 (d, $^4J_{CF}$ = 2.9 Hz), 129.0 (d, $^3J_{CF}$ = 8.3 Hz), 128.7 (2 C), 125.2 (2 C), 124.5 (d, $^4J_{CF}$ = 3.6 Hz), 119.1, 119.0 (2 C), 116.2 (d, $^2J_{CF}$ = 22.8 Hz), 36.8, 34.8, 26.5. [19]F NMR (376 MHz, Chloroform-$d$) δ -118.2. IR (diamond cell, neat) ν$_{max}$: 1661, 1595, 1502, 1450, 1406, 1342, 1255, 1208, 1107, 858, 748, 564, 496 cm$^{-1}$. LRMS (ESI-) m/z: 377 [(M-H)$^-$, 100%].

**4-(2′-fluoro-[1,1′-biphenyl]-4-yl)-N-(ropenam-3-yl)butanamide (10).**

An ice cold magnetically stirred solution of 4-(2′-fluoro-[1,1′-biphenyl]-4-yl)butanoic acid (200 mg, 0.77 mmol, 1.0 eq), 3-aminopyridine (80 mg, 0.85 mmol, 1.1 eq) and $^i$Pr$_2$Net (268 µL, 1.54 mmol, 2.0 eq) in DMF (5 mL) was treated with PyBOP® (400 mg, 0.77 mmol, 1.0 eq), allowed to warm to room temperature and stirring continued for 12 h. The reaction mass was diluted with CH$_2$Cl$_2$ (50 mL) and water (50 mL), the separated organic phase was subsequently washed with NaHCO$_3$ (25 mL of a sat. aq. Solution) and brine (100 mL) before being dried (MgSO$_4$), filtered and purified via flash column chromatography (silica, 1:1 v/v EtOAc:Hex) to give the title compound as a white solid (169 mg, 64%).

R$_f$: 0.14 (SiO$_2$, 1:1 EtOAc:Hex), m.p.: 78 – 80 °C, $^1$H NMR (400 MHz, DMSO-$d_6$) δ 10.11 (s, 1H), 8.73 (d, $J$ = 2.6 Hz, 1H), 8.23 (dd, $J$ = 4.7, 1.5 Hz, 1H), 8.03 (ddd, $J$ = 8.3, 2.5, 1.6 Hz, 1H), 7.52 – 7.46 (m, 3H), 7.42 – 7.37 (m, 1H), 7.34 – 7.26 (m, 5H), 2.69 (t, $J$ = 7.5 Hz, 2H), 2.39 (t, $J$ = 7.5 Hz, 2H), 1.95 (p, $J$ = 7.5 Hz, 2H). $^{13}$C NMR (101 MHz, DMSO-$d_6$) δ 171.6, 159.1 (d, $^1J_{CF}$ = 245.6 Hz), 144.0, 141.4, 140.8, 135.9, 132.7, 130.7 (d, $^4J_{CF}$ = 3.5 Hz, 2 C), 129.3 (d, $^3J_{CF}$ = 8.4 Hz), 128.8 (d, $^4J_{CF}$ = 2.9 Hz), 128.7 (2 C), 128.2 (d, $^3J_{CF}$ = 13.2 Hz), 126.0, 124.9 (d, $^4J_{CF}$ = 3.6 Hz), 123.6, 116.1 (d, $^2J_{CF}$ = 22.6 Hz), 35.6, 34.3, 26.4. $^{19}$F NMR (471 MHz, DMSO) δ -118.4. IR (diamond cell, neat) ν$_{max}$: 3333, 2936, 1644, 1605, 1483, 1431, 1298, 1237, 1214, 1134, 1023, 866, 799, 760, 601, 567, 472 cm$^{-1}$. LRMS (ESI + ) m/z: 335 [(M + H)$^+$, 25%], 357 [(M+Na)$^+$, 100%].

**4-(2′-fluoro-[1,1′-biphenyl]-4-yl)-N-(ropenam-4-yl)butanamide (11).**

An ice cold magnetically stirred solution of 4-(2′-fluoro-[1,1′-biphenyl]-4-yl)butanoic acid (200 mg, 0.77 mmol, 1.0 eq.), 4-aminopyridine (80 mg, 0.85 mmol, 1.1 eq.) and $^i$Pr$_2$Net (268 µL, 1.54 mmol, 2.0 eq.) in DMF (5 mL) was treated with PyBOP® (400 mg, 0.77 mmol, 1.0 eq.), allowed to warm to room temperature and stirring continued for 12 h. The reaction mass was diluted with CH$_2$Cl$_2$ (50 mL) and water (50 mL), the separated organic phase was subsequently washed with NaHCO$_3$ (25 mL of a sat. aq. Solution) and brine (100 mL) before being dried (MgSO$_4$), filtered and purified via flash column chromatography (silica, 1:1 v/v EtOAc:Hex) to give the title compound as a white solid (228 mg, 86%).

R$_f$: 0.11 (SiO$_2$, 1:1 EtOAc:Hex), m.p.: 108 – 110 °C, $^1$H NMR (400 MHz, DMSO-$d_6$) δ 11.92 (s, 1H), 8.67 (d, $J$ = 7.2 Hz, 2H), 8.16 (d, $J$ = 7.3 Hz, 2H), 7.50 – 7.42 (m, 3H), 7.41 – 7.35 (m, 1H), 7.33 – 7.13 (m, 4H), 2.73 – 2.64 (m, 2H), 2.58 (t, $J$ = 7.3 Hz, 2H), 2.02 – 1.92 (m, 2H). $^{13}$C NMR (101 MHz, DMSO-$d_6$) δ 173.8, 159.1 (d, $^1J_{CF}$ = 245.6 Hz), 153.0, 142.0 (2 C), 141.2, 132.7, 131.1, 130.6 (d, $^4J_{CF}$ = 3.4 Hz, 2 C), 129.3 (d, $^3J_{CF}$ = 8.4 Hz), 128.7 (d, $^4J_{CF}$ = 2.9 Hz), 128.6 (2 C), 128.1 (d, $^3J_{CF}$ = 13.2 Hz), 124.9 (d, $^4J_{CF}$ = 3.6 Hz), 116.1 (d, $^2J_{CF}$ = 22.5 Hz), 114.2 (2 C), 36.2, 34.1, 25.9. $^{19}$F NMR (376 MHz, DMSO-$d_6$) δ -118.4. IR (diamond cell, neat) ν$_{max}$: 2926, 1716, 1561, 1500, 1483, 1313, 1135, 823, 754, 514 cm$^{-1}$. LRMS (ESI + ) m/z: 335 [(M + H)$^+$, 100%], 357 [(M+Na)$^+$, 40%].

**3-(4-(2′-fluoro-[1,1′-biphenyl]-4-yl)butanamido)pyridine 1-oxide (12).**

An ice cold magnetically stirred solution of 4-(2′-fluoro-[1,1′-biphenyl]-4-yl)-N-(ropenam-3-yl)butanamide (100 mg, 0.30 mmol, 1 eq.) in dry CH$_2$Cl$_2$ (10 mL) was treated with mCPBA (3 eq.), allowed to warm to room temperature and stirring continued for 12 h. The reaction mass was diluted with CH$_2$Cl$_2$ (50 mL) and water (50 mL), the separated organic phase was subsequently washed with NaHCO$_3$ (25 mL of a sat. aq. Solution) and brine (100 mL) before being dried (MgSO$_4$), filtered and purified via flash column chromatography (silica, 1:1 v/v EtOAc:Hex) to give the title compound as a white solid (80 mg, 76%).

R$_f$: 0.10 (SiO$_2$, 1:1 EtOAc:Hex), m.p.: 58 – 60 °C, $^1$H NMR (500 MHz, DMSO-$d_6$) δ 10.29 (s, 1H), 8.71 (s, 1H), 7.94 – 7.93 (m, 1H), 7.51 – 7.46 (m, 3H), 7.41 – 7.36 (m, 2H), 7.36 – 7.26 (m, 5H), 2.67 (t, $J$ = 7.5 Hz, 2H), 2.38 (t, $J$ = 7.4 Hz, 2H), 1.94 (p, $J$ = 7.6 Hz, 2H). $^{13}$C NMR (126 MHz, DMSO-$d_6$) δ 171.8, 159.1 (d, $^1J_{CF}$ = 245.7 Hz), 141.3, 138.4, 133.5, 132.7, 130.6 (d, $^4J_{CF}$ = 3.4 Hz, 2 C), 129.9, 129.3 (d, $^3J_{CF}$ = 8.4 Hz), 128.8 (d, $^4J_{CF}$ = 2.9 Hz), 128.6 (2 C), 128.2 (d, $^3J_{CF}$ = 13.1 Hz), 126.1, 124.9 (d, $^4J_{CF}$ = 3.4 Hz), 116.1 (d, $^2J_{CF}$ = 22.6 Hz), 115.7, 35.6, 34.2, 26.2. $^{19}$F NMR (471 MHz, DMSO-$d_6$) δ -118.4. IR (diamond cell, neat) ν$_{max}$: 2928, 1693, 1575, 1547, 1482, 1415, 1284, 1207, 1146, 980, 789, 751, 672, 590, 550 cm$^{-1}$. LRMS (ESI + ) m/z: 373 [(M+Na)$^+$, 100%].

**4-(2′-fluoro-[1,1′-biphenyl]-4-yl)-N-(1-methyl-1H-pyrazol-2-yl) butanamide (13).**

According to the general procedure for PyBOP® amidation, the title compound was synthesised from 4-(2′-fluoro-[1,1′-biphenyl]-4-yl) butanoic acid (150 mg, 0.58 mmol, 1.0 eq.), and aminopyrazole (62 mg, 0.64 mmol, 1.1 eq.) in dry DMF (10 mL). The crude product was purified by flash column chromatography (silica, 1:1 v/v EtOAc:Hex) to give the title compound as a gummy solid (110 mg, 56%).

R$_f$: 0.11 (SiO$_2$, 1:1 EtOAc:Hex), $^1$H NMR (500 MHz, DMSO-$d_6$) δ 9.88 (s, 1H), 7.53 – 7.48 (m, 3H), 7.42 – 7.37 (m, 1H), 7.34 – 7.27 (m, 5H), 6.18 (d, $J$ = 1.9 Hz, 1H), 3.65 (s, 3H), 2.69 (t, $J$ = 7.6 Hz, 2H), 2.40 (t, $J$ = 7.5 Hz, 2H), 1.95 (p, $J$ = 7.6 Hz, 2H). $^{13}$C NMR (126 MHz, DMSO-$d_6$) δ 170.9, 159.1 (d, $^1J_{CF}$ = 245.6 Hz), 141.3, 137.3, 136.5, 132.7, 130.6 (d, $^4J_{CF}$ = 3.4 Hz, 2 C), 129.3 (d, $^3J_{CF}$ = 8.4 Hz), 128.7 (d, $^4J_{CF}$ = 2.9 Hz), 128.6 (2 C), 128.2 (d, $^3J_{CF}$ = 13.0 Hz), 124.9 (d, $^4J_{CF}$ = 3.4 Hz), 116.1 (d, $^2J_{CF}$ = 22.5 Hz), 98.7, 35.5, 34.7, 34.2, 26.4. $^{19}$F NMR (471 MHz, DMSO-$d_6$) δ -118.4. IR (diamond cell, neat) ν$_{max}$: 3268, 1663, 1543, 1482, 1249, 1194, 1106, 968, 928, 822, 800, 758, 691, 565, 532, 461 cm$^{-1}$. LRMS (ESI + ) m/z: 360 [(M+Na)$^+$, 100%].

**3-(2′-fluoro-[1,1′-biphenyl]-4-yl)propanoic acid.**

Tetrakis(triphenylphosphine)palladium(0) (554.7 mg, 0.48 mmol, 0.05 mol%) was added to a stirred suspension of 2-fluorophenylboronic acid (1.61 g, 11.5 mmol, 1.2 eq.), Cs$_2$CO$_3$ (11.2 g, 34.5 mmol, 3.6 eq.) and 4-bromophenylpropionic acid (2.2 g, 9.6 mmol, 1.0 eq.) in dry degassed THF/water (100 mL, 9:1 v/v mixture) and the resultant mixture stirred at reflux for 8 h. The resultant solution was diluted with HCl (100 mL, 1 M) and extracted with EtOAc (3 × 50 mL) the combined organic phase was subsequently washed with brine, before being dried (MgSO$_4$), filtered and concentrated under reduced pressure. The resultant acid was purified by flash column chromatography (silica, 1:1 v/v EtOAc:Hex) to give the title compound as a white crystalline solid (1.3 g, 54%).

$^1$H NMR (300 MHz, DMSO-$d_6$) δ 12.16 (br s, 1H), 7.58 – 7.16 (m, 8H), 2.88 (t, $J$ = 7.8 Hz, 2H), 2.67 – 2.54 (m, 2H). $^{13}$C NMR (75 MHz, DMSO-$d_6$) δ 173.7, 159.1 (d, $^1J_{CF}$ = 245.7 Hz), 140.6, 132.8, 130.6 (d, $^4J_{CF}$ = 6.0 Hz, 2 C), 129.3 (d, $^3J_{CF}$ = 7.9 Hz), 128.7 (2 C), 128.5, 128.1 (d, $^3J_{CF}$ = 12.3 Hz), 124.9 (d, $^4J_{CF}$ = 3.6 Hz), 116.0 (d, $^2J_{CF}$ = 22.6 Hz), 35.0, 30.0. $^{19}$F NMR (282 MHz, DMSO-$d_6$) δ -118.4. IR (diamond cell, neat) $v_{max}$: 3187, 1696, 1483, 1410, 1216, 1009, 940, 814, 755, 666, 566, cm$^{-1}$. LRMS (ESI-) m/z: 243 [(M-H)$^-$, 100%].

### 3-(2′-fluoro-[1,1′-biphenyl]-4-yl)propenamide.

3-(2′-fluoro-[1,1′-biphenyl]-4-yl)propanoic acid (1.0 g, 4.1 mmol, 1.0 eq.) and 1,1′-carbonyldiimidazole (854 mg, 5.2 mmol, 1.26 eq.) were stirred for 1 h at room temperature in THF (4 mL) under a N$_2$ atmosphere. The reaction was cooled on ice then aqueous ammonia (28%, 2.25 mL) was added. The reaction was stirred for 4 h, allowing the solution to warm to room temperature. The solvent was removed by rotary evaporation and the residue dissolved in dichloromethane (15 mL) and washed with aqueous sodium hydroxide (1 M, 5 mL), then aqueous hydrochloric acid (1 M, 5 mL) and then water (5 mL). The organic layer was dried (MgSO4), filtered and evaporated to dryness to yield the title compound as a white powder (607 mg, 61%).

$^1$H NMR (300 MHz, DMSO-$d_6$) δ 7.57 – 7.17 (m, 8H), 6.79 (s, 2H), 2.86 (t, $J$ = 7.9 Hz, 2H), 2.41 (t, $J$ = 7.9 Hz, 2H). $^{13}$C NMR (75 MHz, DMSO-$d_6$) δ 173.3, 157.4 (d, $^1J_{CF}$ = 246.3 Hz), 141.2, 132.6, 130.5 (d, $^4J_{CF}$ = 7.9 Hz, 2 C), 129.2 (d, $^3J_{CF}$ = 8.3 Hz), 128.6 (2 C), 128.4, 128.1 (d, $^3J_{CF}$ = 12.1 Hz), 124.8, 116.0 (d, $^2J_{CF}$ = 22.7 Hz), 36.4, 30.5. $^{19}$F NMR (282 MHz, DMSO) δ -118.4. IR (diamond cell, neat) $v_{max}$: 3400, 3180, 1650, 1482, 1412, 1009, 806, 754, 624 cm$^{-1}$. LRMS (ESI + ) m/z: 266 [(M+Na)$^+$, 100%].

### 3-(2′-fluoro-[1,1′-biphenyl]-4-yl)propan-1-amine.

A solution of amide (500 mg, 2.1 mmol, 1.0 eq) in THF (8 mL) was treated with LiAlH$_4$ (312 mg, 8.2 mmol, 3.9 eq) at 0 °C and stirred under a N$_2$ atmosphere whilst warming to room temperature. After 2 h, the reaction was heated at reflux for 16 h and then cooled on ice. Chilled water (300 μL) was added dropwise, with vigorous stirring, and then followed by aqueous sodium hydroxide (15% w/v, 300 μL) and additional water (1 mL). The solution was left stirring at room temperature until effervescence had ceased and the grey powder had turned white (30 min). The solution was dried (MgSO$_4$) and then filtered. The precipitate was washed with additional dichloromethane (2 ×10 mL). The

filtrate in each case was combined, and solvent removed under reduced pressure. The crude oil thus obtained was purified by flash column chromatography (silica, 0.5:9.5 v/v MeOH (saturated with NH$_3$):CH$_2$Cl$_2$) to give the title compound as a colourless wax (375 mg, 78%).

$^1$H NMR (400 MHz, DMSO-$d_6$) δ 7.60 – 7.12 (m, 8H), 4.17 (br s, 2H), 3.05 – 2.88 (m, 2H), 2.75 – 2.52 (m, 2H), 1.77 – 1.64 (m, 2H). $^{19}$F NMR (282 MHz, DMSO-$d_6$) δ -118.4. IR (diamond cell, neat) $v_{max}$: 3334, 2923, 1611, 1481, 1314, 814, 751, 551 cm$^{-1}$. LRMS (ESI + ) m/z: 230 [(M+Na)$^+$, 100%].

### 2-(2′-fluoro-[1,1′-biphenyl]-4-yl)acetic acid.

Tetrakis(triphenylphosphine)palladium(0) (554.7 mg, 0.48 mmol, 0.05 mol%) was added to a stirred suspension of 2-fluorophenylboronic acid (1.61 g, 11.5 mmol, 1.2 eq.), Cs$_2$CO$_3$ (11.2 g, 34.5 mmol, 3.6 eq.) and 4-bromophenylacetic acid (2.1 g, 9.6 mmol, 1.0 eq.) in dry degassed THF/water (100 mL, 9:1 v/v mixture) and the resultant mixture stirred at reflux for 8 h. The resultant solution was diluted with HCl (100 mL, 1 M) and extracted with EtOAc (3 ×50 mL) the combined organic phase was subsequently washed with brine, before being dried (MgSO$_4$), filtered and concentrated under reduced pressure. The resultant acid was purified by flash column chromatography (silica, 1:1 v/v EtOAc:Hex) to give the title compound as a white crystalline solid (2.1 g, 95%).

### 3-(2′-fluoro-[1,1′-biphenyl]-4-yl)-*N*-(4-methoxybenzyl)propenamide (14).

3-(2′-fluoro-[1,1′-biphenyl]-4-yl)propanoic acid (100 mg, 0.41 mmol, 1.0 eq.) and 1,1′-carbonyldiimidazole (85.4 mg, 0.52 mmol, 1.27 eq.) were stirred for 1 h at room temperature in THF (1 mL) under a N$_2$ atmosphere. The reaction was cooled on ice then 4-aminomethyl phenol (67 mg, 0.49 mmol, 1.2 eq.) was added. The reaction was stirred for 4 h, allowing the solution to warm to room temperature. The solvent was removed by rotary evaporation and the residue dissolved in dichloromethane (15 mL) and washed with aqueous sodium hydroxide (1 M, 5 mL), then aqueous hydrochloric acid (1 M, 5 mL) and then water (5 mL). The organic layer was dried (MgSO$_4$), filtered and evaporated to dryness to yield the title compound as a white powder (111 mg, 74%).

R$_f$: 0.37 (SiO$_2$, 1:1 EtOAc:Hex), m.p.: 111 – 113 °C, $^1$H NMR (500 MHz, DMSO-$d_6$) δ 8.27 (t, $J$ = 5.9 Hz, 1H), 7.51 – 7.48 (m, 1H), 7.45 – 7.44 (m, 2H), 7.41 – 7.37 (m, 1H), 7.32 – 7.26 (m, 4H), 7.07 (d, $J$ = 8.5 Hz, 2H), 6.82 (d, $J$ = 8.7 Hz, 2H), 4.19 (d, $J$ = 5.8 Hz, 2H), 3.68 (s, 3H), 2.90 (t, $J$ = 7.6 Hz, 2H), 2.48 (t, $J$ = 7.6 Hz, 2H). $^{13}$C NMR (126 MHz, DMSO-$d_6$) δ 171.0, 159.1 (d, $^1J_{CF}$ = 245.7 Hz), 158.1, 141.0, 132.7, 131.4, 130.6 (d, $^4J_{CF}$ = 3.5 Hz, 2 C), 129.3 (d, $^3J_{CF}$ = 8.3 Hz), 128.6 (d, $^4J_{CF}$ = 3.0 Hz), 128.6 (2 C), 128.4 (2 C), 128.1 (d, $^3J_{CF}$ = 13.0 Hz), 124.9 (d, $^4J_{CF}$ = 3.5 Hz), 116.1 (d, $^2J_{CF}$ = 22.6 Hz), 113.6 (2 C), 55.0, 41.4, 36.8, 30.8. $^{19}$F NMR (471 MHz, DMSO-$d_6$) δ -118.4. IR (diamond cell, neat) $v_{max}$: 3297, 1632, 1511, 1483, 1244, 1217, 1175, 1106, 1037, 823, 754, 565, 515 cm$^{-1}$. LRMS (ESI + ) m/z: 386 [(M+Na)$^+$, 100%].

### 3-(2′-fluoro-[1,1′-biphenyl]-4-yl)-*N*-(4-hydroxybenzyl) propenamide (15).

3-(2′-fluoro-[1,1′-biphenyl]-4-yl)propanoic acid (100 mg, 0.41 mmol, 1.0 eq.) and 1,1′-carbonyldiimidazole (85.4 mg, 0.52 mmol, 1.27 eq.) were stirred for 1 h at room temperature in THF (1 mL) under a N$_2$ atmosphere. The reaction was cooled on ice then 4-aminomethyl phenol (61 mg, 0.49 mmol, 1.2 eq.) was added. The reaction was stirred for 4 h, allowing the solution to warm to room temperature. The solvent was removed by rotary evaporation and the residue dissolved in dichloromethane (15 mL) and washed with aqueous sodium hydroxide (1 M, 5 mL), then aqueous hydrochloric acid (1 M, 5 mL) and then water (5 mL). The organic layer was dried (MgSO4), filtered and evaporated to dryness to yield the title compound as a white powder (126 mg, 88%).

R$_f$: 0.33 (SiO$_2$, 1:1 EtOAc:Hex), m.p.: 153 – 155 °C $^1$H NMR (500 MHz, DMSO-$d_6$) δ 9.25 (s, 1H), 8.23 (t, *J* = 5.8 Hz, 1H), 7.50 (td, *J* = 7.9, 1.7 Hz, 1H), 7.45 (dd, *J* = 8.2, 1.8 Hz, 2H), 7.43 – 7.36 (m, 1H), 7.33 – 7.28 (m, 4H), 6.99 (d, *J* = 8.4 Hz, 2H), 6.68 (d, *J* = 8.4 Hz, 2H), 4.15 (d, *J* = 5.8 Hz, 2H), 2.90 (t, *J* = 7.6 Hz, 2H), 2.47 (t, *J* = 7.6 Hz, 2H). $^{13}$C NMR (126 MHz, DMSO-$d_6$) δ 171.0, 159.1 (d, $^1J_{CF}$ = 245.6 Hz), 156.2, 141.1, 132.7, 130.6 (d, $^4J_{CF}$ = 3.3 Hz, 2 C), 129.6, 129.3 (d, $^3J_{CF}$ = 8.4 Hz), 128.6 (d, *J* = 2.9 Hz), 128.5 (d, $^4J_{CF}$ = 2.3 Hz), 128.2 (d, $^3J_{CF}$ = 13.0 Hz), 124.9 (d, $^4J_{CF}$ = 3.5 Hz), 116.1 (d, $^2J_{CF}$ = 22.5 Hz), 114.9 (2 C), 41.6, 36.8, 30.8. $^{19}$F NMR (471 MHz, DMSO-$d_6$) δ -118.4. IR (diamond cell, neat) ν$_{max}$: 3320, 1613, 1515, 1481, 1434, 1205, 1105, 841, 807, 761, 567, 498, 475 cm$^{-1}$. LRMS (ESI + ) m/z: 372 [(M+Na)$^+$, 100%].

### 2-(2′-fluoro-[1,1′-biphenyl]-4-yl)-*N*-(4-hydroxyphenethyl) acetamide (16).

According to the general procedure for PyBOP® amidation, the title compound was synthesised from 3-(2′-fluoro-[1,1′-biphenyl]-4-yl) acetic acid (200 mg, 0.86 mmol, 1.0 eq.) and 4-aminoethyl phenol (143 mg, 1.0 mmol, 1.16 eq.). The crude product was purified by flash column chromatography (silica, 1:1 v/v EtOAc:Hex) to give the title compound as a white crystalline solid (211 mg, 70%).

R$_f$: 0.30 (SiO$_2$, 1:1 EtOAc:Hex), m.p.: 83 – 87 °C, $^1$H NMR (500 MHz, DMSO-$d_6$) δ 9.16 (s, 1H), 8.31 (s, 1H), 8.11 (t, *J* = 5.6 Hz, 1H), 7.51 (t, *J* = 7.9 Hz, 1H), 7.47 (d, *J* = 8.1 Hz, 1H), 7.42 – 7.38 (m, 1H), 7.32 (d, *J* = 8.1 Hz, 2H), 7.30 – 7.25 (m, 2H), 6.97 (d, *J* = 8.3 Hz, 2H), 6.68 (d, *J* = 8.4 Hz, 2H), 3.45 (s, 2H), 3.26 – 3.21 (m, 2H), 2.61 (t, *J* = 7.3 Hz, 2H). $^{13}$C NMR (126 MHz, DMSO-$d_6$) δ 169.8, 159.1 (d, $^1J_{CF}$ = 245.8 Hz), 155.6, 136.2, 133.1, 131.1 (d, $^2J_{CF}$ = 30.0 Hz), 130.7 (d, $^4J_{CF}$ = 3.4 Hz, 2 C), 129.5 (2 C), 129.3 (d, $^3J_{CF}$ = 8.3 Hz), 129.2 (2 C), 128.6 (d, $^4J_{CF}$ = 2.8 Hz), 128.1 (d, $^3J_{CF}$ = 13.2 Hz), 124.9 (d, $^4J_{CF}$ = 3.6 Hz), 116.1 (d, $^2J_{CF}$ = 22.5 Hz), 115.1 (2 C), 79.2, 42.1, 40.7, 34.3. $^{19}$F NMR (471 MHz, DMSO-$d_6$) δ -118.4. IR (diamond cell, neat) ν$_{max}$: 3267, 2929, 1609, 1561, 1513, 1483, 1360, 1210, 813, 753, 580, 462 cm$^{-1}$. LRMS (ESI + ) m/z: 372 [(M+Na)$^+$, 100%].

### N-(3-(2′-fluoro-[1,1′-biphenyl]-4-yl)propyl)-4-hydroxybenzamide (17).

An ice cold magnetically stirred solution of 3-(2′-fluoro-[1,1′-biphenyl]-4-yl)propan-1-amine (200 mg, 0.87 mmol, 1.0 eq.), 4-hydroxybenzoic acid (141 mg, 1.04 mmol, 1.2 eq.) and $^i$Pr$_2$Net (303 μL, 1.74 mmol, 2.0 eq.) in DMF (5 mL) was treated with PyBOP® (452 mg, 0.87 mmol, 1.0 eq.), allowed to warm to room temperature and stirring continued for 12 h. The reaction mass was diluted with CH$_2$Cl$_2$ (50 mL) and water (50 mL), the separated organic phase was subsequently washed with NaHCO$_3$ (25 mL of a sat. aq. Solution) and brine (100 mL) before being dried (MgSO$_4$), filtered and purified via flash column chromatography (silica, EtOAc).

$^1$H NMR (400 MHz, DMSO-$d_6$) δ 9.03 (dd, *J* = 2.3, 0.9 Hz, 1H), 8.71 – 8.67 (m, 2H), 8.21 – 8.17 (m, 2H), 7.64 – 7.14 (m, 10H), 3.35 – 3.29 (m, 2H), 2.72 – 2.62 (m, 2H), 1.93 – 1.83 (m, 2H). $^{13}$C NMR (101 MHz, DMSO-$d_6$) δ 165.3, 159.6 (d, $^1J_{CF}$ = 245.5 Hz), 152.2, 148.8, 141.9, 135.4, 133.1, 131.1 (d, $^4J_{CF}$ = 3.5 Hz), 130.6, 129.7 (d, $^3J_{CF}$ = 8.3 Hz), 129.4 (d, $^3J_{CF}$ = 7.3 Hz), 129.2 (d, $^4J_{CF}$ = 2.9 Hz), 129.1, 128.8 (d, $^3J_{CF}$ = 4.6 Hz), 127.0 (d, $^2J_{CF}$ = 11.6 Hz), 126.2, 125.3 (d, $^4J_{CF}$ = 3.4 Hz), 123.9, 116.5 (d, $^2J_{CF}$ = 22.7 Hz), 32.8 (2 overlapping signals), 31.1. $^{19}$F NMR (376 MHz, DMSO-$d_6$) δ -118.4. IR (diamond cell, neat) ν$_{max}$: 3302, 3027, 2948, 2465, 1626, 1588, 1544, 1481, 1448, 1431, 1406, 1362, 1317, 1211, 820, 757, 742, 708, 697, 622, 535 cm$^{-1}$. LRMS (ESI-) m/z 348 [(M-H)$^-$, 100%].

### *N*-(3-(2′-fluoro-[1,1′-biphenyl]-4-yl)propyl)nicotinamide (WJA88).

An ice cold magnetically stirred solution of 3-(2′-fluoro-[1,1′-biphenyl]-4-yl)propan-1-amine (200 mg, 0.87 mmol, 1.0 eq.), nicotinic acid (129 mg, 1.04 mmol, 1.2 eq.) and $^i$Pr$_2$Net (303 μL, 1.74 mmol, 2.0 eq.) in DMF (5 mL) was treated with PyBOP® (452 mg, 0.87 mmol, 1.0 eq.), allowed to warm to room temperature and stirring continued for 12 h. The reaction mass was diluted with CH$_2$Cl$_2$ (50 mL) and water (50 mL), the separated organic phase was subsequently washed with NaHCO$_3$ (25 mL of a sat. aq. Solution) and brine (100 mL) before being dried (MgSO$_4$), filtered and purified via flash column chromatography (silica, EtOAc).

R$_f$: 0.35 (SiO$_2$, EtOAc), m.p.: 96 – 98 °C, $^1$H NMR (400 MHz, DMSO-$d_6$) δ 9.01 (d, *J* = 1.5 Hz, 1H), 8.70 – 8.67 (m, 1H), 8.18 (dt, *J* = 7.9, 2.0 Hz, 1H), 7.53 – 7.46 (m, 4H), 7.42 – 7.26 (m, 5H), 3.36 – 3.29 (m, 2H), 2.70 (t, *J* = 7.6 Hz, 2H), 1.89 (p, *J* = 7.6 Hz, 2H). $^{13}$C NMR (101 MHz, DMSO-$d_6$) δ 164.8, 159.1 (d, $^1J_{CF}$ = 245.6 Hz), 151.7, 148.3, 141.4, 134.9, 132.6, 130.6 (d, $^4J_{CF}$ = 3.4 Hz, 2 C), 130.1, 129.3 (d, $^3J_{CF}$ = 8.4 Hz), 128.71 (d, $^4J_{CF}$ = 2.9 Hz), 128.68 (2 C), 128.2 (d, $^3J_{CF}$ = 13.2 Hz), 124.9 (d, $^4J_{CF}$ = 3.6 Hz), 123.4, 116.0 (d, $^2J_{CF}$ = 22.6 Hz), 32.3, 30.6. $^{19}$F NMR (282 MHz, DMSO-$d_6$) δ -118.4. IR (diamond cell, neat) ν$_{max}$: 3302, 3027, 2948, 2465, 1626, 1588, 1544, 1481, 1448, 1431, 1406, 1362, 1317, 1211, 820, 757, 742, 708, 697, 622, 535 cm$^{-1}$. LRMS (ESI + ) m/z 357 [(M+Na)$^+$, 100%]. HRMS (ESI + ) calcd for C$_{21}$H$_{19}$FN$_2$NaO [M+Na]+, 357.13736; Found 357.13704.

### Reporting summary
Further information on research design is available in the Nature Portfolio Reporting Summary linked to this article.

## Data availability

The source data file containing raw data for all figures and uncropped scans of western blots are provided with this paper. All next generation RNAsequencing and ChIPsequencing datasets have been deposited into Gene Expression Omnibus (GEO) under the accession number GSE279066 (URL: https://www.ncbi.nlm.nih.gov/geo/query/acc.cgi?acc=GSE279066). The Heidelberg glioblastoma tumour ChIPseq and RNAseq dataset used for this paper is under accession number GSE121723 (URL: https://www.ncbi.nlm.nih.gov/geo/query/acc.cgi?acc=GSE121723)[39]. The mass spectrometry proteomics raw data files have been deposited to the ProteomeXchange Consortium via the PRIDE partner repository with the dataset identifier PXD050643. Lipidomics raw data (Study ID ST004262; DatatrackID:6506) have been deposited in the Metabolomics Workbench[74] and made publicly available (URL: https://doi.org/10.21228/M82R9H). The remaining data are available within the Article, Supplementary Data, Supplementary Information and Source Data files. Source data are provided with this paper.

## Code availability

Custom RStudio scripts for RNAseq and ChIPseq data analysis can be access in Zenodo (URL: https://doi.org/10.5281/zenodo.17248508; URL: https://doi.org/10.5281/zenodo.17282371)[75,76].

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

## Acknowledgements

We thank Paul Brennan (University of Oxford, UK) for providing KDM5 and KDM6 inhibitors KDOAM-25, KDOPZ-36A, KDOBA-67a and KDOBA-97a. We thank Andrew Jenner (Bioanalytical Mass Spectrometry Facility at the Mark Wainwright Analytical Centre, University of New South Wales, Sydney) for technical and scientific support during the GC-MS analysis of cholesterol. We acknowledge Duan Ni (Charles Perkins Centre, University of Sydney) for intellectual input on patients' data analyses. We thank Jessica Ho (Duke-NUS Medical School, Singapore) for guidance with the design of PRDM9 primers. We acknowledge the Sydney Mass Spectrometry Core Research Facility at the University of

Sydney and thank the technical staff for the maintenance of the instruments and support of the Australian National Fabrication Facility (ANFF) – Materials Node. This study was supported by NHMRC grant APP2003150 to L.M., J.R.B. and T.G.J.; NHMRC grant APP1153961 to L.M., M.K., T.G.J.; Tour de Cure grant (RSP-370-FY2023) to L.M., and the Arto Hardy Family and Tour de Cure (FY2023-2025) grants to J.M.C. and E.T.C. E.K., B.C., R.H.A., J.R.S. and S.L.H. were funded by the Australian Government Research Training Program (RTP) Scholarship.

## Author contributions

G.L.J.: conceptualization, methodology, validation, formal analysis, investigation, data curation, writing – review and editing, visualisation. E.G.K., B.C., J.R.S., R.H.A., D.F., M.S., C.C., O.C.M., W.D.P., A.R., M.H., D.C.I., T.C., T.Y.D., J.K., M.K., R.P.: methodology, investigation and validation. J.K.K.L., H.K., P.Y.: software, formal analysis, visualisation. W.T.J., A.P.M., J.R.B.: chemistry – conceptualisation, methodology, investigation, formal analysis. S.L.H., E.T.C., J.M.C.: methodology, investigation, formal analysis, and funding acquisition. L.L.: methodology and resources. G.G.N.: resources and supervision. B.W.D.: resources, methodology, writing – review & editing. E.G.: methodology, writing – review & editing. T.G.J.: methodology, supervision, funding acquisition. M.K.: conceptualisation, supervision, funding acquisition. Y.F.: methodology, validation, data curation, visualisation. L.H.: methodology, validation, data curation, visualisation, supervision, funding acquisition. A.S.D.: methodology, formal analysis, data curation, writing – review & editing, supervision, funding acquisition. L.M.: conceptualisation, methodology, validation, data curation, writing – original and revised drafts, visualisation, supervision, project administration and funding acquisition.

## Competing interests

L.M., M.K. and W.T.J. are inventors on two patents related to the discovery and development of brain permeable microtubule-targeting agents, including CMPD1 and WJA88 used in this study (WO2016/119017; WO2019/148244). G.L.J., B.C., M.K. and L.M. are inventors on PCT/AU2024/050979 which covers the drug combinations described in this study for therapeutic use in cancer. E.G. received research funds from AZ and Prelude Therapeutics (for unrelated projects), EG is a cofounder and shareholder of Immunoa Pte.Ltd and cofounder shareholder, consultant, and advisory board member of Prometeo Therapeutics. Other authors declare no conflict of interest.

## Additional information

George L. Joun[1,2], Emma G. Kempe [1,2], Brianna Chen[1,2], Jayden R. Sterling[1,2], Ramzi H. Abbassi [1,2], Dana Friess [3,4], Matthew Singleton[3], Chandra Choudhury[3,4], Oana C. Marian [1,2], W. Daniel du Preez[1,2], Ariadna Recasens[1,2], Teleri Clark[2,5], Tian Y. Du [2,5], Jason K. K. Low [5], Hani Kim[2,6], Pengyi Yang [2,6], Jasmine Khor[1,2], Monira Hoque[1,2], Dinesh C. Indurthi [1,2], Mani Kuchibhotla[7,8], Ranjith Palanisamy[7,8], William T. Jorgensen[9], Andrew P. Montgomery[9], Jennifer R. Baker[10], Sarah L. Higginbottom[11,12], Eva Tomaskovic-Crook[1,11,12], Jeremy M. Crook [1,11,12], Lipin Loo [2,5], Bryan W. Day[3], G. Gregory Neely [2,5], Ernesto Guccione [13,14,15], Terrance G. Johns[7,8], Michael Kassiou [9], Yuchen Feng[1,2], Lachlan Harris[3,4,16], Anthony S. Don [1,2] & Lenka Munoz [1,2] ✉

[1]School of Medical Sciences, Faculty of Medicine and Health, The University of Sydney, Sydney, NSW, Australia. [2]Charles Perkins Centre, The University of Sydney, Sydney, NSW, Australia. [3]QIMR Berghofer Medical Research Institute, Brisbane, QLD, Australia. [4]The University of Queensland, Brisbane, QLD, Australia. [5]School of Life and Environmental Sciences, Faculty of Science, The University of Sydney, Sydney, NSW, Australia. [6]School of Mathematics and Statistics, Faculty of Science, The University of Sydney, Sydney, NSW, Australia. [7]Centre for Child Health Research, Medical School, University of Western Australia, Crawley, WA, Australia. [8]Cancer Centre, Telethon Kids Institute, Nedlands, WA, Australia. [9]School of Chemistry, Faculty of Science, The University of Sydney, Sydney, NSW, Australia. [10]School of Science, Faculty of Science, Medicine and Health, University of Wollongong, Wollongong, NSW, Australia. [11]Arto Hardy Family Biomedical Innovation Hub, Chris O'Brien Lifehouse, Camperdown, NSW, Australia. [12]Intelligent Polymer Research Institute, AIIM Facility, Innovation Campus, University of Wollongong, Fairy Meadow, NSW, Australia. [13]Department of Oncological Sciences, Tisch Cancer Institute, Icahn School of Medicine at Mount Sinai, New York, NY, USA. [14]Center for Therapeutics Discovery, Department of Oncological Sciences and Pharmacological Sciences, Tisch Cancer Institute, Icahn School of Medicine at Mount Sinai, New York, NY, USA. [15]Bioinformatics for Next Generation Sequencing (BiNGS) Shared Resource Facility, Icahn School of Medicine at Mount Sinai, New York, NY, USA. [16]Queensland University of Technology, Brisbane, QLD, Australia. ✉e-mail: lenka.munoz@sydney.edu.au

