## [Transparent Peer Review file · Nature Communications]

Histone methyltransferase PRDM9 promotes survival of drug-tolerant persister cells in glioblastoma

Corresponding Author: Professor Lenka Munoz

Version 0:

Reviewer comments:

Reviewer #1

(Remarks to the Author)

In this article, authors uncover the role of PRDM9, a methyltransferase expressed primarily on germ cells, in mediating resistance to microtubule-targeting agents (MTA) in glioblastoma. They show that PRDM9 induces H3K4-me3 in GBM cells following MTA administration, which positively regulates cholesterol production to counteract chemotherapy-induced oxidative stress.

While PRDM9 has been previously reported to play a role in the chromosomal instability in cancer, it has never been linked with treatment resistance or cholesterol metabolism.

Altogether, the authors present exciting data and go as far as developing a novel stable and blood-brain permeable MTA analog. They tested the developed compound in combination with a PRDM9 inhibitor in an aggressive GBM animal model and showed significant survival benefits of this strategy. On the other hand, the data opens some interesting unexplored angles, and some of the data would benefit from further validation.

Major comments:

1) Authors provide a very interesting observation: in drug-tolerant persisters (DTP) cells that survived 14d treatment, H3K4-me3 was significantly downregulated in one DTP cell line and unaffected in the second cell line. Additionally, cholesterol was significantly downregulated in 14d treated cells. On the other hand, in cells that received only 3d of treatment, H3K4-me3 was upregulated, as was cholesterol. However, the underlying reasons why DTP cells cease to depend on cholesterol and H3K4-me3 after the 6th day of MTA treatment remain unexplored.

a. What are levels of PRDM9 on 14d and 6d time points?

b. What changes occur between days 3 and 6 that reduce cell dependence on cholesterol?

c. The importance of timely implementation of PRDM9 inhibition to prevent cells from becoming resistant to MTA is a very interesting point that should be emphasized more.

2) Authors suggest that the upregulation of cholesterol helps to compensate for cholesterol oxidation due to ROS. This does not necessarily align with evidence that cholesterol is essential only up to 6d of treatment. The proposed mechanism can benefit from further validation and can solve the discrepancy indicated in the previous question.

a. What are the ROS levels at 3d, 6d and 14d timepoints?

b. Can cell proliferation be rescued by adding antioxidants in cells treated with CMPD1 and MRK-740/PRDM9 KO?

Minor comments:

1) Genetic manipulation to prevent cholesterol production will be a good second line of evidence to support DepMap data and data observed in the presence of inhibitor and ensure that findings are not related to the inhibitor's off-target effect.

2) In Figure 2F total H3 should be presented.

3) PRDM9 is unlikely to be the main writer regulating H3K4 methylation. While the overall focus on this non-essential enzyme is well justified and explained, understanding whether other H3K4-me3 writers (SET1, MLL) are affected in a similar fashion will provide important insight into the resistance mechanism. Do MTAs activate all the writers of the H3K4-me3, or only PRDM9? This is even more important since data show that MRK-740 caused H3K4-me3 depletion only in CMPD1-treated cells but not in naïve ones.

4) It would be beneficial to additionally validate the safety of newly developed compound in normal human astrocytes and neural stem cells.

5) In Figure 3 I, the effect of the KO should be analyzed in relation to the NT control and not the parent cell line. If NT and transfection cause cytotoxicity, that has to be considered when judging the effect of PRDM9 KO.

6) Having representative images of the tumour bioluminescence in animal model will make in vivo data more impactful.

7) Some western blots (Figure 2 E and G, Figure 3 E, Figure 6 B - C) lack quantification, which should be fixed to provide

statistical robustness.

8) For the cases where densitometry quantification for the western blots is provided, were these numbers normalized to the total protein and/or total histone levels before being compared between groups?

Reviewer #2

(Remarks to the Author)

Joun et al. have taken a multi-scale integrative approach to understand how epigenetic changes support survival of glioblastoma cells resistant to microtubule-targeting (MT) drugs. The presence of drug-tolerant persistent cells post-chemotherapy is the primary cause for tumour re-occurrence. Understanding the molecular mechanisms that drive drug-resistance is valuable for designing combination therapies to increase the effectiveness of the treatment. This study has uncovered an interesting link between epigenetics and cellular metabolism in conferring drug resistance to microtubule-targeting drugs. The authors have also leveraged their discovery to demonstrate the efficacy of combining inhibition of cholesterol efflux along with microtubule depolymerization in increasing the survival of preclinical xenograft models. The manuscript is well written and easy to follow. However, there are some missing links, and all key findings of this study mainly come from one glioblastoma cell line which makes the generalisation of this discovery questionable. Moreover, the dynamics of H3K4me3 within this treatment window raises a concern on the contribution of histone methylation vs. histone proteolytic cleavage in conferring MT-drug resistance (details below). The in vivo studies are also preliminary, convincing evidence of these xenografts being a faithful model of glioblastoma in patients is lacking. Recapitulating key results in a few additional glioblastoma cells, validating the in vivo work, and clarifying the H3K4me3 dynamics in MT-drug resistance will strengthen the findings and provide higher-confidence in generalizing the discovery for glioblastoma biology.

Major concerns:

- (1) The authors demonstrate down regulation of cholesterol biosynthetic genes in RKI1-persister cells compared to the parent cells on a transcriptomic level and further confirm the functional impact by quantifying the lipidomic profile. Does this observation hold true for the second set of cells (FPW1) used in this study? RT-PCR of cholesterol biosynthetic enzymes in persister vs. parent FPW1 will be necessary to understand the changes in metabolism associated with developing MT-drug tolerance.
- (2) The authors have profiled the histone extracts of parent and persister cells. It will be very useful for the scientific community to have the information on all histone peptides identified in this experiment in the form of a table.
- (3) In fig 2f, the faster migration H3 band is missing in H3K27me3 and H3K27me2 blot. Can this be explained? Alternatively, a blot for total H3 in parent vs. persister will be helpful in making the point of proteolytically cleaved isoform and potentially the proportions of two isoforms present in the persistent cells.
- (4) When does the proteolytically cleaved H3 variant appear within the 14-day treatment window. To understand the dynamics of H3K4me3 loss vs. proteolytic cleavage of H3 at T22, a WB against H3K4me3 and total H3 along the time course (upto 14 days of treatment) must be shown.
- (5) Furthermore, how do the levels of PRDM9 change over the 14-day CMPD1 treatment? The authors show the levels of PRDM9 increase 72h post-treatment which is in-line with initial increase in H3K4me3 levels. However, in the persister H3K4me3 levels are lower. Is this due to lower PRDM9 or increased proteolysis of H3?
- (6) The blots showing histones (fig 2f, 2g, 2h) as loading controls are washed out and unclear. Better quality images are required to make informed conclusions.
- (7) The authors have included inhibitors of writers and/or erasers of H3K27me2/3 (UNC1999, GSK-J4 and KDOBA-67a) and H3K4me3 (MRK-70) in combination with CMPD1. Since, they observe a substantial increase in H3K36me3 mark in persisters, inclusion of inhibitors of writers or erasers of this mark will be necessary to delineate the contribution of these epigenetic changes in development of drug-resistance.
- (8) The authors show that MRK-70 treatment stalls cellular growth. Does this compound also cause an arrest of cells in G2/M?
- (9) The authors show that MRK-740 alone or in combination with CMPD1 dramatically reduces cholesterol biosynthetic genes and is due to reduction of H3K4me3 deposition in their gene bodies. Is this observation due to lower amounts of H3K4me3 or increased proteolysis of H3 at T22?
- (10) In the study cited on H3 proteolytic cleavage - PMID: 25394905, it is shown that CTSL1 protease is responsible for the cleavage of H3. Do the transcript levels of this protease go up in the persisters? Does treatment of cells with CTSL1 inhibitor increase the vulnerability of parent cells?
- (11) Why were RKI1 cells not used for xenograft experiments? There is no reference for GBM6, what is the evidence that xenografts of this line faithfully model glioblastoma of the patient they have been derived from? There are no histology pictures and/or molecular data (over and above EGFR status which is not specific of the tumour entity) to convince readers that this is indeed a tumour with features of glioblastoma. Also here, it is difficult to generalise findings if only one line/xenografts is used.
- (12) Would treatment of xenograft models with MT-drug in combination with higher-cholesterol diet improve the survival of mice? This is an essential point to be addressed.

Minor

(1) Statistical analysis description in the methods section need to be more detailed to inform on the type of analysis that has been performed for histone peptide analysis, RNA seq, ChipSeq and lipidomics.

Reviewer #3

(Remarks to the Author)

Reviewer #4

(Remarks to the Author)

Drug resistance continues to be the most significant hurdle to finding a cure for cancer, accounting for an estimated 90% of all cancer-related deaths. The speed with which cancer cells can gain resistance to both conventional chemotherapy and targeted medicines is alarming and there is an urgent need to fully understand the underlying mechanism drug acquired resistance in tumors. Non-genetic mechanisms have lately emerged as major causes of anticancer treatment resistance. Among these, the drug-tolerant persister cells phenotype is gaining increasingly interest as it plays a significant non-genetic role in cancer therapeutic resistance. Within this context the authors describe a drug tolerance mechanism in glioblastoma persister cells regulated by the germ-cell-specific H3K4 methyltransferase PRDM9. The topic of the manuscript is timely / pertinent, and the data here presented is sound and supports the majority of the scientific claims and findings mentioned throughout the text.

The authors have performed transcriptomics and lipidomic analysis and complementary western blot analyses aiming a better understanding the mechanistically role of PRDM9-mediated H3K4me3 in maintaining cholesterol homeostasis and contributing to cancer cells survival under chemotherapy stress. However in order to establish a holistic role of the role o PRDM9 during drug tolerance it is important that the authors complement their work with a large scale multiomics approaches specific proteomics and metabolomics. However, in order to show a comprehensive role for PRDM9 during drug tolerance, the authors should supplement their results using large-scale multiomics methods, specifically proteomics and metabolomics. Specifically, multiomics of untreated RK11 cells versus treatment with CMPD1 +/- MRK-740 and the knockout mutant. These experiments not only would validate some of the results described in the study but provide a holistic system biology overview of the impact of PRDM9 inhibition and ultimately provide an excellent opportunity to identify additional key players contributing to persisters phenotype and potential drug targets.

Reviewer #5

(Remarks to the Author)

Joun and colleagues contributed to shed light on epigenetic mechanisms underlying chemotherapy tolerance of glioblastoma persister cell, although the study is restricted to microtubule-targeting agents. Through epigenetic and transcriptomic analysis authors unveiled chemo-induction of germline methyltransferase PRDM9, and its role in promoting the metabolic rewiring of glioblastoma persister cells and consequently chemotherapy tolerance. In detail, chemotherapy induces oxidative stress and cholesterol oxidation. In this state, glioblastoma persister cells upregulate PRDM9, leading to an increase of the activating mark H3K4me3 on genes involved in de novo cholesterol biosynthesis. The inhibition of PRDM9 disrupts the cholesterol supply, leading to persister cell death.

The study is clear, well conducted, supports the conclusions and claims, and, importantly, the topic is highly relevant and of potential clinical interest.

There are few comments below that need to be addressed:

1. The authors generate glioblastoma persister cells after the treatment with CMPD1 and Tivantinib. It is not clear why the authors include also Tivantinib, if the work is focused entirely on microtubule-targeting agents, namely CMPD1?
2. The authors perform a bulk RNA-seq analysis on parental and CMPD1-derived persister cells. The analysis is limited to one cell line. It will be better to perform the same analysis on more cell lines to confirm the transcriptional profile founded in RK11 cells.
3. How do the authors explain the different trend of H3K9me3 after 3 and 6 day of CMPD1 treatment in RK11 and FPW1 cells? (fig. 2h)
4. The knockouts for SETD1A, SETD1B and PRDM9 must be assessed (e.g., by western blot) to verify the real inactivation of the genes
5. To test if CMPD1 induces mitotic arrest, it might be more quantitative to perform the analysis of cell cycle with propidium iodide.
6. Line 427: The authors state that the loss of H3K4me3 in CMPD1-derived persister cells is caused by the proteolytic cleavage of histone tails, rather than the activities of KDM5 demethylases. Did the authors check the expression of histone demethylases in glioblastoma persister cells since it was demonstrated an increased expression of KDM5A in persister cells of other tumor type (Sharma et al, 2010)?
7. Fig.2G: please improve quality
8. Fig. 3C: quality need to be improved, the quantification does not match what is visible in the figure.
9. Line 172: It is clear why UNC1999, KDOBA-67a and GSK-Ja have been selected foe co-treatment. However, it is not clear how MRK-740 was chosen since it is not included in previous screening.

Version 1:

Reviewer comments:

Reviewer #1

(Remarks to the Author)

The authors have put forth a commendable effort to thoughtfully address the various comments and concerns that were raised during the review process. Their diligent revisions have significantly enhanced the overall quality and clarity of the manuscript. The improvements made are evident and demonstrate a deep commitment to elevating the work, resulting in a substantially more coherent and comprehensive presentation of their research findings.

Reviewer #2

(Remarks to the Author)

The authors have adequately addressed the points I had raised in my review.

Reviewer #3

(Remarks to the Author)

Reviewer #4

(Remarks to the Author)

As response to my previous suggestion the authors simply stated that: "we felt that additional omics analyses were not essential for validating the oncogenic role of PRDM9-regulated cholesterol biosynthesis"

Reviewer #5

(Remarks to the Author)

The authors provided additional experiments and explanations that clarified my doubts and satisfied my revision. I do not have additional revision.

RESPONSE TO REVIEWERS' COMMENTS

Reviewer #1

1) Authors provide a very interesting observation: in drug-tolerant persisters (DTP) cells that survived 14d treatment, H3K4me3 was significantly downregulated in one DTP cell line and unaffected in the second cell line. Additionally, cholesterol was significantly downregulated in 14d treated cells. On the other hand, in cells that received only 3d of treatment, H3K4me3 was upregulated, as was cholesterol. However, the underlying reasons why DTP cells cease to depend on cholesterol and H3K4me3 after the 6th day of MTA treatment remain unexplored.

a. What are levels of PRDM9 on 14d and 6d time points? In a new Figure 4b (results section page 8) we show continuing PRDM9 mRNA increase at 3, 6, 10 and 14 days of treatment.

b. What changes occur between days 3 and 6 that reduce cell dependence on cholesterol? We would like to clarify that cholesterol levels decrease in cells treated with CMPD1 both at day 3 (Figures 7e-g) and at day 14 (Figure 1f). In addition to reduced metabolic activity in non-proliferating cells, our data show that this decrease is also driven by chemotherapy-induced oxidative stress and lipid peroxidation (Figure 7j). We are not suggesting that DTPs become independent of cholesterol between day 3 and 6; rather, we propose that PRDM9-dependent cholesterol biosynthesis provides sufficient new cholesterol to replace oxidised cholesterol and support DTP survival (Figure 7k).

c. The importance of timely implementation of PRDM9 inhibition to prevent cells from becoming resistant to MTA is a very interesting point that should be emphasized more. Based on this suggestion, we performed a DTP assay where cells were treated with CMPD1 and MRK-740 either simultaneously, pre-treated with MRK-740 for 3 days before CMPD1, or treated with MRK-740 3 days after CMPD1 (new Figure 4f). We found that both simultaneous treatment and delayed addition of MRK-740 after CMPD1 reduced the number of persister cells, whereas pre-treatment with MRK-740 had no effect. These results suggest that PRDM9 inhibition is only effective after its upregulation during CMPD1-induced mitotic arrest (Results, page 8).

2) Authors suggest that the upregulation of cholesterol helps to compensate for cholesterol oxidation due to ROS. This does not necessarily align with evidence that cholesterol is essential only up to 6d of treatment. The proposed mechanism can benefit from further validation and can solve the discrepancy indicated in the previous question. We propose that PRDM9 upregulation sustains cholesterol biosynthesis in non-proliferating cells to compensate for cholesterol oxidation induced by CMPD1; however, this does not lead to an overall increase in cholesterol levels (Figure 7e-g, CMPD1 vs Parent/NTC). We do not show evidence of cholesterol upregulation or that cholesterol is essential only up to 6 days of treatment.

a. What are the ROS levels at 3d, 6d and 14d timepoints? In the new Supplementary Figure 7f, we show a 2.5-fold increase in ROS levels following 3 days of CMPD1 treatment, which remains elevated at this level on days 6, 10, and 14 post-treatment (Results, page 13).

b. Can cell proliferation be rescued by adding antioxidants in cells treated with CMPD1 and MRK-740/PRDM9 KO? In the new Supplementary Figure 7h-i (Results, page 13), we show that adding the antioxidant N-acetyl-cysteine (NAC) increases the number of persister cells and rescues the anti-persister effects of MRK-740 and PRDM9 knockout, confirming our working model in Figure 7k. Because NAC is unstable and loses activity upon oxidation, it was replenished daily, and we quantified surviving cells after 7 days instead of the standard 14 days (DTP assay). This explains the weaker efficacy of MRK-740 and PRDM9 knockout in the Supplementary Figure 7h-i compared to MRK-740 efficacy in e.g., Figure 7h (14 days).

Minor comments:

1) Genetic manipulation to prevent cholesterol production will be a good second line of evidence to support DepMap data and data observed in the presence of inhibitor and ensure that findings are not related to the inhibitor's off-target effect. In the new Figure 7f, we show that PRDM9 KO significantly reduced cholesterol levels in RK11 cells, phenocopying the efficacy of MRK-740 (Results, page 11).

2) In Figure 2F total H3 should be presented. Immunoblots of total H3 added have been added to Figure 2f.

3) *PRDM9 is unlikely to be the main writer regulating H3K4 methylation. While the overall focus on this non-essential enzyme is well justified and explained, understanding whether other H3K4-me3 writers (SET1, MLL) are affected in a similar fashion will provide important insight into the resistance mechanism. Do MTAs activate all the writes of the H3K4-me3, or only PRDM9? This is even more important since data show that MRK-740 caused H3K4me3 depletion only in CMPD1-treated cells but not in naïve ones.* We compared the expression of eight KMT2 methyltransferases (which methylate H3K4), including MLL1 (KMT2A), SETD1A (KMT2F), and SETD1B (KMT2G), to PRDM9 in RKI1 and FPW1 persisters. PRDM9 showed the highest upregulation, followed by MLL1, while the expression of other KMT2 members remained unchanged (new Figure 4a, Results page 8), supporting PRDM9 as the major H3K4 methyltransferase in our glioblastoma models.

4) *It would be beneficial to additionally validate the safety of newly developed compound in normal human astrocytes and neural stem cells.* We were unable to test in normal human astrocytes (NHA), as Lonza, the main supplier, has discontinued import of NHAs to Australia. Instead, we evaluated our novel chemotherapy agent WJA88 in combination with MRK-740 and LXR-623 in mouse-derived neural progenitor cells and astrocytes (new Supplementary Figure 8g). WJA88 is a microtubule-targeting agent and, like clinical MTAs paclitaxel and vinblastine, is expected to inhibit the proliferation of all dividing cells. Consistent with this, WJA88 suppressed the proliferation of neural stem cells, which are highly proliferative in culture. However, MRK-740 and LXR-623, either alone or in combination, had no effect on neural stem cell viability and did not enhance WJA88 efficacy. Importantly, in astrocytes, none of the treatments, whether alone or combined, affected cell viability. These results suggest that WJA88 may have a safety profile similar to that of clinical MTAs (Results, page 14).

5) *In Figure 3 I, the effect of the KO should be analyzed in relation to the NT control and not the parent cell line. If NT and transfection cause cytotoxicity, that has to be considered when judging the effect of PRDM9 KO.* In addition to 3 data points presented in the original Figure 3I (now Figure 4k–l), we repeated the DTP assays using NTC and PRDM9 knockout cells three more time. NTC transfection caused evident cytotoxicity in 1 out of 6 repeats. When compared to NTC controls, PRDM9 knockout significantly reduced DTP survival in the PRDM9(1) KO line, with a p-value of 0.098 observed for PRDM9(2) KO.

6) *Having representative images of the tumour bioluminescence in animal model will make in vivo data more impactful.* We agree that bioluminescence images can enhance in vivo data; however, we primarily use it as a guide for starting treatments and monitor general trends. Changes in hypoxia, vascularisation, immune infiltration, or extracellular pH induced by treatments can provide a false impression of treatment response when measured by imaging. Survival, in contrast, is an unambiguous outcome measure and our preferred readout. Further, this experiment was conducted during Covid restrictions, limiting imaging frequency due to reduced animal facility access. We provide images from the electroporation glioblastoma model we have done for the rebuttal experiments (new Figure 8d-e).

7) *Some western blots (Figure 2 E and G, Figure 3 E, Figure 6 B - C) lack quantification, which should be fixed to provide statistical robustness.* All Western blots presented in the main figures are accompanied by corresponding quantification in the Supplementary Figures. For easy reference, each figure legend includes a note indicating where the related quantification can be found.

8) *For the cases where densitometry quantification for the western blots is provided, were these numbers normalized to the total protein and/or total histone levels before being compared between groups?* All immunoblots in Figure 2 and Figure 3e were normalized to total H3 before comparison between groups. All other immunoblots were normalised to housekeeping proteins (Lamin B1, GAPDH, or HSP90). This normalisation method is indicated in the figure legends of the Supplementary Figures presenting the quantification data.

Reviewer #2

(1) The authors demonstrate down regulation of cholesterol biosynthetic genes in RKI1-persister cells compared to the parent cells on a transcriptomic level and further confirm the functional impact by quantifying the lipidomic profile. Does this observation hold true for the second set of cells (FPW1)

used in this study? FPW1 cells were less sensitive to all interventions tested in this study. As shown in new Figure 7i, exogenous cholesterol supplementation increased DTP numbers by only 1.3-fold in FPW1 cells, compared to a 10-fold increase in RKI1 cells (Figure 7h). However, cholesterol supplementation rescued the anti-persister efficacy of PRDM9 inhibition with MRK-740 (Figure 7i), and MRK-740 significantly reduced cholesterol levels in FPW1 cells (Figure 7g). We extended our study to the MMK1 cell line: PRDM9 inhibition downregulated cholesterol biosynthesis enzymes (new Figure 7d), cholesterol supplementation did not increase DTP numbers but rescued MRK-740 efficacy (new Figure 7i). Further, chemotherapy upregulated PRDM9 mRNA in six glioblastoma stem cell lines (Figure 4c), and MRK-740 showed anti-persister efficacy in five genetically diverse glioblastoma cell lines (Extended Figure 3h). Together, these findings support a broader role for PRDM9 in regulating cholesterol biosynthesis across glioblastoma models. While persister cells in all three cell lines depend on cholesterol supply, this dependency appears stronger in RKI1 and MMK1 persisters. In contrast, FPW1 persisters may rely on additional survival mechanisms beyond cholesterol. This has been added to the Discussion on page 15.

RT-PCR of cholesterol biosynthetic enzymes in persister vs. parent FPW1 will be necessary to understand the changes in metabolism associated with developing MT-drug tolerance. RNA sequencing was performed on RKI1 and FPW1 parental and persister cells, revealing that transcripts for cholesterol biosynthesis enzymes are predominantly upregulated in persister cells (new Supplementary Figure 6e), consistent with the transcriptomic profile of cells treated with CMPD1 for 3 days. Despite this transcriptional upregulation, cholesterol levels are reduced after both 3 and 14 days of CMPD1 treatment (persisters). This supports our working model that the decrease in cholesterol is caused by oxidation, and that, to survive, persister cells compensate by upregulating cholesterol biosynthesis pathways (Results, page 12-13).

(2) The authors have profiled the histone extracts of parent and persister cells. It will be very useful for the scientific community to have the information on all histone peptides identified in this experiment in the form of a table. The mass spectrometry proteomics data, including all histone peptides identified in persister cells, have been deposited in the ProteomeXchange Consortium via the PRIDE partner repository under the dataset identifier PXD050643. This will allow the scientific community full access to the complete list of histone peptides identified in persisters.

(3) In fig 2f, the faster migration H3 band is missing in H3K27me3 and H3K27me2 blot. Can this be explained? Alternatively, a blot for total H3 in parent vs. persister will be helpful in making the point of proteolytically cleaved isoform and potentially the proportions of two isoforms present in the persistent cells. We have added total H3 immunoblots to Figure 2f, which also show a faster-migrating band consistent with cleaved histone H3. Accurately quantifying the proportion of full-length versus cleaved H3 is challenging, as the bands run closely together, making separate quantification difficult.

The absence of the faster-migrating band in the H3K27me2 and H3K27me3 blots is likely due to lower antibody sensitivity. Similarly, this band is not visible in the total H3 immunoblots in Figures 2g-h, which were generated using a different lot of the total H3 antibody (same supplier, different batch). Although cleaved H3 is clearly detected with the specific Cleaved H3 (T22) antibody, it is not apparent in the total H3 blots (new Figure 2h). This also suggests that the amount of cleaved H3 is substantially lower than that of full-length H3.

(4) When does the proteolytically cleaved H3 variant appear within the 14-day treatment window. To understand the dynamics of H3K4me3 loss vs. proteolytic cleavage of H3 at T22, a WB against H3K4me3 and total H3 along the time course (up to 14 days of treatment) must be shown. New Figure 2h shows the dynamics of H3K4me3, H3K9me3, and H3 cleavage at days 3, 6, 10, and 14 of treatment. Histone cleavage begins as early as day 3, peaks at day 10, and then declines. H3K4me3 levels display distinct patterns in RKI1 and FPW1 cells but are elevated at days 3, 6, and 10, then steeply decline. In contrast, changes in H3K9me3 are minimal, also aligning with Figure 2f. Together, these data suggest that H3K4 methylation and its removal via histone cleavage may occur concurrently, but the net effect is an overall increase in H3K4me3 up to day 10 in both cell lines (Results, page 5-6).

(5) Furthermore, how do the levels of PRDM9 change over the 14-day CMPD1 treatment? The authors show the levels of PRDM9 increase 72h post-treatment which is in-line with initial increase in H3K4me3 levels. However, in the persister H3K4me3 levels are lower. Is this due to lower PRDM9 or increased proteolysis of H3? New Figure 4b shows a time course (days 3, 6, 10, and 14) of progressively increasing PRDM9 mRNA levels. However, this does not correlate with H3K4me3 dynamics (Figure 2h), as H3K4me3 abundance first increases and then declines. Data in the new Supplementary Figure 3c–d demonstrate an increase in the H3K4 demethylase KDM5 and the histone-cleaving protease CTSL1 upon CMPD1 treatment. Hence, both enzymes may contribute to the decline in H3K4me3 observed at later time points (Results, page 7).

(6) The blots showing histones (fig 2f, 2g, 2h) as loading controls are washed out and unclear. Better quality images are required to make informed conclusions. The loading controls were Amido Black stains of membranes but have been replaced with total H3 immunoblots.

(7) The authors have included inhibitors of writers and/or erasers of H3K27me2/3 (UNC1999, GSK-J4 and KDOBA-67a) and H3K4me3 (MRK-740) in combination with CMPD1. Since, they observe a substantial increase in H3K36me3 mark in persisters, inclusion of inhibitors of writers or erasers of this mark will be necessary to delineate the contribution of these epigenetic changes in development of drug-resistance. New Figure 3c shows that inhibiting the H3K36 methyltransferase SETD2 with EPZ-719 reduces persister cell survival, particularly in RK11 cells, which carry the SETD2 R472H mutation, while FPW1 cells are SETD2 wild-type. As this mutation is unique to RK11 and not present in any of the other 11 cell lines in our QCell panel, the efficacy of EPZ-719 may represent a cell line-specific effect rather than a broadly applicable mechanism. This prompted us to focus further on PRDM9 and the anti-persister efficacy of MRK-740 (clarified in Results, page 6).

H3K36me3 is demethylated by KDM4, and while we initially tested the KDM4 inhibitor QC6352 (Supplementary Figure S3a), we recently found that its anti-cancer activity is driven by an off-target (manuscript under review/revision, *Nature Chemical Biology*). Due to its promiscuity, we chose not to include QC6352 data here, as its efficacy cannot be solely attributed to KDM4 inhibition.

(8) The authors show that MRK-70 treatment stalls cellular growth. Does this compound also cause an arrest of cells in G2/M? New Supplementary Figure 3e shows that MRK-740 increases the percentage of cells in the G₀/G₁ phase while reducing the proportion of cells in S phase (Results, page 7).

(9) The authors show that MRK-740 alone or in combination with CMPD1 dramatically reduces cholesterol biosynthetic genes and is due to reduction of H3K4me3 deposition in their gene bodies. Is this observation due to lower amounts of H3K4me3 or increased proteolysis of H3 at T22? Based on H3K4me3 ChIP-seq results, we propose that the primary mechanism involves reduced H3K4me3 deposition at cholesterol biosynthesis gene promoters (Figure 5i). However, we tested whether MRK-740 induces H3 cleavage - and found that it does not (data on the right, not included in the paper).

(10) In the study cited on H3 proteolytic cleavage - PMID: 25394905, it is shown that CTSL1 protease is responsible for the cleavage of H3. Do the transcript levels of this protease go up in the persisters? Does treatment of cells with CTSL1 inhibitor increase the vulnerability of parent cells? CTSL1 transcript levels increased over the course of CMPD1 treatment, reaching a maximum increase in DTPs (day 14; new Supplementary Figure 3d). However, inhibition of CTSL1 did not reduce the number of persister cells (new Figure 3e-f).

(11) Why were RK11 cells not used for xenograft experiments? There is no reference for GBM6, what is the evidence that xenografts of this line faithfully model glioblastoma of the patient they have been derived from? There are no histology pictures and/or molecular data (over and above EGFR status which is not specific of the tumour entity) to convince readers that this is indeed a tumour with features of glioblastoma. Also here, it is difficult to generalise findings if only one line/xenografts is used. The median survival for RK11 xenografts is 248 days - the longest among the 12 glioblastoma stem cell lines in our QCell panel. BAH1 and MMK1 xenografts show median survivals of 210 and 157 days, respectively (Ref #31; *Sci Rep* 9, 4902, 2019). Based on our findings that WJA88+LXR-

623 significantly extended survival in the aggressive GBM6 model (nearly doubling it), we anticipated a similarly pronounced effect in RKI1/BAH1/MMK1 xenografts, which would require an extremely lengthy study. We used GBM6, a well-established PDX model that closely recapitulates key biological and clinical features of human GBM, as supported by references #48, 49, 50 cited in the manuscript (Results, page 15). GBM6 tumours grow rapidly in vivo and exhibit highly aggressive behaviour, making them suitable for time-efficient experimental workflows. In addition, GBM6 model was previously employed in studies of brain-penetrant microtubule-targeting agents such as lisavanbulin (Ref #48, Neuro Oncol 24, 384, 2022), which supports comparability across studies.

In the revised manuscript, we present data obtained with a new glioblastoma model generated by brain electroporation and knockout of the three most commonly mutated genes in mesenchymal glioblastoma (*Nf1*, *Trp53*, *Pten*). This model produces tumours that closely recapitulate the histopathological and molecular features of human glioblastoma within 3 weeks (Ref #45, 46: STAR Protoc 5, 102928, 2024; Cell Rep 42, 112472, 2023). To our knowledge, this model has not previously been used to test new therapeutic modalities.

We applied it to evaluate the combination of WJA88 and LXR-623, following the same treatment protocol as in the GBM6 mice. However, because the animals in this model are young pups, unlike the adult mice used in the GBM6 model, they did not tolerate twice-daily oral gavage administration of LXR-623, and this treatment arm had to be terminated. We did, however, establish the effect of WJA88 alone in reducing the pool of proliferating cells in these tumours, confirming both in vivo target engagement and safety (new Figure 8d-e, Supplementary Figure S8d).

As LXR-623 was used solely as a surrogate for the PRDM9 inhibitor MRK-740 (unsuitable for in vivo use) to generate proof-of-concept, and given both our inability to administer it in young mice and its previous failure in human clinical trials, we believe that further investment of time and resources into animal models would not be justified or ethically appropriate. The aim of this study is to present proof-of-concept evidence for PRDM9 as a vulnerability in persister cells and stimulate the development of clinically viable PRDM9 inhibitors.

(12) Would treatment of xenograft models with MT-drug in combination with higher-cholesterol diet improve the survival of mice? This is an essential point to be addressed. Cholesterol supplementation to the cell culture media increases persister survival (Figure 7h; new Figure 7i), so we would expect that animals on a high-cholesterol diet would have more persister cells post-chemotherapy, leading to shorter survival and/or more aggressive tumour recurrence due to an expanded DTP pool. However, as the blood-brain barrier is impermeable to peripheral cholesterol, glioblastomas are unlikely to be affected by a high-cholesterol diet. This is included on page 12.

Minor

(1) Statistical analysis description in the methods section need to be more detailed to inform on the type of analysis that has been performed for histone peptide analysis, RNA seq, ChipSeq and lipidomics. Statistical considerations for specific experiments including RNA-seq, ChIP-seq, histone peptide LC-MS/MS, and lipidomics have been included in the Data Analysis section on page 32.

Reviewer #4

The authors have performed transcriptomics and lipidomic analysis and complementary western blot analyses aiming a better understanding the mechanistically role of PRDM9-mediated H3K4me3 in maintaining cholesterol homeostasis and contributing to cancer cells survival under chemotherapy stress. However in order to establish a holistic role of the role o PRDM9 during drug tolerance it is important that the authors complement their work with a large scale multiomics approaches specific proteomics and metabolomics. Specifically, multiomics of untreated RK11 cells versus treatment with CMPD1 +/- MRK-740 and the knockout mutant. These experiments not only would validate some of the results described in the study but provide a holistic system biology overview of the impact of PRDM9 inhibition and ultimately provide an excellent opportunity to identify additional key players contributing to persisters phenotype and potential drug targets. We agree that a multi-omics approach would provide valuable insights into the role of PRDM9 in glioblastoma and potentially other cancers, and we plan to further explore this in future studies. However, as this manuscript already presents a substantial amount of data, including 8 main figures and 8 supplementary figures,

each with at least 8 panels; we felt that additional omics analyses were not essential for validating the oncogenic role of PRDM9-regulated cholesterol biosynthesis.

Reviewer #5

1. The authors generate glioblastoma persister cells after the treatment with CMPD1 and Tivantinib. It is not clear why the authors include also Tivantinib, if the work is focused entirely on microtubule-targeting agents, namely CMPD1? Tivantinib was originally developed as a c-Met inhibitor, but later studies revealed its anti-cancer activity is due to microtubule targeting (ref #23). Similarly, CMPD1 was developed as an MK2 inhibitor, but we validated its mechanism of action via targeting microtubules (ref #21). Therefore, both CMPD1 and tivantinib function as small-molecule MTAs, and tivantinib was included as a control. This is now clarified in the Results on the page 4.

2. The authors perform a bulk RNA-seq analysis on parental and CMPD1-derived persister cells. The analysis is limited to one cell line. It will be better to perform the same analysis on more cell lines to confirm the transcriptional profile founded in RKI1 cells. We performed transcriptomic analysis of persisters derived from FPW1 cells (new Supplementary Figure 1g, h) and compared DEGs with those from RKI1-derived persisters. The analysis revealed a high degree of correlation and overlap (new Figure 1g, new Supplementary Figure 1i), suggesting a shared persister transcriptional profile (Results, page 5). This is particularly noteworthy given the completely distinct genetic backgrounds of the two cell lines: RKI1 (MYC amplification; ATRX D808G; SETD2 R472H) and FPW1 (PTEN R130Q; CDKN2A/2B deletion; NF1 splice mutation).

3. How do the authors explain the different trend of H3K9me3 after 3 and 6 day of CMPD1 treatment in RKI1 and FPW1 cells? (fig. 2h) As noted above, RKI1 and FPW1 have distinct genetic backgrounds, which can influence histone methylation patterns. In the original Figure 2h, RKI1 cells showed a decrease in H3K9me3 (n=2), while FPW1 cells showed an apparent increase, albeit with high variability between replicates. These data were obtained from whole cell lysates. Given the semi-quantitative nature of immunoblotting, the potentially transient dynamics of H3K9 methylation, and the genetic differences between the two cell lines, variability in their H3K9me3 response is not unexpected. Yet, to clarify this, we extended and repeated the time-course experiment and used histone extracts for better quality. The immunoblots in the new Figure 2h have been updated to show H3K9me3 levels at days 3, 6, 10, and 14. Again, we observed variability in H3K9me3 signal following CMPD1 treatment, and the quantification revealed only minimal overall changes in both RKI1 and FPW1 cells, consistent with the findings in Figure 2f (Results, page 5-6).

4. The knockouts for SETD1A, SETD1B and PRDM9 must be assessed (e.g., by western blot) to verify the real inactivation of the genes. Knockout of SETD1A, a common essential gene, was detrimental to RKI1 cells, leaving no viable cells for analysis. Immunoblots of successful knockout of the non-essential genes SETD1B and PRDM9 are shown the new Figures 4i-j (Results, page 8).

5. To test if CMPD1 induces mitotic arrest, it might be more quantitative to perform the analysis of cell cycle with propidium iodide. New Figure 4d shows mitotic arrest induced by CMPD1 and Rho-3306 (Results, page 8).

6. Line 427: The authors state that the loss of H3K4me3 in CMPD1-derived persister cells is caused by the proteolytic cleavage of histone tails, rather than the activities of KDM5 demethylases. Did the authors check the expression of histone demethylases in glioblastoma persister cells since it was demonstrated an increased expression of KDM5A in persister cells of other tumor type (Sharma et al, 2010)? New Supplementary Figure S3c shows that KDM5A mRNA levels remain unchanged at day 3 and 6 but increase at days 10 and 14 (DTP time point) post-CMPD1 treatment. New Supplementary Figure 3d shows that CTSL1 mRNA expression begins to increase from day 3, and immunoblotting shows histone cleavage as early as day 3. This suggests that the H3K4me3 may be driven by both KDM5A and CTSL1. However, since inhibition of either CTSL1 or KDM5 did not eliminate persisters (Figure 3a, new Figure 3f), we did not investigate further which enzyme, if not both, contributes to the H3K4me3 decrease in persisters. Importantly, our findings indicate that early changes in response to chemotherapy, *i.e.* increased H3K4 methylation, rather than alterations found in established persister cells (*i.e.* H3K4me3 demethylation), are the key targets for preventing the emergence of glioblastoma persister cells. This is discussed on page 15.

7. Fig.2G: please improve quality. The original Figure 2g included immunoblots showing histone cleavage in DTPs, with Amido Black staining used as the loading control (hence the washed-out appearance). Since these samples were not run using the total H3 antibody, and because we performed time-course experiments for histone cleavage, where day 14 corresponds to the DTP samples, we removed the original DTP histone cleavage blots. The time-course immunoblots in the new Figure 2h show histone cleavage across multiple time points, including day 14 (DTP timepoint).

8. Fig. 3C: quality need to be improved, the quantification does not match what is visible in the figure. The quantification corresponds to the right panel of the original Figure 3c, showing CMPD1 ± MRK-740 ± MRK-740-NC treatments. Untreated cells and those treated with MRK-740 or MRK-740-NC alone (left panel of the original Figure 3c) were not quantified, and were included only as controls to demonstrate that MRK-740 and MRK-740-NC alone do not inhibit proliferation. To avoid confusion, the single-agent MRK-740 and MRK-740-NC images have been moved to Supplementary Figure 3h. We quantified colony areas only for the CMPD1, CMPD1 + MRK-740, and CMPD1 + MRK-740-NC treatment groups, showing that co-treatment with MRK-740 (but not MRK-740-NC) significantly reduces colony formation compared to CMPD1 alone (set as 1 in the quantification). Figure 3c has also been updated to include additional cell line (MMK1) and is now presented as Figure 3h.

9. Line 172: It is clear why UNC1999, KDOBA-67a and GSK-J4 have been selected for co-treatment. However, it is not clear how MRK-740 was chosen since it is not included in previous screening. MRK-740 was included in the initial screen shown in Supplementary Figure 3a but was tested only in RKI1 cells. It showed no efficacy, which is consistent with our later findings that there is a limited therapeutic window for targeting PRDM9, likely most effective when administered concurrently (new Figure 4f). For the DTP assays presented in Figure 3a–d, we selected epigenetic inhibitors targeting writers and erasers of H3K4, H3K27, and H3K36 marks (clarified in the manuscript, page 6):

UNC1999 targets H3K27 methyltransferase KMT6 (EZH2)

GSK-J4 targets H3K27 demethylases KDM6

MRK-740 targets H3K4 methyltransferase PRDM9

BAY-6035 targets H3K3 methyltransferase SMYD3

KDOBA-67a targets H3K4 demethylase KDM5

EPZ-719 targets H3K36 methyltransferase SETD2

H3K36me3 is demethylated by KDM4, and while we initially tested the KDM4 inhibitor QC6352 (Supplementary Figure S3a), we later found that its anti-cancer activity is driven by an off-target effect (manuscript under revision, *Nature Chemical Biology*). Due to its promiscuity, we chose not to include QC6352 in this DTP study, as its efficacy cannot be solely attributed to KDM4 inhibition.